# Improved Last-Iterate Convergence of Shuffling Gradient Methods for Nonsmooth Convex Optimization

**Zijian Liu** [1]  **Zhengyuan Zhou** [1] [2]

## Abstract

We study the convergence of the shuffling gradient method, a popular algorithm employed to minimize the finite-sum function with regularization, in which functions are passed to apply (Proximal) Gradient Descent (GD) one by one whose order is determined by a permutation on the indices of functions. In contrast to its easy implementation and effective performance in practice, the theoretical understanding remains limited. A recent advance by (Liu & Zhou, 2024b) establishes the first last-iterate convergence results under various settings, especially proving the optimal rates for smooth (strongly) convex optimization. However, their bounds for nonsmooth (strongly) convex functions are only as fast as Proximal GD. In this work, we provide the first improved last-iterate analysis for the nonsmooth case demonstrating that the widely used Random Reshuffle (RR) and Single Shuffle (SS) strategies are both provably faster than Proximal GD, reflecting the benefit of randomness. As an important implication, we give the first (nearly) optimal convergence result for the suffix average under the RR sampling scheme in the general convex case, matching the lower bound shown by (Koren et al., 2022).

## 1. Introduction

This work considers a common machine learning problem, minimizing a finite-sum function with regularization, i.e.,

$$\min_{\boldsymbol{x} \in \mathbb{R}^d} F(\boldsymbol{x}) \triangleq f(\boldsymbol{x}) + \psi(\boldsymbol{x}) \text{ where } f(\boldsymbol{x}) \triangleq \frac{1}{n} \sum_{i=1}^{n} f_i(\boldsymbol{x}),$$

in which $f_i$ and $\psi$ are convex and potentially satisfy other properties, e.g., Lipschitz continuity. Due to the famous empirical risk minimization framework (Shalev-Shwartz & Ben-David, 2014), such a problem arises in a wide range of applications (e.g., SVMs (Cortes & Vapnik, 1995)) and has been extensively studied in the past few years.

Two text-book level algorithms for solving the problem are Proximal Gradient Descent (GD) and its variant Proximal Stochastic Gradient Descent (SGD) (Nemirovski & Yudin, 1983; Nesterov et al., 2018; Bubeck et al., 2015; Lan, 2020), where the former requires a true gradient in every step in contrast to the latter only computing the gradient of a single function selected based on a random index uniformly sampled from $[n] \triangleq \{1, \cdots, n\}$.

Whereas neither of the above classic algorithms is widely adopted in practice since, when $n$ is large (the standard scenario nowadays), Proximal GD incurs large computational overhead and Proximal SGD suffers from cache misses. Instead, the shuffling gradient method is arguably the most popular and practical choice, in which functions (or data points) are passed to apply (proximal) gradient descent one by one whose order is determined by a permutation on $[n]$. In particular, three shuffling strategies named Random Reshuffle (RR), Single Shuffle (SS), and Incremental Gradient (IG) are mostly used, where the permutation varies randomly in every epoch (containing $n$ steps) for RR, is randomly sampled at the beginning and employed through all updates for SS, and is deterministically picked in advance for IG.

Compared to its easy implementation, lightweight computation, and effective performance (Bottou, 2009; 2012; Bengio, 2012), the theoretical understanding of the shuffling gradient method remains limited, especially for the most common output, the last iterate. A recent advance by (Liu & Zhou, 2024b) establishes the first last-iterate convergence results measured by the function value gap under various settings, particularly, proving the optimal rates for smooth (strongly) convex optimization under the RR/SS/IG sampling schemes mentioned before. However, their bounds for nonsmooth (strongly) convex functions are proved for any kind of shuffling strategy (not limited to RR/SS/IG) and only as fast as Proximal GD, leaving the following unaddressed research question as also mentioned by (Liu & Zhou, 2024b):

[1]Stern School of Business, New York University [2]Arena Technologies. Correspondence to: Zijian Liu <zl3067@stern.nyu.edu>.

*Proceedings of the 42nd International Conference on Machine Learning*, Vancouver, Canada. PMLR 267, 2025. Copyright 2025 by the author(s).

*Table 1.* Summary of our new convergence rates and the best-known upper/lower bounds under different settings when $T = Kn$ where $K \in \mathbb{N}$. All results use the function value gap as the convergence measurement. In the "Shuffling" column, ANY means the rate in the same row holds for any type of shuffling scheme not limited to RR/SS/IG. In the "Rate" column, $D_\star \triangleq \|\boldsymbol{x}_\star - \boldsymbol{x}_1\|$ denotes the Euclidean distance (or any upper bound on it) from the optimal solution $\boldsymbol{x}_\star$ and the initial point $\boldsymbol{x}_1$. $\wedge$ and $\vee$ indicate $\min$ and $\max$ operations, respectively. In the last column, $\boldsymbol{x}_{Kn+1}^{\mathsf{avg}} \triangleq \frac{1}{Kn} \sum_{t=1}^{Kn} \boldsymbol{x}_{t+1}$ and $\boldsymbol{x}_{Kn+1}^{\mathsf{suffix}} \triangleq \frac{1}{n} \sum_{t=Kn-n+1}^{Kn} \boldsymbol{x}_{t+1}$ respectively refer to the average iterate and the suffix average of the last one epoch.

| $F = f + \psi$ where $f = \frac{1}{n}\sum_{i=1}^{n} f_i$, each $f_i$ is convex and $G$-Lipschitz, and $\psi$ is $\mu$-strongly convex | | | | |
|---|---|---|---|---|
| Setting | Shuffling | Reference | Rate | Output |
| $\mu = 0$ | ANY | (Liu & Zhou, 2024b) | $O\left(\frac{GD_\star}{\sqrt{K}}\right)^a$ | $\boldsymbol{x}_{Kn+1}$ |
| | RR | (Koren et al., 2022) | $O\left(\frac{GD_\star}{n^{1/4}\sqrt{K}}\right)^b$ | $\boldsymbol{x}_{Kn+1}^{\mathsf{avg}}$ |
| | | **Ours** (Theorem 4.2) | $\widetilde{O}\left(\frac{GD_\star}{n^{1/4}\sqrt{K}}\right)^c$ | $\boldsymbol{x}_{Kn+1}$ |
| | | **Ours** (Corollary 4.3) | $\widetilde{O}\left(\frac{GD_\star}{n^{1/4}\sqrt{K}}\right)$ | $\boldsymbol{x}_{Kn+1}^{\mathsf{suffix}}$ |
| | SS | (Koren et al., 2022) | $O\left(\frac{GD_\star}{n^{1/4}K^{1/4}} \vee \frac{GD_\star}{\sqrt{n}}\right)^b$ | $\boldsymbol{x}_{Kn+1}^{\mathsf{avg}}$ |
| | | **Ours** (Theorem 4.5) | $\widetilde{O}\left(\frac{GD_\star}{n^{1/4}K^{1/4}} \vee \frac{GD_\star}{\sqrt{n}}\right)$ | $\boldsymbol{x}_{Kn+1}$ |
| | | **Ours** (Theorem 4.6) | $\widetilde{O}\left(\frac{GD_\star}{n^{1/4}K^{1/4}} \wedge \frac{GD_\star}{\sqrt{K}}\right)^b$ | $\boldsymbol{x}_{Kn+1}$ |
| | RR/SS | (Koren et al., 2022) | $\Omega\left(\frac{1}{n^{1/4}\sqrt{K}}\right)^d$ | $\boldsymbol{x}_{Kn+1}^{\mathsf{suffix}}$ |
| $\mu > 0$ | ANY | (Liu & Zhou, 2024b) | $\widetilde{O}\left(\frac{\mu D_\star^2}{K^2} + \frac{G^2}{\mu K}\right)^a$ | $\boldsymbol{x}_{Kn+1}$ |
| | RR | **Ours** (Theorem 4.4) | $\widetilde{O}\left(\frac{\mu D_\star^2}{n^2 K^2} + \frac{G^2}{\mu\sqrt{n}K}\right)$ | $\boldsymbol{x}_{Kn+1}$ |
| | SS | **Ours** (Theorem 4.7) | $\widetilde{O}\left(\frac{\mu D_\star^2}{n^2 K^2} + \frac{G^2}{\mu\sqrt{n}K} + \frac{G^2}{\mu n}\right)$ | $\boldsymbol{x}_{Kn+1}$ |

---

[a] The same rates hold for the last iterate of Proximal GD when the gradient budget is $Kn$. Also, we remark that these rates cannot apply to our Algorithm 1 once $\psi \neq 0$ due to the difference from the method studied in (Liu & Zhou, 2024b). See Section 3 for details.

[b] These rates are proved under $\psi = I_{\mathcal{C}}$ where $I_{\mathcal{C}}$ is the characteristic function for the nonempty closed convex set $\mathcal{C}$ in $\mathbb{R}^d$.

[c] This rate can automatically improve to $O\left(\frac{GD_\star}{n^{1/4}\sqrt{K}}\right)$, i.e., no extra logarithmic factor, if $K = \Omega(\log n)$.

[d] This bound is built under $G = 4$, $D_\star = 1$, and $\psi = I_{\mathcal{C}}$ for $\mathcal{C}$ being the unit ball centered at $\boldsymbol{0}$. See also discussions in Subsection 1.2.

*For nonsmooth (strongly) convex optimization, can we prove better last-iterate convergence rates than Proximal GD for RR/SS to reflect the benefit of randomness?*

### 1.1. Our Contributions

We answer the question affirmatively by establishing the first improved last-iterate convergence rates for nonsmooth (strongly) convex optimization under both RR and SS sampling schemes, as summarized in Table 1.

For RR, our new rates are better than the best-known bounds in (Liu & Zhou, 2024b) by up to a factor of $\Theta(n^{-1/4})$ in the general convex case and a factor of $\Theta(n^{-1/2})$ in the strongly convex case. As such, our results provide the first concrete evidence indicating that the RR sampling scheme indeed converges faster than Proximal GD, reflecting the benefit of randomness. As an important implication, we give the first provable and (nearly) optimal convergence result for the suffix average of the last $n$ points in the optimization

trajectory, matching the lower bound shown by (Koren et al., 2022) and thus filling in the gap.

For SS, our new rates are better than the bounds in (Liu & Zhou, 2024b) when the time horizon is below a certain threshold. Specifically, suppose $T = Kn$ where $K \in \mathbb{N}$, then there exists a critical value $K_\star \in (1, n]$ such that once $K \leq K_\star$, our bounds decay faster than (Liu & Zhou, 2024b) (and also Proximal GD) for both general and strongly convex optimization. In the special case of constrained optimization (i.e., $\psi = I_{\mathcal{C}}$ where $I_{\mathcal{C}}$ is the characteristic function for the nonempty closed convex set $\mathcal{C}$ in $\mathbb{R}^d$), we sharpen our bound further and obtain an improved rate better than (Liu & Zhou, 2024b) for any $K \in \mathbb{N}$. These results suggest that the SS strategy also beats Proximal GD (at least partially), demonstrating the benefit of using random permutations.

We also highlight that our results (except Theorem 4.6) hold for *any* $T \in \mathbb{N}$, which as far as we know is new in the literature on shuffling gradient methods for convex optimization.

Moreover, we propose a novel sufficient condition to guarantee the last-iterate convergence when the index is selected in a general manner, not limited to shuffling-based methods.

## 1.2. Related Work

Due to limited space, we will only review the study of shuffling-based gradient methods for nonsmooth (strongly) convex optimization. For details and progress in smooth optimization, the reader could refer to (Gurbuzbalaban et al., 2019; Gürbüzbalaban et al., 2021; Ying et al., 2018; Haochen & Sra, 2019; Nagaraj et al., 2019; Rajput et al., 2020; Safran & Shamir, 2020; Ahn et al., 2020; Mishchenko et al., 2020; Nguyen et al., 2021; Safran & Shamir, 2021; Rajput et al., 2022; Mishchenko et al., 2022; Tran et al., 2022; Cha et al., 2023; Cai et al., 2024; Cai & Diakonikolas, 2025) for the convex case and (Solodov, 1998; Li et al., 2019; Mishchenko et al., 2020; Nguyen et al., 2021; Tran et al., 2021; Pauwels, 2021; Lu et al., 2022b; Mohtashami et al., 2022; Lu et al., 2022a; Li et al., 2023; Nguyen & Tran, 2023; Yu & Li, 2023; Qiu et al., 2023; Koloskova et al., 2024; Qiu & Milzarek, 2024; Josz et al., 2024) for the nonconvex case.

In the following, we assume the time horizon $T = Kn$ where $K \in \mathbb{N}$ for simplicity and only focus on the dependence of $n$ and $K$ in the convergence rate.

**Upper Bound.** In the general convex case, the first $O\left(\frac{1}{\sqrt{K}}\right)$ rate is established by (Nedic & Bertsekas, 2001) for IG when $\psi = I_{\mathcal{C}}$. Later in (Bertsekas, 2011), the requirement $\psi = I_{\mathcal{C}}$ is relaxed to $\psi = \varphi + I_{\mathcal{C}}$ where $\varphi$ needs to be Lipschitz on $\mathcal{C}$. (Shamir, 2016) studies the generalized linear model and provides the $O\left(\frac{1}{\sqrt{n}}\right)$ upper bound for RR/SS when $K = 1$. As for general objectives and general $K \in \mathbb{N}$, (Koren et al., 2022) is the only work showing the $O\left(\frac{1}{n^{1/4}\sqrt{K}}\right)$ rate for RR and the $O\left(\frac{1}{n^{1/4}K^{1/4}} \vee \frac{1}{\sqrt{n}}\right)$ rate for SS, both under $\psi = I_{\mathcal{C}}$. However, all the results mentioned until now only work for the average iterate. Recently, (Liu & Zhou, 2024b) gives the first last-iterate convergence result $O\left(\frac{1}{\sqrt{K}}\right)$ being applied to any convex $\psi$ and any type of shuffling strategy not limited to RR/SS/IG.

In the strongly convex case, (Kibardin, 1979; Nedić & Bertsekas, 2001) show the IG sampling scheme guarantees the $O\left(\frac{1}{K}\right)$ convergence measured by the squared distance from the optimal solution and the last iterate, assuming strongly convex $f$ and $\psi = I_{\mathcal{C}}$. (Liu & Zhou, 2024b) proves a similar $\widetilde{O}\left(\frac{1}{K}\right)$ rate for strongly convex $\psi$ with improvements in two aspects: one is using a stronger criterion, the function value gap, to measure convergence, the other is that their result holds for any shuffling scheme not restricted to RR/SS/IG.

**Lower Bound.** The lower complexity bound of shuffling

gradient methods for nonsmooth (strongly) convex optimization is a long-open problem. The first insightful observation is by (Nagaraj et al., 2019) pointing out that any lower bound established for the deterministic case is also valid here as one can take $f_i \equiv f$ and $\psi = I_{\mathcal{C}}$ where $\mathcal{C}$ is a certain convex set (usually a ball in $\mathbb{R}^d$ centered at $\mathbf{0}$). Such a reduction immediately implies two results (not limited to the last iterate) working for any shuffling strategy, i.e., $\Omega\left(\frac{1}{\sqrt{nK}}\right)$ for the general convex case and $\Omega\left(\frac{1}{nK}\right)$[1] for the strongly convex case (Nemirovski & Yudin, 1983; Nesterov et al., 2018; Bubeck et al., 2015).

Though these two lower bounds may be too optimistic for shuffling-based gradient methods, no further progress has been made until (Koren et al., 2022), showing that both RR and SS sampling schemes with a constant stepsize $\eta$ in the general convex case admit the lower bound $\Omega\left(\min\left\{1, \eta\sqrt{\frac{n}{J}} + \eta + \frac{1}{\eta n K}\right\}\right)$ for the suffix average of the last $J$ epochs (i.e., $\frac{1}{Jn}\sum_{j=K-J}^{K-1}\sum_{i=1}^{n} \boldsymbol{x}_{jn+i+1}$). Noticing $\Omega\left(\eta\sqrt{\frac{n}{J}} + \frac{1}{\eta n K}\right) \geq \Omega\left(\frac{1}{J^{1/4}n^{1/4}\sqrt{K}}\right)$ and $\Omega\left(\eta + \frac{1}{\eta n K}\right) \geq \Omega\left(\frac{1}{\sqrt{nK}}\right)$, we hence can simplify the bound into $\Omega\left(\frac{1}{J^{1/4}n^{1/4}\sqrt{K}} + \frac{1}{\sqrt{nK}}\right)$. Especially, this implies the $\Omega\left(\frac{1}{n^{1/4}\sqrt{K}}\right)$ barrier for the suffix average of the last one epoch as listed in Table 1.

However, whether the $\Omega\left(\frac{1}{n^{1/4}\sqrt{K}}\right)$ bound also holds for the last iterate under RR/SS is still unclear. Furthermore, whether the general lower bound $\Omega\left(\frac{1}{nK}\right)$ for the strongly convex case mentioned above is tight for shuffling gradient methods remains unknown as well.

## 2. Preliminary

**Notation.** $\mathbb{N}$ is the set of all positive integers and $[m] \triangleq \{1, \ldots, m\}, \forall m \in \mathbb{N}$. $a \wedge b$ and $a \vee b$ respectively indicate $\min\{a, b\}$ and $\max\{a, b\}$. $X \stackrel{\mathcal{D}}{=} Y$ means that two random variables $X$ and $Y$ have the same probability distribution. $\langle \cdot, \cdot \rangle$ denotes the Euclidean inner product on $\mathbb{R}^d$. $\|\cdot\| \triangleq \sqrt{\langle \cdot, \cdot \rangle}$ is the 2-norm. Given an extended real-valued convex function $h : \mathbb{R}^d \to \overline{\mathbb{R}}$ where $\overline{\mathbb{R}} \triangleq (-\infty, +\infty]$, $\text{dom}h \triangleq \{\boldsymbol{x} \in \mathbb{R}^d : h(\boldsymbol{x}) < +\infty\}$. For any $\boldsymbol{x} \in \text{dom}h$, $\partial h(\boldsymbol{x}) \triangleq \{\boldsymbol{g} \in \mathbb{R}^d : h(\boldsymbol{y}) \geq h(\boldsymbol{x}) + \langle \boldsymbol{g}, \boldsymbol{y} - \boldsymbol{x} \rangle, \forall \boldsymbol{y} \in \mathbb{R}^d\}$ is the set of subgradients at $\boldsymbol{x}$. We denote by $\nabla h(\boldsymbol{x})$ an element in $\partial h(\boldsymbol{x})$ when $\partial h(\boldsymbol{x}) \neq \varnothing$. Throughout the paper, $\mathcal{C}$ always denotes a nonempty closed convex set in $\mathbb{R}^d$ and $I_{\mathcal{C}}$ repre-

---

[1]A subtle point is that this lower bound is established for strongly convex $f$ instead of $\psi$, not strictly fitting our Assumption 2.1. However, by slightly modifying the existing proof (Bubeck et al., 2015) to make it work for the first-order algorithm containing a proximal update step, we can show the same bound still holds for strongly convex $\psi$. See Appendix D for details.

sents its characteristic function, i.e., $I_{\mathcal{C}}(\boldsymbol{x}) = 0$ if $\boldsymbol{x} \in \mathcal{C}$, $+\infty$ otherwise.

In this work, we study the following optimization problem

$$\min_{\boldsymbol{x} \in \mathbb{R}^d} F(\boldsymbol{x}) \triangleq f(\boldsymbol{x}) + \psi(\boldsymbol{x}) \text{ where } f(\boldsymbol{x}) \triangleq \frac{1}{n} \sum_{i=1}^{n} f_i(\boldsymbol{x}).$$

Our analysis relies on two mild assumptions.

**Assumption 2.1.** Each $f_i : \mathbb{R}^d \to \mathbb{R}$ is convex. $\psi : \mathbb{R}^d \to \overline{\mathbb{R}}$ is proper, closed, and convex. Moreover, there exists $\mu \geq 0$ such that $\psi(\boldsymbol{x}) - \psi(\boldsymbol{y}) - \langle \nabla\psi(\boldsymbol{y}), \boldsymbol{x} - \boldsymbol{y} \rangle \geq \frac{\mu}{2} \|\boldsymbol{x} - \boldsymbol{y}\|^2, \forall \boldsymbol{x} \in \mathbb{R}^d, \boldsymbol{y} \in \mathrm{dom}\psi, \nabla\psi(\boldsymbol{y}) \in \partial\psi(\boldsymbol{y})$ whenever $\partial\psi(\boldsymbol{y}) \neq \varnothing$.

Under Assumption 2.1, $\partial f_i(\boldsymbol{x})$ is always nonempty for any $\boldsymbol{x} \in \mathbb{R}^d$ and $i \in [n]$ since $\mathrm{dom} f_i = \mathbb{R}^d$.

**Assumption 2.2.** Each $f_i$ is $G_i$-Lipschitz on $\mathrm{dom}\psi$ for some $G_i > 0$.

We remark that Assumption 2.2 only requires $f_i$ to be Lipschitz on $\mathrm{dom}\psi$ instead of the whole space $\mathbb{R}^d$. Hence, it is also possible to consider the case of strongly convex $f_i$ or $f$ without the domain issue pointed out by (Nguyen et al., 2018). However, to keep it simple, we only focus on the situation of $\psi$ being possibly strongly convex.

## 3. General Proximal Gradient Method

---
**Algorithm 1** General Proximal Gradient Method

---
**Input:** initial point $\boldsymbol{x}_1 \in \mathrm{dom}\psi$, stepsize $\eta_t > 0, \forall t \in [T]$.
**for** $t = 1$ to $T$ **do**
  Generate an index $\mathsf{i}(t) \in [n]$
  $\boldsymbol{x}_{t+1} = \mathrm{argmin}_{\boldsymbol{x} \in \mathbb{R}^d} \psi(\boldsymbol{x}) + \langle \nabla f_{\mathsf{i}(t)}(\boldsymbol{x}_t), \boldsymbol{x} \rangle + \frac{\|\boldsymbol{x} - \boldsymbol{x}_t\|^2}{2\eta_t}$
**Output:** $\boldsymbol{x}_{T+1}$

---

*Remark* 3.1. Algorithm 1 is also known as Incremental Subgradient-Proximal Method (Bertsekas, 2011). However, we use a different name here to distinguish it from the term Incremental Gradient in the literature.

The algorithmic framework studied in the paper, General Proximal Gradient Method (Bertsekas, 2011), is provided in Algorithm 1. We highlight three key differences from the prior proximal shuffling gradient methods (Kibardin, 1979; Mishchenko et al., 2022; Liu & Zhou, 2024b; Josz et al., 2024). First, Algorithm 1 is more general since the generation process of $\mathsf{i}(t)$ is not limited to shuffling-based. Second, Algorithm 1 works for any $T \in \mathbb{N}$, in contrast to $T = Kn$ where $K \in \mathbb{N}$ required in the studies mentioned above. Moreover, the proximal update in Algorithm 1 happens in every step instead of at the end of every epoch (containing $n$ gradient descent steps) in the existing algorithms.

Now we provide some concrete examples of how to generate the index $\mathsf{i}(t)$. The first one is Example 3.2, showing that Proximal SGD is a special case of Algorithm 1.

**Example 3.2.** When $\mathsf{i}(1)$ to $\mathsf{i}(T)$ are mutually independent random variables uniformly distributed on $[n]$, Algorithm 1 recovers the famous Proximal SGD algorithm.

Next, to formally define different shuffling strategies, we require some new notations. Henceforth, $\mathsf{r}(t)$ denotes the modulo operation of $n$, i.e., $\mathsf{r}(t) \triangleq t \mod n$, where we use the convention $Kn \mod n = n, \forall K \in \mathbb{N}$. In addition, we let $\mathsf{q}(t)$ be the smallest integer greater than or equal to $\frac{t}{n}$, i.e., $\mathsf{q}(t) \triangleq \lceil \frac{t}{n} \rceil$ where $\lceil \cdot \rceil$ is the ceiling function. Remarkably, the equation $t = (\mathsf{q}(t) - 1)n + \mathsf{r}(t), \forall t \in \mathbb{N}$ always holds. Lastly, we denote by $S_n$ the symmetric group of $[n]$, i.e., the set containing all permutations of $[n]$.

Equipped with these notations, we introduce the commonly used RR/SS/IG shuffling schemes as follows.

**Example 3.3.** When $\mathsf{i}(t) = \pi_{\mathsf{q}(t)}^{\mathsf{r}(t)}$ where $\pi_1$ to $\pi_{\mathsf{q}(T)}$ are mutually independent random permutations uniformly distributed on $S_n$, it is called the RR sampling scheme.

**Example 3.4.** When $\mathsf{i}(t) = \pi^{\mathsf{r}(t)}$ where $\pi$ is a random permutation uniformly distributed on $S_n$, it is called the SS sampling scheme.

**Example 3.5.** When $\mathsf{i}(t) = \pi^{\mathsf{r}(t)}$ where $\pi$ is a deterministic permutation in $S_n$, it is called the IG sampling scheme.

## 4. Improved Last-Iterate Convergence Rates

In this section, we provide our improved last-iterate convergence rates for Algorithm 1 under both RR and SS sampling schemes. To simplify the notation, we define $G_{f,1} \triangleq \frac{1}{n} \sum_{i=1}^{n} G_i$ and $G_{f,2} \triangleq \sqrt{\frac{1}{n} \sum_{i=1}^{n} G_i^2}$, respectively representing the arithmetic mean and the root mean square of Lipschitz parameters. Notably, the following inequality always holds

$$G_{f,1} \leq G_{f,2} < \sqrt{n} G_{f,1}. \tag{1}$$

Next, to make easy and fair comparisons to the best existing convergence results of shuffling gradient methods for nonsmooth (strongly) convex optimization (Koren et al., 2022; Liu & Zhou, 2024b), we make two extra assumptions here. One is the existence of a point $\boldsymbol{x}_\star \in \mathbb{R}^d$ attaining the minimum value of $F$, i.e., $F(\boldsymbol{x}_\star) = F_\star \triangleq \inf_{\boldsymbol{x} \in \mathbb{R}^d} F(\boldsymbol{x})$. Under this assumption, let $D_\star \triangleq \|\boldsymbol{x}_\star - \boldsymbol{x}_1\|$ denote the distance between the optimal solution and the initial point. The other is assuming the time horizon satisfies $T \geq n$ (or one can simply think $T = Kn$ for $K \in \mathbb{N}$ as in prior works).

*Remark* 4.1. We clarify that the above assumptions are both unnecessary in the full statement of every theorem (except Theorem 4.6). Concretely, for any reference point $\boldsymbol{z} \in \mathbb{R}^d$

and $T \in \mathbb{N}$, the gap $\mathbb{E}\left[F(\boldsymbol{x}_{T+1}) - F(\boldsymbol{z})\right]$ can always be properly upper bounded. See Appendix A for details.

In addition, following the convention in nonsmooth optimization for the general convex case (Nesterov et al., 2018; Lan, 2020), the value of $\eta$ used in the stepsize $\eta_t$ in Theorems 4.2, 4.5, and 4.6 has been optimized to obtain the best dependence on problem-dependent parameters, e.g., $G_{f,1}$, $G_{f,2}$, and $D_\star$. Rates working for arbitrarily picked $\eta$ are deferred to the corresponding full version of each theorem in Appendix A, in which the precise logarithmic factor hidden in the $\widetilde{O}$ notation is also provided.

### 4.1. RR Sampling Scheme

This subsection will focus on the RR sampling scheme. As the reader will see, our new last-iterate bounds are always better than the best-known results in (Liu & Zhou, 2024b).

**Theorem 4.2.** *Under Assumptions 2.1 (with $\mu = 0$) and 2.2, suppose the RR sampling scheme is employed with one of the following three stepsizes $\eta_t, \forall t \in [T]$:*

- $\eta_t = \eta \frac{\mathsf{q}(T) - \mathsf{q}(t) + 1}{\mathsf{q}(T)\sqrt{T}}$ *and* $\eta = \frac{D_\star}{n^{1/4}\sqrt{G_{f,1}G_{f,2}\left(1 + \frac{\log n}{\mathsf{q}(T)}\right)}}$.

- $\eta_t = \frac{\eta}{\sqrt{T}}$ *and* $\eta = \frac{D_\star}{n^{1/4}\sqrt{G_{f,1}G_{f,2}(1 + \log T)}}$.

- $\eta_t = \frac{\eta}{\sqrt{t}}$ *and* $\eta = \frac{D_\star}{n^{1/4}\sqrt{G_{f,1}G_{f,2}}}$.

*Then Algorithm 1 guarantees*

$$\mathbb{E}\left[F(\boldsymbol{x}_{T+1}) - F_\star\right] \leq \widetilde{O}\left(\frac{n^{1/4}\sqrt{G_{f,1}G_{f,2}}D_\star}{\sqrt{T}}\right).$$

*If additionally assuming $T = \Omega(n \log n)$, then the first stepsize choice achieves the following improved rate*

$$\mathbb{E}\left[F(\boldsymbol{x}_{T+1}) - F_\star\right] \leq O\left(\frac{n^{1/4}\sqrt{G_{f,1}G_{f,2}}D_\star}{\sqrt{T}}\right).$$

We start with the general convex case and discuss Theorem 4.2 in detail here. As far as we know, the best and only last-iterate bound that can be applied to the same setting is $O\left(\frac{G_{f,1}D_\star}{\sqrt{K}}\right)$ for $T = Kn$ where $K \in \mathbb{N}$ (Liu & Zhou, 2024b). In that case, Theorem 4.2 achieves the rate $O\left(\frac{\sqrt{G_{f,1}G_{f,2}}D_\star}{n^{1/4}\sqrt{K}}\right)$. Note that by (1), there is

$$\frac{\sqrt{G_{f,1}G_{f,2}}D_\star}{n^{1/4}\sqrt{K}} \Big/ \frac{G_{f,1}D_\star}{\sqrt{K}} = \frac{1}{n^{1/4}}\sqrt{\frac{G_{f,2}}{G_{f,1}}} \in \left[\frac{1}{n^{1/4}}, 1\right).$$

Therefore, our new result is always better than (Liu & Zhou, 2024b) by up to a factor of $\Theta(n^{-1/4})$.

In particular, the $\Theta(n^{-1/4})$ improvement can be achieved when $G_i \equiv G$, leading to the rate $O\left(\frac{GD_\star}{n^{1/4}\sqrt{K}}\right)$. Remarkably, such a rate is as fast as the previously best-known bound established only for the average iterate $\boldsymbol{x}_{Kn+1}^{\mathsf{avg}} \triangleq \frac{1}{Kn}\sum_{t=1}^{Kn} \boldsymbol{x}_{t+1}$ when $\psi = I_{\mathcal{C}}$ (Koren et al., 2022).

An important implication of Theorem 4.2 is to provide the convergence of $\boldsymbol{x}_{T+1}^{\mathsf{suffix}} \triangleq \frac{1}{n}\sum_{t=T-n+1}^{T} \boldsymbol{x}_{t+1}$ as follows.

**Corollary 4.3.** *Under the same setting in Theorem 4.2 (using the third stepsize), Algorithm 1 guarantees*

$$\mathbb{E}\left[F(\boldsymbol{x}_{T+1}^{\mathsf{suffix}}) - F_\star\right] \leq \widetilde{O}\left(\frac{n^{1/4}\sqrt{G_{f,1}G_{f,2}}D_\star}{\sqrt{T}}\right).$$

*Proof.* Due to the convexity of $F$, $\mathbb{E}\left[F(\boldsymbol{x}_{T+1}^{\mathsf{suffix}}) - F_\star\right] \leq \frac{1}{n}\sum_{t=T-n+1}^{T} \mathbb{E}\left[F(\boldsymbol{x}_{t+1}) - F_\star\right]$. We conclude from Theorem 4.2 and the inequality $\frac{1}{n}\sum_{t=T-n+1}^{T} \frac{1}{\sqrt{t}} \leq \frac{2}{\sqrt{T}}$. $\square$

To our best knowledge, Corollary 4.3 is not only the first provable but also the first optimal rate for the suffix average since when $T = Kn$ and $G_i \equiv G$, it matches the lower bound $\Omega\left(\frac{1}{n^{1/4}\sqrt{K}}\right)$ (up to logarithmic factors) shown by (Koren et al., 2022) proved for $\psi = I_{\mathcal{C}}$. However, the careful reader may argue that Corollary 4.3 is not convincing because the original lower bound is established for the constant stepsize $\eta_t \equiv \eta$ and has a stronger version depending on the value of $\eta$ (see Subsection 1.2). In Corollary A.2, we close the gap by giving $\mathbb{E}\left[F(\boldsymbol{x}_{T+1}^{\mathsf{suffix}}) - F_\star\right] \leq \widetilde{O}\left(\frac{D_\star^2}{\eta T} + \eta\sqrt{n}G_{f,1}G_{f,2}\right)$ when $\eta_t \equiv \eta$ and $T \geq 2(n-1)$, which perfectly matches the original lower bound in (Koren et al., 2022) by up to logarithmic factors.

Moreover, we want to talk about the first stepsize choice $\eta_t = \eta\frac{\mathsf{q}(T) - \mathsf{q}(t) + 1}{\mathsf{q}(T)\sqrt{T}}$, which is inspired by (Liu & Zhou, 2024b) who showed that the stepsize schedule proportional to $K - k + 1$ in the $k$-th epoch when $T = Kn$ can remove any extra logarithmic factor in the final rate. Here, we prove this strategy can also be applied to arbitrary $T \in \mathbb{N}$ but will incur an additional $O\left(\sqrt{\frac{\log n}{\mathsf{q}(T)}}\right)$ factor, which is at most in the order of $O(\sqrt{\log n})$ and is automatically shaved off once $\mathsf{q}(T) = \Omega(\log n) \Leftrightarrow T = \Omega(n \log n)$. We emphasize that, though the principle of our first stepsize is highly similar to (Liu & Zhou, 2024b), showing it indeed works for any $T \in \mathbb{N}$ requires a refined analysis, for example, see Lemma C.1 in the appendix.

**Theorem 4.4.** *Under Assumptions 2.1 (with $\mu > 0$) and 2.2, suppose the RR sampling scheme is employed with the stepsize $\eta_t = \frac{2}{\mu t}, \forall t \in [T]$, then Algorithm 1 guarantees*

$$\mathbb{E}\left[F(\boldsymbol{x}_{T+1}) - F_\star\right] \leq \widetilde{O}\left(\frac{\mu D_\star^2}{T^2} + \frac{\sqrt{n}G_{f,1}G_{f,2}}{\mu T}\right).$$

Now we turn our attention to the strongly convex case. As before, we first check the case $T = Kn$, under which Theorem 4.4 gives us the rate $\widetilde{O}\left(\frac{\mu D_\star^2}{n^2 K^2} + \frac{G_{f,1}G_{f,2}}{\mu\sqrt{n}K}\right)$. In comparison, the only last-iterate bound in the same setting is $\widetilde{O}\left(\frac{\mu D_\star^2}{K^2} + \frac{G_{f,1}^2}{\mu K}\right)$ (Liu & Zhou, 2024b). As one can see, the higher order term $\frac{\mu D_\star^2}{n^2 K^2}$ achieves acceleration by a factor of $\Theta(n^{-2})$. Notably, the lower order term $\frac{G_{f,1}G_{f,2}}{\mu\sqrt{n}K}$ is also always faster due to $G_{f,2} < \sqrt{n}G_{f,1}$ in the aforementioned inequality (1). If further assuming $G_i \equiv G$, Theorem 4.4 yields the rate $\widetilde{O}\left(\frac{G^2}{\mu\sqrt{n}K}\right)$ (suppose $K$ is large for simplicity) significantly improving upon the bound $\widetilde{O}\left(\frac{G^2}{\mu K}\right)$ in (Liu & Zhou, 2024b) by a factor of $\Theta(n^{-1/2})$.

In addition, we also extend the convergence result to the stepsize $\eta_t = \frac{m}{\mu t}$ for any $m \in \mathbb{N}$ like (Liu & Zhou, 2024b). The interested reader could refer to Theorem A.3 for details.

Before ending this subsection, we should mention that though both Theorems 4.2 and 4.4 improve upon (Liu & Zhou, 2024b) and indicate that RR enjoys faster convergence than Proximal GD suggesting the benefit of randomness, these results are however still slower than Proximal SGD, whose last iterate is known to converge in $O\left(\frac{G_{f,2}D_\star}{\sqrt{T}}\right)$ and $\widetilde{O}\left(\frac{\mu D_\star^2}{T^2} + \frac{G_{f,2}^2}{\mu T}\right)$ for general and strongly functions, respectively (Harvey et al., 2019; Jain et al., 2021; Orabona, 2020; Liu & Zhou, 2024a). Whether RR can be proved to guarantee the same rates is unclear[2].

## 4.2. SS Sampling Scheme

We study the SS sampling scheme in this subsection. Unlike the always faster rates achieved by RR presented before, our new results for SS are better than (Liu & Zhou, 2024b) only when a certain condition is met.

**Theorem 4.5.** *Under Assumptions 2.1 (with $\mu = 0$) and 2.2, suppose the SS sampling scheme is employed with the stepsize $\eta_t = \frac{\eta}{\sqrt{T}}, \forall t \in [T]$ and $\eta = \frac{D_\star}{\sqrt{\left(\mathsf{q}(T)G_{f,2}^2 \vee \sqrt{n\mathsf{q}(T)}G_{f,1}G_{f,2}\right)(1+\log T)}}$, then Algorithm 1 guarantees*

$$\mathbb{E}\left[F(\boldsymbol{x}_{T+1}) - F_\star\right] \le \widetilde{O}\left(\frac{\sqrt{G_{f,1}G_{f,2}}D_\star}{T^{1/4}} \vee \frac{G_{f,2}D_\star}{\sqrt{n}}\right).$$

We begin with the general convex case in Theorem 4.5. To understand how the above rate compares to the bound $O\left(\frac{G_{f,1}D_\star}{\sqrt{K}}\right)$ when $T = Kn$ where $K \in \mathbb{N}$ (Liu & Zhou, 2024b), we introduce a critical value $K_\star \triangleq \frac{nG_{f,1}^2}{G_{f,2}^2}$, which

falls into $(1, n]$ due to (1). If $K \le K_\star$, we notice the rate in Theorem 4.5 equals $\widetilde{O}\left(\frac{\sqrt{G_{f,1}G_{f,2}}D_\star}{n^{1/4}K^{1/4}}\right)$ and is faster than $O\left(\frac{G_{f,1}D_\star}{\sqrt{K}}\right)$. Otherwise, the rate in Theorem 4.5 equals $\widetilde{O}\left(\frac{G_{f,2}D_\star}{\sqrt{n}}\right)$ and is slower than $O\left(\frac{G_{f,1}D_\star}{\sqrt{K}}\right)$. This observation suggests an interesting convergence phenomenon for the SS sampling scheme, that is the function value gap will decay in the rate no slower than $\widetilde{O}\left(\frac{\sqrt{G_{f,1}G_{f,2}}D_\star}{n^{1/4}K^{1/4}}\right)$ during the first $\Theta(K_\star)$ epochs until reaching the $\widetilde{O}\left(\frac{G_{f,2}D_\star}{\sqrt{n}}\right)$ error regime. We emphasize that this does not necessarily imply the SS strategy must bear constant optimization error since Theorem 4.5 only states a convergence upper bound. In other words, Theorem 4.5 improves upon (Liu & Zhou, 2024b) when $K$ is small. Especially, if $G_i \equiv G$, the rate in Theorem 4.5 and the threshold $K_\star = n$ match the convergence behavior of $\boldsymbol{x}_{Kn+1}^{\mathsf{avg}}$ when $\psi = I_{\mathcal{C}}$ proved in (Koren et al., 2022).

The reader may not feel satisfied with Theorem 4.5 since the rate does not vanish as $T$ becomes larger and could ask whether this is an inherent issue of Algorithm 1 under SS. The answer turns out to be negative. A simple but important observation is that when $\psi = 0$ and $T = Kn$, Algorithm 1 with shuffling-based $i(t)$ and the algorithm studied in (Liu & Zhou, 2024b) are actually the same method. We recall a key fact that the rate $O\left(\frac{G_{f,1}D_\star}{\sqrt{K}}\right)$ in (Liu & Zhou, 2024b) is proved for any shuffling type as mentioned in Subsection 1.2 (or see Table 1). Hence, at least in the case of $\psi = 0$ and $T = Kn$, we should expect a refined upper bound that converges to 0 when $K$ approaches infinity.

Built upon this thought, we prove the following sharper rate when $\psi = I_{\mathcal{C}}$ (i.e., constrained optimization) and $T = Kn$.

**Theorem 4.6.** *Under Assumptions 2.1 (with $\psi = I_{\mathcal{C}}$) and 2.2, suppose $T = Kn$ where $K \in \mathbb{N}$ and the SS sampling scheme is employed with the stepsize $\eta_t = \frac{\eta}{\sqrt{T}}, \forall t \in [T]$ and $\eta = \frac{D_\star}{\sqrt{\left(\sqrt{nK}G_{f,1}G_{f,2} \wedge nG_{f,1}^2\right)(1+\log nK)}}$, then Algorithm 1 guarantees*

$$\mathbb{E}\left[F(\boldsymbol{x}_{Kn+1}) - F_\star\right] \le \widetilde{O}\left(\frac{\sqrt{G_{f,1}G_{f,2}}D_\star}{n^{1/4}K^{1/4}} \wedge \frac{G_{f,1}D_\star}{\sqrt{K}}\right).$$

Remarkably, the above result integrates the advantages of Theorem 4.5 and (Liu & Zhou, 2024b) and is thereby faster than both. One point we remind the reader is that the existing analysis in (Liu & Zhou, 2024b) immediately invalids once $\mathcal{C} \ne \mathbb{R}^d$ (equivalently $\psi \ne 0$) since in that case Algorithm 1 is no longer the same as the method in (Liu & Zhou, 2024b) due to different places for the proximal update as discussed in Section 3. Therefore, a more careful analysis is required to overcome this issue. We also mention that the condition

---

[2]We incline to a negative answer at least for the general convex case. See our discussion after Corollary A.2 for why.

$\psi = I_{\mathcal{C}}$ can be further relaxed to $\psi = \varphi + I_{\mathcal{C}}$ for $\varphi$ being Lipschitz on $\mathcal{C}$ (see Lemma B.3 for details) as previously assumed in (Bertsekas, 2011). However, how to extend Theorem 4.6 to any general $\psi$ and $T \in \mathbb{N}$ is unknown to us currently, which is left as an important future work.

**Theorem 4.7.** *Under Assumptions 2.1 (with $\mu > 0$) and 2.2, suppose the SS sampling scheme is employed with the stepsize $\eta_t = \frac{2}{\mu t}, \forall t \in [T]$, then Algorithm 1 guarantees*

$$\mathbb{E}\left[F(\boldsymbol{x}_{T+1}) - F_\star\right] \leq \widetilde{O}\left(\frac{\mu D_\star^2}{T^2} + \frac{G_{f,1} G_{f,2}}{\mu \sqrt{T}} + \frac{G_{f,2}^2}{\mu n}\right).$$

Lastly, we move to the strongly convex case. When $T = Kn$ where $K \in \mathbb{N}$, one can check the critical time to balance the last two terms is still $K_\star = \frac{n G_{f,1}^2}{G_{f,2}^2} \in (1, n]$. Consequently, compared to $\widetilde{O}\left(\frac{\mu D_\star^2}{K^2} + \frac{G_{f,1}^2}{\mu K}\right)$ (Liu & Zhou, 2024b), Theorem 4.7 outperforms in the regime $K \leq K_\star$.

One may expect that when $T$ is a multiple of $n$ and $\psi$ satisfies certain conditions, we can again obtain an improved rate better than both Theorem 4.7 and (Liu & Zhou, 2024b). However, even under some additional assumptions, the best improvement upon Theorem 4.7 we can obtain currently is the rate $\widetilde{O}\left(\frac{n G_{f,1}^2}{\mu K}\right)^3$ (suppose $K$ is large enough). Unfortunately, this is worse than $\widetilde{O}\left(\frac{G_{f,1}^2}{\mu K}\right)$ in (Liu & Zhou, 2024b). Therefore, we simply leave Theorem 4.7 in its current form without providing any further refined result and hope that a bound faster than both Theorem 4.7 and (Liu & Zhou, 2024b) could be found in the future.

Like the strongly convex case for RR, the stepsize in Theorem 4.7 can also be generalized to $\eta_t = \frac{m}{\mu t}$ for any $m \in \mathbb{N}$. For details, see Theorem A.6.

To summarize, this subsection provides a new picture for the last-iterate convergence of the SS sampling scheme. Precisely, we show SS indeed boosts the convergence leading to a faster rate than Proximal GD at least in the early optimization phase, which partially improves upon (Liu & Zhou, 2024b) and reflects the benefit of randomness. In the special case of constrained general convex optimization, we give an even sharper bound demonstrating that SS provably beats Proximal GD no matter how long the time horizon is. In contrast, such information is missed in the best previous bounds (Liu & Zhou, 2024b). However, the same as the RR strategy, there is still a gap between SS and Proximal SGD. Understanding this difference is an important future work.

---

[3]The reader who wonders why we cannot do better could refer to the discussion of Lemma B.3.

## 5. Key Ideas and Proof Sketch

This section describes our key ideas in the analysis, provides the most important Lemma 5.1, and then uses it to sketch the proof. Any omitted detail can be found in the appendix.

### 5.1. The Existing Analysis Misses Randomness

Before talking about our analysis, we first provide an intuitive explanation for why (Liu & Zhou, 2024b) can only achieve the same rate as Proximal GD. Roughly speaking, this is because (Liu & Zhou, 2024b) analyzes the shuffling gradient methods in a way similar to Proximal GD. More technically, their analysis views one whole epoch as a single update step (this also explains why their proximal step happens at the end of every epoch) and then measures how close the progress made in every epoch is to Proximal GD (see their Lemmas D.1 and D.3). Afterwards, they follow the way developed in (Zamani & Glineur, 2023; Liu & Zhou, 2024a) to prove the last-iterate convergence rate.

However, one key point missed in (Liu & Zhou, 2024b) is the randomness. Precisely, their proof for the nonsmooth case goes through whenever the index $\mathsf{i}(t)$ satisfies $\{\mathsf{i}((k-1)n+1), \cdots, \mathsf{i}(kn))\} = [n], \forall k \in [K]$ if $T = Kn$ where $K \in \mathbb{N}$ (this also explains why their rates work for any shuffling scheme not limited to RR/SS/IG). As such, results in (Liu & Zhou, 2024b) cannot reflect any potential benefit for randomly generated $\mathsf{i}(t)$.

### 5.2. Our Analysis Considers Randomness

Due to the above discussion, a natural thought arises, identifying the role of randomness for RR/SS and trying to integrate it into the analysis.

Let us first understand what randomness RR and SS have. A simple but important observation is that for both of them, regardless of the dependence between indices, every $\mathsf{i}(t)$ has the same distribution as the random variable uniformly distributed on $[n]$, i.e., $\mathsf{i}(t) \overset{\mathcal{D}}{=} \mathrm{Uniform}\,[n], \forall t \in [T]$. Hence, we can make an abstraction here, i.e., consider any possible random process $\mathsf{i}(t)$ satisfying $\mathsf{i}(t) \overset{\mathcal{D}}{=} \mathrm{Uniform}\,[n], \forall t \in [T]$ instead of limited to RR/SS.

Next, we think about how to inject this abstraction into a concrete analysis. A potential template we can try to follow is the last-iterate analysis for the standard Proximal SGD method (Liu & Zhou, 2024a), where indices $\mathsf{i}(1)$ to $\mathsf{i}(T)$ are exactly mutually independent random variables uniformly distributed on $[n]$, which can be recognized as the extreme case. Moreover, note that Proximal SGD converges for any $T \in \mathbb{N}$, it is hence a perfect candidate to help us remove the requirement $T = Kn$ for $K \in \mathbb{N}$ used in prior works. Unfortunately, proofs in (Liu & Zhou, 2024a) cannot work directly due to the potential dependence between every $\mathsf{i}(t)$

in the general case. Thus, our analysis contains careful changes and naturally diverges from (Liu & Zhou, 2024a).

In brief, our plan is to analyze the last-iterate convergence of Algorithm 1 (for general indices satisfying $\mathrm{i}(t) \overset{\mathcal{D}}{=}$ Uniform $[n], \forall t \in [T]$) in a manner inspired by Proximal SGD instead of Proximal GD. We clarify that though some analyses motivated by SGD have also appeared in (Nagaraj et al., 2019; Koren et al., 2022) before, there are however many differences. The first and most important one is that they only focus on shuffling-based methods without further abstraction on $\mathrm{i}(t)$. Moreover, their goal is to establish convergence for the average iterate in contrast to the harder last iterate. In addition, both of them only take $\psi = I_{\mathcal{C}}$ (i.e., constrained optimization) into account instead of general $\psi$ studied by us. Even more, (Nagaraj et al., 2019) additionally requires smoothness on $f_i$, (Koren et al., 2022) only has results for the general convex case, and both of them assume $G_i \equiv G$. As the reader will see, our core Lemma 5.1 holds for general $\mathrm{i}(t)$ and gives a novel sufficient condition for the last-iterate convergence working any $\psi$ under minimal conditions, Assumptions 2.1 and 2.2. Hence, although the high-level idea may sound similar, our analysis significantly differs from theirs.

### 5.3. A New Sufficient Lemma for the Last-Iterate Rate

Due to limited space, we will not provide formal analysis but only state the core result, Lemma 5.1, a new sufficient lemma giving the last-iterate convergence rate.

**Lemma 5.1.** *Under Assumptions 2.1 and 2.2, suppose the following three conditions hold:*

1. *The index satisfies* $\mathrm{i}(t) \overset{\mathcal{D}}{=}$ Uniform $[n], \forall t \in [T]$.

2. *The stepsize* $\eta_t, \forall t \in [T]$ *is non-increasing.*

3. $|\mathbb{E}[\Omega_t(\boldsymbol{x}_s)]| \leq \Phi \eta_s, \forall t \in [T], s \in [t]$ *where* $\Omega_t(\cdot) \triangleq f_{\mathrm{i}(t)}(\cdot) - f(\cdot), \forall t \in [T]$ *and* $\Phi \geq 0$ *is a constant probably depending on* $T$, $n$, $\mu$, *and* $G_1$ *to* $G_n$.

*Then Algorithm 1 guarantees*

$\mathbb{E}[F(\boldsymbol{x}_{T+1}) - F_\star]$

$$\leq O\left(\frac{D_\star^2}{\sum_{t=1}^T \gamma_t \eta_t} + \left(G_{f,2}^2 + \Phi\right)\sum_{t=1}^T \frac{\gamma_t \eta_t^2}{\sum_{s=t}^T \gamma_s \eta_s}\right),$$

*where* $\gamma_t \triangleq \prod_{s=1}^{t-1}(1 + \mu\eta_s), \forall t \in [T+1]$.

*Remark* 5.2. We again assume the existence of an optimum $\boldsymbol{x}_\star \in \mathbb{R}^d$ and use the notation $D_\star = \|\boldsymbol{x}_\star - \boldsymbol{x}_1\|$. See Lemma B.2 for the full version valid for any $\boldsymbol{z} \in \mathbb{R}^d$.

We first talk about the three conditions in Lemma 5.1. Condition 1 is from the abstraction of the randomness in RR/SS

mentioned before. Condition 2 is satisfied by almost all common stepsizes. However, we remark that it is actually not necessary and only for simplicity. Even without it, the rate still holds with the only change $\eta_t^2$ to $\eta_t(\eta_t \vee \eta_{t+1})$. Condition 3 is the most important. Intuitively, it says that even $\mathrm{i}(t)$ and $\boldsymbol{x}_s$ are possibly dependent leading to $\mathbb{E}[\Omega_t(\boldsymbol{x}_s)] \neq 0$ (equivalently, $\mathbb{E}[f_{\mathrm{i}(t)}(\boldsymbol{x}_s)] \neq \mathbb{E}[f(\boldsymbol{x}_s)]$), Algorithm 1 still guarantees the last-iterate convergence once $|\mathbb{E}[\Omega_t(\boldsymbol{x}_s)]|$ can be controlled by the value of $\eta_s$ with another multiplicative factor $\Phi \geq 0$. Notably, the smaller $\Phi$ is, the better the convergence rate is. As a sanity check, we consider Proximal SGD and notice that $\Phi = 0$ in this case. Then Lemma 5.1 recovers the existing bound in (Liu & Zhou, 2024a).

Armed with Lemma 5.1, to prove a last-iterate convergence rate for Algorithm 1 employing any general random index $\mathrm{i}(t)$, it is sufficient to find the corresponding constant $\Phi$. Here we claim that under the same setting of every theorem in Section 4, there is always

$$\Phi = \begin{cases} \Theta\left(\sqrt{n}G_{f,1}G_{f,2}\right) & \text{for RR} \\ \Theta\left(\sqrt{T}G_{f,1}G_{f,2} + \frac{T}{n}G_{f,2}^2\right) & \text{for SS} \end{cases}.$$

With these two values of $\Phi$ and the inequality (1), i.e., $G_{f,1} \leq G_{f,2} < \sqrt{n}G_{f,1}$, we immediately conclude Theorems 4.2, 4.4, 4.5, and 4.7 after plugging in the stepsize. However, we remark that finding these two values is not a trivial task for RR/SS. Though some clues on how to bound $|\mathbb{E}[\Omega_t(\boldsymbol{x}_t)]|$ can be found in prior works (Nagaraj et al., 2019; Koren et al., 2022), our goal is more general and hence harder since we need to bound $|\mathbb{E}[\Omega_t(\boldsymbol{x}_s)]|$ by a time-dependent stepsize $\eta_s$ for any $s \leq t$. More importantly, our fine-grained analysis on $G_{f,1}$ and $G_{f,2}$ is the key to establishing the superiority of our convergence results over (Liu & Zhou, 2024b), which cannot be found in (Nagaraj et al., 2019; Koren et al., 2022) due to their simpler settings as mentioned earlier.

Lastly, we briefly discuss Theorem 4.6. Due to the dependence on $T$ in $\Phi$ for SS, the above analysis is inadequate. However, as hinted by (Liu & Zhou, 2024b) (though algorithms are different), one should expect that SS at least provably converges as fast as Proximal GD when $T = Kn$ for $K \in \mathbb{N}$. As such, we will combine some other techniques to obtain Theorem 4.6 and not discuss them here.

## 6. Conclusion and Future Work

We provide improved last-iterate convergence rates for nonsmooth (strongly) convex optimization under both RR and SS sampling schemes, showing they are faster than Proximal GD (conditionally for SS) and enjoy the benefit of randomness. Two valued problems are left to be addressed in the future. One is to explore whether our rates for RR are tight. The other is to prove better bounds for SS if possible.

## Acknowledgments

This work is supported by the NSF grant ECCS-2419564. We also thank the anonymous reviewers for their valuable comments.

## Impact Statement

This paper presents work whose goal is to advance the field of Machine Learning. There are many potential societal consequences of our work, none which we feel must be specifically highlighted here.

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

# A. Full Theorems

In this section, we provide the full version of each theorem presented in Section 4 and the corresponding proof. The proofs of lemmas used in the analysis are deferred to Sections B and C later. One point here we want to remind the reader is that our new last-iterate results (except Theorem A.5) hold for any $T \in \mathbb{N}$ instead of only $T = Kn$ for $K \in \mathbb{N}$ used in most of the existing works for shuffling gradient methods mentioned in Subsection 1.2.

## A.1. RR Sampling Scheme

We first consider the general convex case in Theorem A.1. One notable thing is that, under the first stepsize choice, the extra logarithmic factor can be shaved off once $\mathsf{q}(T) = \Omega(\log n) \Leftrightarrow T = \Omega(n \log n)$ as previously mentioned in Theorem 4.2. Finding a stepsize that can remove any extra logarithmic factor without requiring $T = \Omega(n \log n)$ will be an interesting task.

**Theorem A.1.** *(Full version of Theorem 4.2) Under Assumptions 2.1 (with $\mu = 0$) and 2.2, suppose the RR sampling scheme is employed:*

- *Taking the stepsize $\eta_t = \eta \frac{\mathsf{q}(T) - \mathsf{q}(t) + 1}{\mathsf{q}(T)\sqrt{T}}, \forall t \in [T]$, then for any $z \in \mathbb{R}^d$, Algorithm 1 guarantees*

$$\mathbb{E}\left[F(\boldsymbol{x}_{T+1}) - F(\boldsymbol{z})\right] \leq O\left(\left(\frac{\|\boldsymbol{z} - \boldsymbol{x}_1\|^2}{\eta} + \eta\sqrt{n}G_{f,1}G_{f,2}\left(1 + \frac{\log n}{\mathsf{q}(T)}\right)\right)\frac{1}{\sqrt{T}}\right).$$

  *Setting $\eta = \frac{\|\boldsymbol{z} - \boldsymbol{x}_1\|}{n^{1/4}\sqrt{G_{f,1}G_{f,2}\left(1 + \frac{\log n}{\mathsf{q}(T)}\right)}}$ to optimize the dependence on parameters.*

- *Taking the stepsize $\eta_t = \frac{\eta}{\sqrt{T}}, \forall t \in [T]$, then for any $z \in \mathbb{R}^d$, Algorithm 1 guarantees*

$$\mathbb{E}\left[F(\boldsymbol{x}_{T+1}) - F(\boldsymbol{z})\right] \leq O\left(\left(\frac{\|\boldsymbol{z} - \boldsymbol{x}_1\|^2}{\eta} + \eta\sqrt{n}G_{f,1}G_{f,2}(1 + \log T)\right)\frac{1}{\sqrt{T}}\right).$$

  *Setting $\eta = \frac{\|\boldsymbol{z} - \boldsymbol{x}_1\|}{n^{1/4}\sqrt{G_{f,1}G_{f,2}(1 + \log T)}}$ to optimize the dependence on parameters.*

- *Taking the stepsize $\eta_t = \frac{\eta}{\sqrt{t}}, \forall t \in [T]$, then for any $z \in \mathbb{R}^d$, Algorithm 1 guarantees*

$$\mathbb{E}\left[F(\boldsymbol{x}_{T+1}) - F(\boldsymbol{z})\right] \leq O\left(\left(\frac{\|\boldsymbol{z} - \boldsymbol{x}_1\|^2}{\eta} + \eta\sqrt{n}G_{f,1}G_{f,2}(1 + \log T)\right)\frac{1}{\sqrt{T}}\right).$$

  *Setting $\eta = \frac{\|\boldsymbol{z} - \boldsymbol{x}_1\|}{n^{1/4}\sqrt{G_{f,1}G_{f,2}}}$ to optimize the dependence on parameters.*

*Proof.* Note that the RR sampling scheme satisfies $\mathsf{i}(t) \overset{\mathcal{D}}{=} \text{Uniform}[n], \forall t \in [T]$, and all these stepsizes listed are non-increasing. Hence, Conditions 1 and 2 in Lemma B.2 are fulfilled. If Condition 3 also holds, i.e., $|\mathbb{E}[\Omega_t(\boldsymbol{x}_s)]| \leq \Phi\eta_s, \forall t \in [T], s \in [t]$ for some $\Phi \geq 0$, we can invoke Lemma B.2 to obtain for any $\boldsymbol{z} \in \mathbb{R}^d$,

$$\mathbb{E}\left[F(\boldsymbol{x}_{T+1}) - F(\boldsymbol{z})\right] \leq O\left(\frac{\|\boldsymbol{z} - \boldsymbol{x}_1\|^2}{\sum_{t=1}^T \gamma_t \eta_t} + (G_{f,2}^2 + \Phi)\sum_{t=1}^T \frac{\gamma_t \eta_t^2}{\sum_{s=t}^T \gamma_s \eta_s}\right)$$

$$= O\left(\frac{\|\boldsymbol{z} - \boldsymbol{x}_1\|^2}{\sum_{t=1}^T \eta_t} + (G_{f,2}^2 + \Phi)\sum_{t=1}^T \frac{\eta_t^2}{\sum_{s=t}^T \eta_s}\right), \tag{2}$$

where the second line is by $\gamma_t = \prod_{s=1}^{t-1}(1 + \mu\eta_s) = 1, \forall t \in [T+1]$ since $\mu = 0$ now.

Our next task is to find a constant $\Phi \geq 0$ satisfying

$$|\mathbb{E}[\Omega_t(\boldsymbol{x}_s)]| \leq \Phi\eta_s, \forall t \in [T], s \in [t].$$

For all stepsizes listed, we claim

$$\Phi = 4\left(G_{f,2}^2 + 2\sqrt{n}G_{f,1}G_{f,2}\right). \tag{3}$$

To prove our statement, we apply Lemma B.4 with $\mu = 0$ to have:

- If $s \in [(\mathsf{q}(t)-1)n]$, there is
$$\mathbb{E}\left[\Omega_t(\boldsymbol{x}_s)\right] = 0 \Rightarrow |\mathbb{E}\left[\Omega_t(\boldsymbol{x}_s)\right]| = 0 \leq \Phi\eta_s.$$

- If $s \in [t] \setminus [(\mathsf{q}(t)-1)n]$ (which implies $\mathsf{q}(s) = \mathsf{q}(t)$), there is

$$
\begin{aligned}
|\mathbb{E}\left[\Omega_t(\boldsymbol{x}_s)\right]| &\leq \frac{\sqrt{2}G_{f,2}^2}{n} \sum_{j=(\mathsf{q}(t)-1)n+1}^{s-1} \eta_j + \frac{2\sqrt{2}G_{f,1}G_{f,2}}{n} \sum_{i=(\mathsf{q}(t)-1)n+1}^{s-1} \sqrt{\sum_{j=i}^{s-1} \eta_j^2}. \\
&\overset{(a)}{\leq} \frac{\sqrt{2}G_{f,2}^2}{n} \sum_{j=(\mathsf{q}(t)-1)n+1}^{s-1} \eta_j + \frac{2\sqrt{2}G_{f,1}G_{f,2}}{n} \sum_{i=(\mathsf{q}(t)-1)n+1}^{s-1} \sqrt{s-i}\,\eta_i \\
&\overset{(b)}{\leq} \frac{\sqrt{2}\left(G_{f,2}^2 + 2\sqrt{n}G_{f,1}G_{f,2}\right)}{n} \sum_{j=(\mathsf{q}(t)-1)n+1}^{s-1} \eta_j,
\end{aligned}
\tag{4}
$$

where $(a)$ is because $\eta_j$ is non-increasing and $(b)$ is due to $\sqrt{s-i} \leq \sqrt{s-(\mathsf{q}(t)-1)n-1} = \sqrt{\mathsf{r}(s)-1} \leq \sqrt{n}$ when $i \in \{(\mathsf{q}(t)-1)n+1, \cdots, s-1\}$. Note that if $s = (\mathsf{q}(t)-1)n+1$, $|\mathbb{E}\left[\Omega_t(\boldsymbol{x}_s)\right]| = 0 \leq \Phi\eta_s$. Hence, we assume $(\mathsf{q}(t)-1)n+2 \leq s \leq t$ in the following.

  - For the stepsize satisfying $\eta_t = \eta_{(\mathsf{q}(t)-1)n+1}, \forall t \in [T]$ (the first two), we observe that $(\mathsf{q}(t)-1)n+1 \leq j \leq s-1 \Rightarrow \eta_j = \eta_{(\mathsf{q}(t)-1)n+1} = \eta_s$. Therefore,
$$\sum_{j=(\mathsf{q}(t)-1)n+1}^{s-1} \eta_j = (\mathsf{r}(s)-1)\eta_s \leq n\eta_s. \tag{5}$$

  Thus, there is
$$|\mathbb{E}\left[\Omega_t(\boldsymbol{x}_s)\right]| \overset{(4),(5)}{\leq} \sqrt{2}\left(G_{f,2}^2 + 2\sqrt{n}G_{f,1}G_{f,2}\right)\eta_s \overset{(3)}{\leq} \Phi\eta_s.$$

  - For the stepsize $\eta_t = \frac{\eta}{\sqrt{t}}, \forall t \in [T]$, we know

$$
\begin{aligned}
\sum_{j=(\mathsf{q}(t)-1)n+1}^{s-1} \eta_j &= \eta \sum_{j=(\mathsf{q}(t)-1)n+1}^{s-1} \frac{1}{\sqrt{j}} \leq \eta \int_{(\mathsf{q}(t)-1)n}^{s-1} \frac{1}{\sqrt{j}}\mathrm{d}j = 2\eta(\sqrt{s-1} - \sqrt{(\mathsf{q}(t)-1)n}) \\
&= \frac{2\eta(\mathsf{r}(s)-1)}{\sqrt{s-1} + \sqrt{(\mathsf{q}(t)-1)n}} = \frac{2(\mathsf{r}(s)-1)\sqrt{s}}{\sqrt{s-1} + \sqrt{(\mathsf{q}(t)-1)n}}\eta_s \leq 2\sqrt{2}n\eta_s,
\end{aligned}
\tag{6}
$$

  where the last step is by $\frac{2(\mathsf{r}(s)-1)\sqrt{s}}{\sqrt{s-1}+\sqrt{(\mathsf{q}(t)-1)n}} \leq 2\sqrt{\frac{s}{s-1}}n \leq 2\sqrt{2}n$ when $s \geq (\mathsf{q}(t)-1)n+2 \geq 2$. Thus, there is
$$|\mathbb{E}\left[\Omega_t(\boldsymbol{x}_s)\right]| \overset{(4),(6)}{\leq} 4\left(G_{f,2}^2 + 2\sqrt{n}G_{f,1}G_{f,2}\right)\eta_s \overset{(3)}{=} \Phi\eta_s.$$

By (2) and (3), we have for all these stepsizes

$$
\begin{aligned}
\mathbb{E}\left[F(\boldsymbol{x}_{T+1}) - F(\boldsymbol{z})\right] &\leq O\left(\frac{\|\boldsymbol{z} - \boldsymbol{x}_1\|^2}{\sum_{t=1}^{T} \eta_t} + \left(G_{f,2}^2 + \sqrt{n}G_{f,1}G_{f,2}\right) \sum_{t=1}^{T} \frac{\eta_t^2}{\sum_{s=t}^{T} \eta_s}\right) \\
&\leq O\left(\frac{\|\boldsymbol{z} - \boldsymbol{x}_1\|^2}{\sum_{t=1}^{T} \eta_t} + \sqrt{n}G_{f,1}G_{f,2} \sum_{t=1}^{T} \frac{\eta_t^2}{\sum_{s=t}^{T} \eta_s}\right),
\end{aligned}
$$

where the last step is by noticing $G_{f,2} < \sqrt{n}G_{f,1}$.

- For the stepsize $\eta_t = \eta\frac{\mathsf{q}(T)-\mathsf{q}(t)+1}{\mathsf{q}(T)\sqrt{T}}, \forall t \in [T]$, by applying Lemma C.1 with $\eta_\star = \frac{\eta}{\mathsf{q}(T)\sqrt{T}}$, we know $\sum_{t=1}^{T} \eta_t \geq \frac{\eta\sqrt{T}}{2}$ and $\sum_{t=1}^{T} \frac{\eta_t^2}{\sum_{s=t}^{T} \eta_s} \leq \frac{9\eta}{2\sqrt{T}}\left(1 + \frac{\log n}{\mathsf{q}(T)}\right)$, which implies

$$\mathbb{E}\left[F(\boldsymbol{x}_{T+1}) - F(\boldsymbol{z})\right] \leq O\left(\left(\frac{\|\boldsymbol{z} - \boldsymbol{x}_1\|^2}{\eta} + \eta\sqrt{n}G_{f,1}G_{f,2}\left(1 + \frac{\log n}{\mathsf{q}(T)}\right)\right)\frac{1}{\sqrt{T}}\right).$$

Setting $\eta = \frac{\|\boldsymbol{z} - \boldsymbol{x}_1\|}{n^{1/4}\sqrt{G_{f,1}G_{f,2}\left(1 + \frac{\log n}{\mathsf{q}(T)}\right)}}$ to obtain

$$\mathbb{E}\left[F(\boldsymbol{x}_{T+1}) - F(\boldsymbol{z})\right] \leq O\left(\frac{n^{1/4}\sqrt{G_{f,1}G_{f,2}}\,\|\boldsymbol{z} - \boldsymbol{x}_1\|\,\sqrt{1 + \frac{\log n}{\mathsf{q}(T)}}}{\sqrt{T}}\right).$$

Particularly, if $\mathsf{q}(T) = \Omega(\log n) \Leftrightarrow T = \Omega(n \log n)$, there is

$$\mathbb{E}\left[F(\boldsymbol{x}_{T+1}) - F(\boldsymbol{z})\right] \leq O\left(\frac{n^{1/4}\sqrt{G_{f,1}G_{f,2}}\,\|\boldsymbol{z} - \boldsymbol{x}_1\|}{\sqrt{T}}\right).$$

- For the stepsize $\eta_t = \frac{\eta}{\sqrt{T}}, \forall t \in [T]$, we know

$$\sum_{t=1}^{T}\frac{\eta_t^2}{\sum_{s=t}^{T}\eta_s} \leq \frac{\eta}{\sqrt{T}}\sum_{t=1}^{T}\frac{1}{T - t + 1} \leq \frac{\eta(1 + \log T)}{\sqrt{T}}.$$

Hence, we have

$$\mathbb{E}\left[F(\boldsymbol{x}_{T+1}) - F(\boldsymbol{z})\right] \leq O\left(\left(\frac{\|\boldsymbol{z} - \boldsymbol{x}_1\|^2}{\eta} + \eta\sqrt{n}G_{f,1}G_{f,2}(1 + \log T)\right)\frac{1}{\sqrt{T}}\right).$$

Setting $\eta = \frac{\|\boldsymbol{z} - \boldsymbol{x}_1\|}{n^{1/4}\sqrt{G_{f,1}G_{f,2}(1 + \log T)}}$ to obtain

$$\mathbb{E}\left[F(\boldsymbol{x}_{T+1}) - F(\boldsymbol{z})\right] \leq O\left(\frac{n^{1/4}\sqrt{G_{f,1}G_{f,2}}\,\|\boldsymbol{z} - \boldsymbol{x}_1\|\,\sqrt{1 + \log T}}{\sqrt{T}}\right).$$

- For the stepsize $\eta_t = \frac{\eta}{\sqrt{t}}, \forall t \in [T]$, we know for any $t \in [T]$,

$$\sum_{s=t}^{T}\eta_s = \eta\sum_{s=t}^{T}\frac{1}{\sqrt{s}} \geq \eta\int_{t}^{T+1}\frac{1}{\sqrt{s}}\mathrm{d}s = 2\eta(\sqrt{T+1} - \sqrt{t}),$$

which implies

$$\sum_{t=1}^{T}\frac{\eta_t^2}{\sum_{s=t}^{T}\eta_s} \leq \eta\sum_{t=1}^{T}\frac{1}{t(\sqrt{T+1} - \sqrt{t})} = \eta\sum_{t=1}^{T}\frac{\sqrt{T+1} + \sqrt{t}}{t(T+1-t)} \leq 2\eta\sum_{t=1}^{T}\frac{\sqrt{T+1}}{t(T+1-t)}$$

$$= \frac{2\eta}{\sqrt{T+1}}\sum_{t=1}^{T}\frac{1}{t} + \frac{1}{T+1-t} \leq \frac{4\eta(1 + \log T)}{\sqrt{T+1}}.$$

Hence, we have

$$\mathbb{E}\left[F(\boldsymbol{x}_{T+1}) - F(\boldsymbol{z})\right] \leq O\left(\left(\frac{\|\boldsymbol{z} - \boldsymbol{x}_1\|^2}{\eta} + \eta\sqrt{n}G_{f,1}G_{f,2}(1 + \log T)\right)\frac{1}{\sqrt{T}}\right).$$

Setting $\eta = \frac{\|\boldsymbol{z} - \boldsymbol{x}_1\|}{n^{1/4}\sqrt{G_{f,1}G_{f,2}}}$ to obtain

$$\mathbb{E}\left[F(\boldsymbol{x}_{T+1}) - F(\boldsymbol{z})\right] \leq O\left(\frac{n^{1/4}\sqrt{G_{f,1}G_{f,2}}\,\|\boldsymbol{z} - \boldsymbol{x}_1\|\,(1 + \log T)}{\sqrt{T}}\right).$$

$\square$

Built upon Theorem A.1, we give the following rate for $\boldsymbol{x}_{T+1}^{\mathsf{suffix}} = \frac{1}{n}\sum_{t=T-n+1}^{T} \boldsymbol{x}_{t+1}$.

**Corollary A.2.** *(Full version of Corollary 4.3) Under Assumptions 2.1 (with $\mu = 0$) and 2.2, suppose $T \geq n$ and the* RR *sampling scheme is employed:*

- *Taking the stepsize $\eta_t = \eta, \forall t \in [T]$, then for any $\boldsymbol{z} \in \mathbb{R}^d$, Algorithm 1 guarantees (additionally assuming $T \geq 2(n-1)^4$)*

$$\mathbb{E}\left[F(\boldsymbol{x}_{T+1}^{\mathsf{suffix}}) - F(\boldsymbol{z})\right] \leq O\left(\frac{\|\boldsymbol{z} - \boldsymbol{x}_1\|^2}{\eta T} + \eta\sqrt{n}G_{f,1}G_{f,2}(1 + \log T)\right).$$

- *Taking the stepsize $\eta_t = \frac{\eta}{\sqrt{t}}, \forall t \in [T]$, then for any $\boldsymbol{z} \in \mathbb{R}^d$, Algorithm 1 guarantees*

$$\mathbb{E}\left[F(\boldsymbol{x}_{T+1}^{\mathsf{suffix}}) - F(\boldsymbol{z})\right] \leq O\left(\left(\frac{\|\boldsymbol{z} - \boldsymbol{x}_1\|^2}{\eta} + \eta\sqrt{n}G_{f,1}G_{f,2}(1 + \log T)\right)\frac{1}{\sqrt{T}}\right).$$

*Proof.* Both results are directly implied by Theorem A.1.

- For the stepsize $\eta_t = \eta, \forall t \in [T]$, we invoke the second result in Theorem A.1 (under changing $\eta$ to $\eta\sqrt{T}$ in it) to obtain

$$\mathbb{E}\left[F(\boldsymbol{x}_{T+1}) - F(\boldsymbol{z})\right] \leq O\left(\frac{\|\boldsymbol{z} - \boldsymbol{x}_1\|^2}{\eta T} + \eta\sqrt{n}G_{f,1}G_{f,2}(1 + \log T)\right).$$

Note that the above result actually holds for any $T \in \mathbb{N}$. Hence, there is by convexity

$$\mathbb{E}\left[F(\boldsymbol{x}_{T+1}^{\mathsf{suffix}}) - F(\boldsymbol{z})\right] \leq \frac{1}{n}\sum_{t=T-n+1}^{T} \mathbb{E}\left[F(\boldsymbol{x}_{t+1}) - F(\boldsymbol{z})\right] \leq \frac{1}{n}\sum_{t=T-n+1}^{T} O\left(\frac{\|\boldsymbol{z} - \boldsymbol{x}_1\|^2}{\eta t} + \eta\sqrt{n}G_{f,1}G_{f,2}(1 + \log t)\right).$$

Moreover, we have

$$\frac{1}{n}\sum_{t=T-n+1}^{T} \frac{1}{t} \leq \frac{1}{T-n+1} \overset{(a)}{\leq} \frac{2}{T} \quad \text{and} \quad \frac{1}{n}\sum_{t=T-n+1}^{T} 1 + \log t \leq 1 + \log T,$$

where we use $T \geq 2(n-1)$ in $(a)$. Finally, we obtain

$$\mathbb{E}\left[F(\boldsymbol{x}_{T+1}^{\mathsf{suffix}}) - F(\boldsymbol{z})\right] \leq O\left(\frac{\|\boldsymbol{z} - \boldsymbol{x}_1\|^2}{\eta T} + \eta\sqrt{n}G_{f,1}G_{f,2}(1 + \log T)\right).$$

- For the stepsize $\eta_t = \frac{\eta}{\sqrt{t}}, \forall t \in [T]$, we invoke the third result in Theorem A.1 and follow almost the same steps for proving Corollary 4.3 to conclude.

$\square$

We emphasize two important implications of Corollary A.2.

The first one is that under the same setting in the proof of lower bound (Koren et al., 2022), i.e., $\psi = I_{\mathcal{C}}$ where $\mathcal{C}$ is the unit ball, $G_i \equiv 4$, $T = Kn$, $\boldsymbol{x}_1 = \boldsymbol{0}$, the existence of $\boldsymbol{x}_\star \in \operatorname{argmin}_{\boldsymbol{x}\in\mathbb{R}^d}F(\boldsymbol{x})$, and $\eta_t \equiv \eta$, we have

$$\mathbb{E}\left[F(\boldsymbol{x}_{Kn+1}^{\mathsf{suffix}}) - F_\star\right] \leq O\left(\frac{1}{\eta nK} + \eta\sqrt{n}(1 + \log nK)\right),$$

where $F_\star = F(\boldsymbol{x}_\star)$. In addition, notice that now

$$F(\boldsymbol{x}_{Kn+1}^{\mathsf{suffix}}) - F_\star = f(\boldsymbol{x}_{Kn+1}^{\mathsf{suffix}}) - f(\boldsymbol{x}_\star) \leq G_{f,1}\left\|\boldsymbol{x}_{Kn+1}^{\mathsf{suffix}} - \boldsymbol{x}_\star\right\| \leq 8 = O(1).$$

---

[4]This requirement can be further relaxed to $T \geq (1 + c)(n - 1)$ for any $c > 0$. We simply choose $c = 1$ here.

We hence obtain

$$\mathbb{E}\left[F(\boldsymbol{x}_{Kn+1}^{\mathsf{suffix}}) - F_\star\right] \le O\left(\min\left\{1, \frac{1}{\eta n K} + \eta\sqrt{n}(1 + \log nK)\right\}\right). \tag{7}$$

Remarkably, this rate matches the lower bound $\Omega\left(\min\left\{1, \frac{1}{\eta n K} + \eta\sqrt{n}\right\}\right)$ of the suffix average for the last one epoch exactly by up to an extra logarithmic factor. Therefore, we (almost) close the gap.

The second one is related to the discussion in Footnote 2, which is that we suspect the rate $O\left(\frac{n^{1/4}}{\sqrt{T}}\right)$ (or $O\left(\frac{1}{n^{1/4}\sqrt{K}}\right)$ when $T = Kn$ where $K \in \mathbb{N}$) is tight for the last iterate under the RR sampling scheme in the general convex case. By the first implication above, one can see our last-iterate bound $\widetilde{O}\left(\frac{1}{\eta T} + \eta\sqrt{n}\right)$ for the constant stepsize $\eta_t \equiv \eta$ (we only include $\eta$, $n$, and $T$ in the rate for simplicity) is almost tight by up to extra logarithmic factors. This is because if one can establish the following bound

$$\mathbb{E}\left[F(\boldsymbol{x}_{T+1}) - F_\star\right] \le o\left(\frac{1}{\eta T} + \eta\sqrt{n}\right),$$

then, using the same proof of (7), there will be

$$\mathbb{E}\left[F(\boldsymbol{x}_{T+1}^{\mathsf{suffix}}) - F_\star\right] \le o\left(\min\left\{1, \frac{1}{\eta T} + \eta\sqrt{n}\right\}\right),$$

which contradicts the lower bound $\Omega\left(\min\left\{1, \frac{1}{\eta n K} + \eta\sqrt{n}\right\}\right)$ in (Koren et al., 2022). Hence, at least in the case $\eta_t \equiv \eta$, our last-iterate result $\widetilde{O}\left(\frac{1}{\eta T} + \eta\sqrt{n}\right)$ should be tight, which immediately implies that $\widetilde{O}\left(\frac{n^{1/4}}{\sqrt{T}}\right)$ is also tight by picking the optimal $\eta$. As such, we conjecture that the rate $O\left(\frac{n^{1/4}}{\sqrt{T}}\right)$ is tight for RR in the general convex case though we cannot prove it when the stepsize is not constant.

Next, Theorem A.3 shows the convergence rate when $\psi$ is $\mu$-strongly convex, e.g., the common regularizer $\psi(\boldsymbol{x}) = \frac{\mu}{2}\|\boldsymbol{x}\|^2$.

**Theorem A.3.** *(Full version of Theorem 4.4) Under Assumptions 2.1 (with $\mu > 0$) and 2.2, suppose the RR sampling scheme is employed with the stepsize $\eta_t = \frac{m}{\mu t}, \forall t \in [T]$ where $m \in \mathbb{N}$, then for any $\boldsymbol{z} \in \mathbb{R}^d$, Algorithm 1 guarantees*

$$\mathbb{E}\left[F(\boldsymbol{x}_{T+1}) - F(\boldsymbol{z})\right] \le O\left(\frac{\mu\|\boldsymbol{z} - \boldsymbol{x}_1\|^2}{\binom{m+T}{m}} + \frac{m\sqrt{n}G_{f,1}G_{f,2}(1 + \log T)}{\mu T}\right).$$

*Proof.* Note that the RR sampling scheme satisfies $\mathrm{i}(t) \overset{\mathcal{D}}{=} \text{Uniform}[n], \forall t \in [T]$, and the stepsize $\eta_t = \frac{m}{\mu t}$ is non-increasing. Hence, Conditions 1 and 2 in Lemma B.2 are fulfilled. If Condition 3 also holds, i.e., $|\mathbb{E}[\Omega_t(\boldsymbol{x}_s)]| \le \Phi\eta_s, \forall t \in [T], s \in [t]$ for some $\Phi \ge 0$, we can invoke Lemma B.2 to obtain for any $\boldsymbol{z} \in \mathbb{R}^d$,

$$\mathbb{E}\left[F(\boldsymbol{x}_{T+1}) - F(\boldsymbol{z})\right] \le O\left(\frac{\|\boldsymbol{z} - \boldsymbol{x}_1\|^2}{\sum_{t=1}^T \gamma_t\eta_t} + (G_{f,2}^2 + \Phi)\sum_{t=1}^T \frac{\gamma_t\eta_t^2}{\sum_{s=t}^T \gamma_s\eta_s}\right), \tag{8}$$

where $\gamma_t = \prod_{s=1}^{t-1}(1 + \mu\eta_s), \forall t \in [T+1]$.

When $\eta_t = \frac{m}{\mu t}, \forall t \in [T]$, there is

$$\gamma_t = \prod_{s=1}^{t-1}\frac{s + m}{s} = \frac{(m+t-1)!}{m!(t-1)!} = \binom{m+t-1}{m}, \forall t \in [T+1],$$

which implies

$$\gamma_t\eta_t = \frac{(m+t-1)!}{m!(t-1)!} \cdot \frac{m}{\mu t} = \frac{1}{\mu}\binom{m+t-1}{m-1}, \forall t \in [T]. \tag{9}$$

Therefore, we know

$$\sum_{t=1}^T \gamma_t\eta_t = \frac{1}{\mu}\sum_{t=1}^T \binom{m+t-1}{m-1} = \frac{1}{\mu}\sum_{t=1}^T \binom{m+t}{m} - \binom{m+t-1}{m} = \frac{1}{\mu}\left[\binom{m+T}{m} - 1\right], \tag{10}$$

and

$$\sum_{t=1}^{T} \frac{\gamma_t \eta_t^2}{\sum_{s=t}^{T} \gamma_s \eta_s} \overset{(a)}{\leq} \sum_{t=1}^{T} \frac{\eta_t}{T-t+1} = \sum_{t=1}^{T} \frac{m}{\mu(T-t+1)t} = \frac{m}{\mu(T+1)} \sum_{t=1}^{T} \frac{1}{t} + \frac{1}{T+1-t} \leq \frac{2m(1+\log T)}{\mu(T+1)}, \quad (11)$$

where $(a)$ is because $\gamma_t \eta_t$ is non-decreasing in $t$ as shown in (9). We combine (8), (10) and (11) to have

$$\mathbb{E}\left[F(\boldsymbol{x}_{T+1}) - F(\boldsymbol{z})\right] \leq O\left(\frac{\mu \|\boldsymbol{z} - \boldsymbol{x}_1\|^2}{\binom{m+T}{m}} + \frac{m\left(G_{f,2}^2 + \Phi\right)(1+\log T)}{\mu T}\right). \quad (12)$$

Our last task is to find a constant $\Phi \geq 0$ satisfying

$$|\mathbb{E}\left[\Omega_t(\boldsymbol{x}_s)\right]| \leq \Phi \eta_s, \forall t \in [T], s \in [t].$$

Here we claim

$$\Phi = \sqrt{2} G_{f,2}^2 + 2\sqrt{2n} G_{f,1} G_{f,2}. \quad (13)$$

To prove our statement, we use Lemma B.4 to have:

- If $s \in [(\mathsf{q}(t)-1)n]$, there is

$$\mathbb{E}\left[\Omega_t(\boldsymbol{x}_s)\right] = 0 \Rightarrow |\mathbb{E}\left[\Omega_t(\boldsymbol{x}_s)\right]| = 0 \leq \Phi \eta_s.$$

- If $s \in [t] \setminus [(\mathsf{q}(t)-1)n]$, there is

$$
\begin{aligned}
|\mathbb{E}\left[\Omega_t(\boldsymbol{x}_s)\right]| &\leq \frac{\sqrt{2} G_{f,2}^2}{n} \sum_{j=(\mathsf{q}(t)-1)n+1}^{s-1} \frac{\gamma_j \eta_j}{\gamma_s} + \frac{2\sqrt{2} G_{f,1} G_{f,2}}{n} \sum_{i=(\mathsf{q}(t)-1)n+1}^{s-1} \sqrt{\sum_{j=i}^{s-1} \frac{\gamma_j^2 \eta_j^2}{\gamma_s^2}} \\
&\overset{(b)}{\leq} \frac{\sqrt{2} G_{f,2}^2}{n} \sum_{j=(\mathsf{q}(t)-1)n+1}^{s-1} \eta_s + \frac{2\sqrt{2} G_{f,1} G_{f,2}}{n} \sum_{i=(\mathsf{q}(t)-1)n+1}^{s-1} \sqrt{\sum_{j=i}^{s-1} \eta_s^2} \\
&\overset{(c)}{\leq} \left(\sqrt{2} G_{f,2}^2 + 2\sqrt{2n} G_{f,1} G_{f,2}\right) \eta_s \overset{(13)}{=} \Phi \eta_s,
\end{aligned}
$$

where $(b)$ holds by $\frac{\gamma_j \eta_j}{\gamma_s} \leq \eta_s, \forall j \in [s-1]$ since $\gamma_t \eta_t$ is non-decreasing as shown in (9) and $(c)$ is due to

$$\sum_{j=(\mathsf{q}(t)-1)n+1}^{s-1} \eta_s = (\mathsf{r}(s)-1)\eta_s \leq n\eta_s,$$

$$\sum_{i=(\mathsf{q}(t)-1)n+1}^{s-1} \sqrt{\sum_{j=i}^{s-1} \eta_s^2} = \sum_{i=(\mathsf{q}(t)-1)n+1}^{s-1} \sqrt{s-i}\eta_s \leq n^{\frac{3}{2}}\eta_s.$$

By (12) and (13), we finally have

$$
\begin{aligned}
\mathbb{E}\left[F(\boldsymbol{x}_{T+1}) - F(\boldsymbol{z})\right] &\leq O\left(\frac{\mu \|\boldsymbol{z} - \boldsymbol{x}_1\|^2}{\binom{m+T}{m}} + \frac{m\left(G_{f,2}^2 + \sqrt{n} G_{f,1} G_{f,2}\right)(1+\log T)}{\mu T}\right) \\
&\leq O\left(\frac{\mu \|\boldsymbol{z} - \boldsymbol{x}_1\|^2}{\binom{m+T}{m}} + \frac{m\sqrt{n} G_{f,1} G_{f,2}(1+\log T)}{\mu T}\right),
\end{aligned}
$$

where the last step is by using $G_{f,2} < \sqrt{n} G_{f,1}$. $\qquad \square$

### A.2. SS Sampling Scheme

First, Theorem A.4 below gives the convergence guarantee for general convex functions (i.e., $\mu = 0$). Unlike the previous Theorem A.1, we currently do not know how to design a proper stepsize to get rid of the extra logarithmic factor, which we leave as a future direction. Another crucial fact is that we also have no idea how to set a time-varying stepsize. Loosely speaking, this is because $\Phi$ depends on $T$ now. See our analysis for details.

**Theorem A.4.** *(Full version of Theorem 4.5) Under Assumptions 2.1 and (with $\mu = 0$) and 2.2, suppose the SS sampling scheme is employed with the stepsize $\eta_t = \frac{\eta}{\sqrt{T}}, \forall t \in [T]$, then for any $\boldsymbol{z} \in \mathbb{R}^d$, Algorithm 1 guarantees*

$$\mathbb{E}\left[F(\boldsymbol{x}_{T+1}) - F(\boldsymbol{z})\right] \leq O\left(\left(\left(\frac{\|\boldsymbol{z} - \boldsymbol{x}_1\|^2}{\eta} + \eta\left(\mathsf{q}(T)G_{f,2}^2 \vee \sqrt{n\mathsf{q}(T)}G_{f,1}G_{f,2}\right)(1 + \log T)\right)\frac{1}{\sqrt{T}}\right).$$

*Setting $\eta = \frac{\|\boldsymbol{z} - \boldsymbol{x}_1\|}{\sqrt{\left(\mathsf{q}(T)G_{f,2}^2 \vee \sqrt{n\mathsf{q}(T)}G_{f,1}G_{f,2}\right)(1 + \log T)}}$ to optimize the dependence on parameters.*

*Proof.* Similar to (2), if we can find a constant $\Phi \geq 0$ such that

$$\left|\mathbb{E}\left[\Omega_t(\boldsymbol{x}_s)\right]\right| \leq \Phi\eta_s, \forall t \in [T], s \in [t].$$

Then there is

$$\mathbb{E}\left[F(\boldsymbol{x}_{T+1}) - F(\boldsymbol{z})\right] \leq O\left(\frac{\|\boldsymbol{z} - \boldsymbol{x}_1\|^2}{\sum_{t=1}^{T}\eta_t} + (G_{f,2}^2 + \Phi)\sum_{t=1}^{T}\frac{\eta_t^2}{\sum_{s=t}^{T}\eta_s}\right), \tag{14}$$

Here, we claim

$$\Phi = 8\mathsf{q}(T)G_{f,2}^2 + 2\sqrt{2n\mathsf{q}(T)}G_{f,1}G_{f,2}. \tag{15}$$

To prove our statement, we invoke Lemma B.6 with $\mu = 0$ to have for any $t \in [T]$ and $s \in [t]$,

$$\begin{aligned}
\left|\mathbb{E}\left[\Omega_t(\boldsymbol{x}_s)\right]\right| \leq & 4G_{f,2}^2\sum_{j=1}^{s-1}\eta_j\left(\mathbb{1}\left[\mathsf{r}(j) = \mathsf{r}(t)\right] + \frac{\mathbb{1}\left[\mathsf{r}(j) \neq \mathsf{r}(t)\right]}{n-1}\right) \\
& + \frac{2}{n}\sum_{i=1}^{n}G_i\sqrt{\sum_{j=1}^{s-1}\eta_j^2\left(G_{f,2}^2 + G_i^2\mathbb{1}\left[\mathsf{r}(j) = \mathsf{r}(t)\right] + \frac{nG_{f,2}^2 - G_i^2}{n-1}\mathbb{1}\left[\mathsf{r}(j) \neq \mathsf{r}(t)\right]\right)} \\
\leq & 4G_{f,2}^2\sum_{j=1}^{\mathsf{q}(T)n}\eta_j\left(\mathbb{1}\left[\mathsf{r}(j) = \mathsf{r}(t)\right] + \frac{\mathbb{1}\left[\mathsf{r}(j) \neq \mathsf{r}(t)\right]}{n-1}\right) \\
& + \frac{2}{n}\sum_{i=1}^{n}G_i\sqrt{\sum_{j=1}^{\mathsf{q}(T)n}\eta_j^2\left(G_{f,2}^2 + G_i^2\mathbb{1}\left[\mathsf{r}(j) = \mathsf{r}(t)\right] + \frac{nG_{f,2}^2 - G_i^2}{n-1}\mathbb{1}\left[\mathsf{r}(j) \neq \mathsf{r}(t)\right]\right)}, \tag{16}
\end{aligned}$$

where the second step is by $s \leq \mathsf{q}(s)n \leq \mathsf{q}(T)n$. Note that when the stepsize is constant, there are

$$\sum_{j=1}^{\mathsf{q}(T)n}\eta_j\left(\mathbb{1}\left[\mathsf{r}(j) = \mathsf{r}(t)\right] + \frac{\mathbb{1}\left[\mathsf{r}(j) \neq \mathsf{r}(t)\right]}{n-1}\right) = \eta_s\sum_{j=1}^{\mathsf{q}(T)n}\left(\mathbb{1}\left[\mathsf{r}(j) = \mathsf{r}(t)\right] + \frac{\mathbb{1}\left[\mathsf{r}(j) \neq \mathsf{r}(t)\right]}{n-1}\right) = 2\mathsf{q}(T)\eta_s, \tag{17}$$

and

$$\begin{aligned}
& \sum_{j=1}^{\mathsf{q}(T)n}\eta_j^2\left(G_{f,2}^2 + G_i^2\mathbb{1}\left[\mathsf{r}(j) = \mathsf{r}(t)\right] + \frac{nG_{f,2}^2 - G_i^2}{n-1}\mathbb{1}\left[\mathsf{r}(j) \neq \mathsf{r}(t)\right]\right) \\
= & \eta_s^2\sum_{j=1}^{\mathsf{q}(T)n}\left(G_{f,2}^2 + G_i^2\mathbb{1}\left[\mathsf{r}(j) = \mathsf{r}(t)\right] + \frac{nG_{f,2}^2 - G_i^2}{n-1}\mathbb{1}\left[\mathsf{r}(j) \neq \mathsf{r}(t)\right]\right) = 2n\mathsf{q}(T)G_{f,2}^2\eta_s^2. \tag{18}
\end{aligned}$$

Combine (16), (17) and (18) to obtain

$$|\mathbb{E}\left[\Omega_t(\boldsymbol{x}_s)\right]| \leq \left(8\mathsf{q}(T)G_{f,2}^2 + 2\sqrt{2n\mathsf{q}(T)}G_{f,1}G_{f,2}\right)\eta_s \overset{(15)}{=} \Phi\eta_s.$$

By (14) and (15), we have

$$\mathbb{E}\left[F(\boldsymbol{x}_{T+1}) - F(\boldsymbol{z})\right] \leq O\left(\frac{\|\boldsymbol{z} - \boldsymbol{x}_1\|^2}{\sum_{t=1}^{T}\eta_t} + \left(G_{f,2}^2 + \mathsf{q}(T)G_{f,2}^2 + \sqrt{n\mathsf{q}(T)}G_{f,1}G_{f,2}\right)\sum_{t=1}^{T}\frac{\eta_t^2}{\sum_{s=t}^{T}\eta_s}\right)$$

$$\overset{(a)}{\leq} O\left(\frac{\|\boldsymbol{z} - \boldsymbol{x}_1\|^2}{\sum_{t=1}^{T}\eta_t} + \left(\mathsf{q}(T)G_{f,2}^2 + \sqrt{n\mathsf{q}(T)}G_{f,1}G_{f,2}\right)\sum_{t=1}^{T}\frac{\eta_t^2}{\sum_{s=t}^{T}\eta_s}\right)$$

$$\overset{(b)}{\leq} O\left(\left(\frac{\|\boldsymbol{z} - \boldsymbol{x}_1\|^2}{\eta} + \eta\left(\mathsf{q}(T)G_{f,2}^2 + \sqrt{n\mathsf{q}(T)}G_{f,1}G_{f,2}\right)(1+\log T)\right)\frac{1}{\sqrt{T}}\right)$$

$$= O\left(\left(\frac{\|\boldsymbol{z} - \boldsymbol{x}_1\|^2}{\eta} + \eta\left(\mathsf{q}(T)G_{f,2}^2 \vee \sqrt{n\mathsf{q}(T)}G_{f,1}G_{f,2}\right)(1+\log T)\right)\frac{1}{\sqrt{T}}\right),$$

where $(a)$ is due to $1 \leq \mathsf{q}(T)$ and $(b)$ holds by plugging in $\eta_t = \frac{\eta}{\sqrt{T}}, \forall t \in [T]$.

Finally, setting $\eta = \frac{\|\boldsymbol{z} - \boldsymbol{x}_1\|}{\sqrt{\left(\mathsf{q}(T)G_{f,2}^2 \vee \sqrt{n\mathsf{q}(T)}G_{f,1}G_{f,2}\right)(1+\log T)}}$ to obtain

$$\mathbb{E}\left[F(\boldsymbol{x}_{T+1}) - F(\boldsymbol{z})\right] \leq O\left(\frac{\sqrt{\mathsf{q}(T)G_{f,2}^2 \vee \sqrt{n\mathsf{q}(T)}G_{f,1}G_{f,2}}\|\boldsymbol{z} - \boldsymbol{x}_1\|\sqrt{1+\log T}}{\sqrt{T}}\right)$$

$$\overset{(c)}{\leq} O\left(\left(\frac{n^{1/4}\sqrt{G_{f,1}G_{f,2}}}{\sqrt{T}} \vee \frac{\sqrt{G_{f,1}G_{f,2}}}{T^{1/4}} \vee \frac{G_{f,2}}{\sqrt{n}}\right)\|\boldsymbol{z} - \boldsymbol{x}_1\|\sqrt{1+\log T}\right), \qquad (19)$$

where $(c)$ is by noticing $\mathsf{q}(T) \leq \frac{T}{n} + 1$ and $G_{f,2} < \sqrt{n}G_{f,1}$, together implying

$$\mathsf{q}(T)G_{f,2}^2 \vee \sqrt{n\mathsf{q}(T)}G_{f,1}G_{f,2} \leq \left(\frac{T}{n}G_{f,2}^2 + G_{f,2}^2\right) \vee \sqrt{T+n}G_{f,1}G_{f,2}$$

$$= O\left(\frac{T}{n}G_{f,2}^2 \vee G_{f,2}^2 \vee \left(\sqrt{T \vee n}G_{f,1}G_{f,2}\right)\right)$$

$$\leq O\left(\frac{T}{n}G_{f,2}^2 \vee \sqrt{n}G_{f,1}G_{f,2} \vee \left(\sqrt{T \vee n}G_{f,1}G_{f,2}\right)\right)$$

$$= O\left(\sqrt{n}G_{f,1}G_{f,2} \vee \sqrt{T}G_{f,1}G_{f,2} \vee \frac{T}{n}G_{f,2}^2\right).$$

When $T \geq n$, we observe that $\frac{n^{1/4}}{\sqrt{T}} \leq \frac{1}{T^{1/4}}$, (19) thus reduces to the following form used in Theorem 4.5

$$\mathbb{E}\left[F(\boldsymbol{x}_{T+1}) - F(\boldsymbol{z})\right] \leq O\left(\left(\frac{\sqrt{G_{f,1}G_{f,2}}}{T^{1/4}} \vee \frac{G_{f,2}}{\sqrt{n}}\right)\|\boldsymbol{z} - \boldsymbol{x}_1\|\sqrt{1+\log T}\right).$$

$\square$

The above Theorem A.4 (or see (19) for the final bound) is not very satisfying as the rate will be blocked at $\widetilde{O}\left(\frac{1}{\sqrt{n}}\right)$ even if $T$ approaches $+\infty$. As discussed in the main text, if $\psi = I_{\mathcal{C}}$ and $T = Kn$ where $K \in \mathbb{N}$, we actually can do better. Note that $\psi = I_{\mathcal{C}}$ can be further relaxed to $\psi = \varphi + I_{\mathcal{C}}$ for $\varphi$ being $G_{\psi}$-Lipschitz on $\mathcal{C}$. However, this will introduce new parameters $G_{\psi}$ in the final rate. Instead, we choose to keep the following simple form. For why the relaxation holds, see Lemma B.3.

**Theorem A.5.** *(Full version of Theorem 4.6) Under Assumptions 2.1 (with $\psi = I_{\mathcal{C}}$) and 2.2, suppose $T = Kn$ where $K \in \mathbb{N}$ and the* **SS** *sampling scheme is employed with the stepsize $\eta_t = \frac{\eta}{\sqrt{T}}, \forall t \in [T]$, then for any $\boldsymbol{z} \in \mathbb{R}^d$, Algorithm 1 guarantees*

$$\mathbb{E}\left[F(\boldsymbol{x}_{Kn+1}) - F(\boldsymbol{z})\right] \le O\left(\left(\frac{\|\boldsymbol{z} - \boldsymbol{x}_1\|^2}{\eta} + \eta\left(\sqrt{nK}G_{f,1}G_{f,2} \wedge nG_{f,1}^2\right)(1 + \log nK)\right)\frac{1}{\sqrt{nK}}\right).$$

*Setting $\eta = \frac{\|\boldsymbol{z} - \boldsymbol{x}_1\|}{\sqrt{\left(\sqrt{nK}G_{f,1}G_{f,2} \wedge nG_{f,1}^2\right)(1 + \log nK)}}$ to optimize the dependence on parameters.*

*Proof.* Under the stepsize $\eta_t = \frac{\eta}{\sqrt{T}}, \forall t \in [T]$, we first know by Theorem A.4,

$$\mathbb{E}\left[F(\boldsymbol{x}_{T+1}) - F(\boldsymbol{z})\right] \le O\left(\left(\frac{\|\boldsymbol{z} - \boldsymbol{x}_1\|^2}{\eta} + \eta\left(\mathsf{q}(T)G_{f,2}^2 \vee \sqrt{n\mathsf{q}(T)}G_{f,1}G_{f,2}\right)(1 + \log T)\right)\frac{1}{\sqrt{T}}\right)$$

$$\Rightarrow \mathbb{E}\left[F(\boldsymbol{x}_{Kn+1}) - F(\boldsymbol{z})\right] \le O\left(\left(\frac{\|\boldsymbol{z} - \boldsymbol{x}_1\|^2}{\eta} + \eta\left(KG_{f,2}^2 \vee \sqrt{nK}G_{f,1}G_{f,2}\right)(1 + \log nK)\right)\frac{1}{\sqrt{nK}}\right), \quad (20)$$

where in the second line we use $\mathsf{q}(T) = K$ when $T = Kn$.

Next, note that the **SS** sampling scheme and the stepsize $\eta_t = \frac{\eta}{\sqrt{T}}, \forall t \in [T]$ respectively satisfy Conditions 1 and 2 in Lemma B.3. Moreover, we can take $\varphi = 0$ and $G_\psi = 0$ in Condition 3. Hence there is almost surely for any $\boldsymbol{z} \in \mathbb{R}^d$,

$$F(\boldsymbol{x}_{Kn+1}) - F(\boldsymbol{z}) \le O\left(\left(\frac{\|\boldsymbol{z} - \boldsymbol{x}_1\|^2}{\eta} + \eta n G_{f,1}^2 \sum_{k=1}^{K} \frac{1}{K - k + 1}\right)\frac{1}{\sqrt{nK}}\right)$$

$$\le O\left(\left(\frac{\|\boldsymbol{z} - \boldsymbol{x}_1\|^2}{\eta} + \eta n G_{f,1}^2 (1 + \log K)\right)\frac{1}{\sqrt{nK}}\right),$$

which implies

$$\mathbb{E}\left[F(\boldsymbol{x}_{Kn+1}) - F(\boldsymbol{z})\right] \le O\left(\left(\frac{\|\boldsymbol{z} - \boldsymbol{x}_1\|^2}{\eta} + \eta n G_{f,1}^2 (1 + \log K)\right)\frac{1}{\sqrt{nK}}\right). \quad (21)$$

We combine (20) and (21) to obtain

$$\mathbb{E}\left[F(\boldsymbol{x}_{Kn+1}) - F(\boldsymbol{z})\right] \le O\left(\left(\frac{\|\boldsymbol{z} - \boldsymbol{x}_1\|^2}{\eta} + \eta\left(\left(KG_{f,2}^2 \vee \sqrt{nK}G_{f,1}G_{f,2}\right) \wedge nG_{f,1}^2\right)(1 + \log nK)\right)\frac{1}{\sqrt{nK}}\right)$$

$$= O\left(\left(\frac{\|\boldsymbol{z} - \boldsymbol{x}_1\|^2}{\eta} + \eta\left(\sqrt{nK}G_{f,1}G_{f,2} \wedge nG_{f,1}^2\right)(1 + \log nK)\right)\frac{1}{\sqrt{nK}}\right),$$

where we use the fact $(a \vee \sqrt{ab}) \wedge b = \sqrt{ab} \wedge b$ for $a = KG_{f,2}^2$ and $b = nG_{f,1}^2$. Finally, Setting $\eta = \frac{\|\boldsymbol{z} - \boldsymbol{x}_1\|}{\sqrt{\left(\sqrt{nK}G_{f,1}G_{f,2} \wedge nG_{f,1}^2\right)(1 + \log nK)}}$ to obtain

$$\mathbb{E}\left[F(\boldsymbol{x}_{Kn+1}) - F(\boldsymbol{z})\right] \le O\left(\left(\frac{\sqrt{G_{f,1}G_{f,2}}}{n^{1/4}K^{1/4}} \wedge \frac{G_{f,1}}{\sqrt{K}}\right)\|\boldsymbol{z} - \boldsymbol{x}_1\|\sqrt{1 + \log nK}\right).$$

$\square$

Finally, we establish the convergence upper bound for strongly convex functions in Theorem A.6. As mentioned after Theorem 4.7, we actually can improve the rate further to avoid the $\widetilde{O}\left(\frac{1}{n}\right)$ barrier when $T = Kn$ where $K \in \mathbb{N}$ and $\psi = \varphi + I_{\mathcal{C}}$ for $\varphi$ being Lipschitz on $\mathcal{C}$. However, the rate in that case will be in the order $\widetilde{O}\left(\frac{n}{K}\right)$ for larger $K$, which is still slower than the bound $\widetilde{O}\left(\frac{1}{K}\right)$ in (Liu & Zhou, 2024b). Therefore, we do not provide it here. See Lemma B.3 for why we can do at most $\widetilde{O}\left(\frac{n}{K}\right)$.

**Theorem A.6.** *(Full version of Theorem 4.7) Under Assumptions 2.1 (with $\mu > 0$) and 2.2, suppose the $\mathsf{SS}$ sampling scheme is employed with the stepsize $\eta_t = \frac{m}{\mu t}, \forall t \in [T]$ where $m \in \mathbb{N}$, then for any $\boldsymbol{z} \in \mathbb{R}^d$, Algorithm 1 guarantees*

$$\mathbb{E}\left[F(\boldsymbol{x}_{T+1}) - F(\boldsymbol{z})\right] \le O\left(\frac{\mu \|\boldsymbol{z} - \boldsymbol{x}_1\|^2}{\binom{m+T}{m}} + \frac{m(1 + \log T)}{\mu}\left(\frac{\sqrt{n}G_{f,1}G_{f,2}}{T} + \frac{G_{f,1}G_{f,2}}{\sqrt{T}} + \frac{G_{f,2}^2}{n}\right)\right).$$

*Proof.* Note that the $\mathsf{SS}$ sampling scheme satisfies $\mathsf{i}(t) \overset{\mathcal{D}}{=} \mathrm{Uniform}\,[n]\,, \forall t \in [T]$, and the stepsize $\eta_t = \frac{m}{\mu t}$ is non-increasing. Hence, Conditions 1 and 2 in Lemma B.2 are fulfilled. If Condition 3 also holds, i.e., $|\mathbb{E}\left[\Omega_t(\boldsymbol{x}_s)\right]| \le \Phi\eta_s, \forall t \in [T]\,, s \in [t]$ for some $\Phi \ge 0$, we can invoke Lemma B.2 and follow the same steps of proving (12) to obtain for any $\boldsymbol{z} \in \mathbb{R}^d$,

$$\mathbb{E}\left[F(\boldsymbol{x}_{T+1}) - F(\boldsymbol{z})\right] \le O\left(\frac{\mu \|\boldsymbol{z} - \boldsymbol{x}_1\|^2}{\binom{m+T}{m}} + \frac{m\left(G_{f,2}^2 + \Phi\right)(1 + \log T)}{\mu T}\right), \tag{22}$$

and know

$$\gamma_t \eta_t = \frac{1}{\mu}\binom{m+t-1}{m-1}, \forall t \in [T]. \tag{23}$$

Our last task is to find a constant $\Phi \ge 0$ satisfying

$$|\mathbb{E}\left[\Omega_t(\boldsymbol{x}_s)\right]| \le \Phi\eta_s, \forall t \in [T]\,, s \in [t].$$

Here we claim

$$\Phi = 8\left(\frac{T}{n} + 1\right)G_{f,2}^2 + 2\sqrt{2(T+n)}G_{f,1}G_{f,2}. \tag{24}$$

To prove our statement, we invoke Lemma B.6 to have for any $t \in [T]$ and $s \in [t]$,

$$
\begin{aligned}
|\mathbb{E}\left[\Omega_t(\boldsymbol{x}_s)\right]| \le& 4G_{f,2}^2 \sum_{j=1}^{s-1} \frac{\gamma_j \eta_j}{\gamma_s}\left(\mathbb{1}\left[\mathsf{r}(j) = \mathsf{r}(t)\right] + \frac{\mathbb{1}\left[\mathsf{r}(j) \ne \mathsf{r}(t)\right]}{n-1}\right) \\
&+ \frac{2}{n}\sum_{i=1}^{n} G_i \sqrt{\sum_{j=1}^{s-1} \frac{\gamma_j^2 \eta_j^2}{\gamma_s^2}\left(G_{f,2}^2 + G_i^2 \mathbb{1}\left[\mathsf{r}(j) = \mathsf{r}(t)\right] + \frac{nG_{f,2}^2 - G_i^2}{n-1}\mathbb{1}\left[\mathsf{r}(j) \ne \mathsf{r}(t)\right]\right)} \\
\le& 4G_{f,2}^2 \eta_s \sum_{j=1}^{s-1} \mathbb{1}\left[\mathsf{r}(j) = \mathsf{r}(t)\right] + \frac{\mathbb{1}\left[\mathsf{r}(j) \ne \mathsf{r}(t)\right]}{n-1} \\
&+ \frac{2\eta_s}{n}\sum_{i=1}^{n} G_i \sqrt{\sum_{j=1}^{s-1} G_{f,2}^2 + G_i^2 \mathbb{1}\left[\mathsf{r}(j) = \mathsf{r}(t)\right] + \frac{nG_{f,2}^2 - G_i^2}{n-1}\mathbb{1}\left[\mathsf{r}(j) \ne \mathsf{r}(t)\right]}, \tag{25}
\end{aligned}
$$

where the last step is due to $\frac{\gamma_j \eta_j}{\gamma_s} \le \eta_s, \forall j \in [s-1]$ because $\gamma_t \eta_t$ is non-decreasing in $t$ as shown in (23). Note that $s \le \mathsf{q}(s)n$, we therefore have

$$\sum_{j=1}^{s-1} \mathbb{1}\left[\mathsf{r}(j) = \mathsf{r}(t)\right] + \frac{\mathbb{1}\left[\mathsf{r}(j) \ne \mathsf{r}(t)\right]}{n-1} \le \sum_{j=1}^{\mathsf{q}(s)n} \mathbb{1}\left[\mathsf{r}(j) = \mathsf{r}(t)\right] + \frac{\mathbb{1}\left[\mathsf{r}(j) \ne \mathsf{r}(t)\right]}{n-1} = 2\mathsf{q}(s), \tag{26}$$

and

$$
\begin{aligned}
\sum_{j=1}^{s-1}&\left(G_{f,2}^2 + G_i^2 \mathbb{1}\left[\mathsf{r}(j) = \mathsf{r}(t)\right] + \frac{nG_{f,2}^2 - G_i^2}{n-1}\mathbb{1}\left[\mathsf{r}(j) \ne \mathsf{r}(t)\right]\right) \\
\le& \sum_{j=1}^{\mathsf{q}(s)n} G_{f,2}^2 + G_i^2 \mathbb{1}\left[\mathsf{r}(j) = \mathsf{r}(t)\right] + \frac{nG_{f,2}^2 - G_i^2}{n-1}\mathbb{1}\left[\mathsf{r}(j) \ne \mathsf{r}(t)\right] \\
=& 2n\mathsf{q}(s)G_{f,2}^2. \tag{27}
\end{aligned}
$$

Combine (25), (26) and (27) to obtain

$$\left| \mathbb{E}\left[\Omega_t(\boldsymbol{x}_s)\right]\right| \le 8\mathsf{q}(s)G_{f,2}^2\eta_s + \frac{2\eta_s}{n}\sum_{i=1}^{n}G_i\sqrt{2n\mathsf{q}(s)G_{f,2}^2} = \left(8\mathsf{q}(s)G_{f,2}^2 + 2\sqrt{2n\mathsf{q}(s)}G_{f,1}G_{f,2}\right)\eta_s$$

$$\le \left(8\left(\frac{T}{n}+1\right)G_{f,2}^2 + 2\sqrt{2(T+n)}G_{f,1}G_{f,2}\right)\eta_s \overset{(24)}{=} \Phi\eta_s,$$

where the second inequality holds by $\mathsf{q}(s) \le \mathsf{q}(T) \le \frac{T}{n}+1$.

By (22) and (24), we finally have

$$\mathbb{E}\left[F(\boldsymbol{x}_{T+1}) - F(\boldsymbol{z})\right] \le O\left(\frac{\mu\|\boldsymbol{z}-\boldsymbol{x}_1\|^2}{\binom{m+T}{m}} + \frac{m\left(G_{f,2}^2 + \left(\frac{T}{n}+1\right)G_{f,2}^2 + \sqrt{T+n}G_{f,1}G_{f,2}\right)(1+\log T)}{\mu T}\right)$$

$$= O\left(\frac{\mu\|\boldsymbol{z}-\boldsymbol{x}_1\|^2}{\binom{m+T}{m}} + \frac{m\left(G_{f,2}^2 + \sqrt{n}G_{f,1}G_{f,2} + \sqrt{T}G_{f,1}G_{f,2} + \frac{T}{n}G_{f,2}^2\right)(1+\log T)}{\mu T}\right)$$

$$\le O\left(\frac{\mu\|\boldsymbol{z}-\boldsymbol{x}_1\|^2}{\binom{m+T}{m}} + \frac{m(1+\log T)}{\mu}\left(\frac{\sqrt{n}G_{f,1}G_{f,2}}{T} + \frac{G_{f,1}G_{f,2}}{\sqrt{T}} + \frac{G_{f,2}^2}{n}\right)\right),$$

where the last step is by $G_{f,2} < \sqrt{n}G_{f,1}$. To recover the rate stated in Theorem 4.7, we only need to take $m = 2$ and observe that $\frac{\sqrt{n}}{T} \le \frac{1}{\sqrt{T}}$ when $T \ge n$. $\qquad\square$

## B. Theoretical Analysis

This section includes our theoretical analysis in detail.

### B.1. General Lemmas

As mentioned in 5, the high-level idea is inspired by (Liu & Zhou, 2024a). However, several important modifications are required to circumvent the potential issue caused by the general $\mathsf{i}(t)$ considered in our setting. To begin with, we first characterize the progress made in every single step, provided in the following Lemma B.1. Note this result holds for any $\mathsf{i}(t)$ even if it does not equal $\mathrm{Uniform}\,[n]$ in distribution.

**Lemma B.1.** *Under Assumptions 2.1 and 2.2, for any $t \in [T]$ and $\boldsymbol{y} \in \mathbb{R}^d$, Algorithm 1 guarantees*

$$F(\boldsymbol{x}_{t+1}) - F(\boldsymbol{y}) \le \frac{\|\boldsymbol{y}-\boldsymbol{x}_t\|^2}{2\eta_t} - (1+\mu\eta_t)\frac{\|\boldsymbol{y}-\boldsymbol{x}_{t+1}\|^2}{2\eta_t} + \eta_t\left(G_{\mathsf{i}(t)}^2 + G_{f,1}^2\right) + \Omega_t(\boldsymbol{y}) - \Omega_t(\boldsymbol{x}_t),$$

*where $\Omega_t(\boldsymbol{y}) \triangleq f_{\mathsf{i}(t)}(\boldsymbol{y}) - f(\boldsymbol{y}), \forall \boldsymbol{y} \in \mathbb{R}^d$.*

*Proof.* Let $t \in [T]$ be fixed. If $\boldsymbol{y} \notin \mathrm{dom}\psi$, then the inequality holds automatically since $F(\boldsymbol{y}) = +\infty$ and $F(\boldsymbol{x}_{t+1}) < +\infty$ almost surely. Thus, we only need to consider the case $\boldsymbol{y} \in \mathrm{dom}\psi$. By the convexity of $f_{\mathsf{i}(t)}$, we have

$$f_{\mathsf{i}(t)}(\boldsymbol{x}_t) - f_{\mathsf{i}(t)}(\boldsymbol{y}) \le \langle\nabla f_{\mathsf{i}(t)}(\boldsymbol{x}_t), \boldsymbol{x}_t - \boldsymbol{y}\rangle = \langle\nabla f_{\mathsf{i}(t)}(\boldsymbol{x}_t), \boldsymbol{x}_{t+1} - \boldsymbol{y}\rangle + \langle\nabla f_{\mathsf{i}(t)}(\boldsymbol{x}_t), \boldsymbol{x}_t - \boldsymbol{x}_{t+1}\rangle. \tag{28}$$

According to the update rule of $\boldsymbol{x}_{t+1}$, there exists $\nabla\psi(\boldsymbol{x}_{t+1}) \in \partial\psi(\boldsymbol{x}_{t+1})$ such that

$$\boldsymbol{0} = \nabla\psi(\boldsymbol{x}_{t+1}) + \nabla f_{\mathsf{i}(t)}(\boldsymbol{x}_t) + \frac{\boldsymbol{x}_{t+1} - \boldsymbol{x}_t}{\eta_t} \Rightarrow \nabla f_{\mathsf{i}(t)}(\boldsymbol{x}_t) = \frac{\boldsymbol{x}_t - \boldsymbol{x}_{t+1}}{\eta_t} - \nabla\psi(\boldsymbol{x}_{t+1}),$$

which gives us

$$\left\langle \nabla f_{i(t)}(\boldsymbol{x}_t), \boldsymbol{x}_{t+1} - \boldsymbol{y} \right\rangle = \frac{\left\langle \boldsymbol{x}_t - \boldsymbol{x}_{t+1}, \boldsymbol{x}_{t+1} - \boldsymbol{y} \right\rangle}{\eta_t} + \left\langle \nabla \psi(\boldsymbol{x}_{t+1}), \boldsymbol{y} - \boldsymbol{x}_{t+1} \right\rangle$$

$$= \frac{\|\boldsymbol{y} - \boldsymbol{x}_t\|^2 - \|\boldsymbol{y} - \boldsymbol{x}_{t+1}\|^2 - \|\boldsymbol{x}_{t+1} - \boldsymbol{x}_t\|^2}{2\eta_t} + \left\langle \nabla \psi(\boldsymbol{x}_{t+1}), \boldsymbol{y} - \boldsymbol{x}_{t+1} \right\rangle$$

$$\leq \frac{\|\boldsymbol{y} - \boldsymbol{x}_t\|^2}{2\eta_t} - (1 + \mu\eta_t)\frac{\|\boldsymbol{y} - \boldsymbol{x}_{t+1}\|^2}{2\eta_t} - \frac{\|\boldsymbol{x}_{t+1} - \boldsymbol{x}_t\|^2}{2\eta_t} + \psi(\boldsymbol{y}) - \psi(\boldsymbol{x}_{t+1}), \quad (29)$$

where the last step holds by the $\mu$-strong convexity of $\psi$. In addition, there is

$$\left\langle \nabla f_{i(t)}(\boldsymbol{x}_t), \boldsymbol{x}_t - \boldsymbol{x}_{t+1} \right\rangle = \left\langle \nabla f_{i(t)}(\boldsymbol{x}_t) - \nabla f(\boldsymbol{x}_{t+1}), \boldsymbol{x}_t - \boldsymbol{x}_{t+1} \right\rangle + \left\langle \nabla f(\boldsymbol{x}_{t+1}), \boldsymbol{x}_t - \boldsymbol{x}_{t+1} \right\rangle$$

$$\leq \left\langle \nabla f_{i(t)}(\boldsymbol{x}_t) - \nabla f(\boldsymbol{x}_{t+1}), \boldsymbol{x}_t - \boldsymbol{x}_{t+1} \right\rangle + f(\boldsymbol{x}_t) - f(\boldsymbol{x}_{t+1}), \quad (30)$$

where the last inequality is due to the convexity of $f$.

Plug (29) and (30) back into (28) and rearrange terms to get (recall $\Omega_t(\cdot) = f_{i(t)}(\cdot) - f(\cdot)$)

$$F(\boldsymbol{x}_{t+1}) - F(\boldsymbol{y}) \leq \frac{\|\boldsymbol{y} - \boldsymbol{x}_t\|^2}{2\eta_t} - (1 + \mu\eta_t)\frac{\|\boldsymbol{y} - \boldsymbol{x}_{t+1}\|^2}{2\eta_t} + \Omega_t(\boldsymbol{y}) - \Omega_t(\boldsymbol{x}_t)$$

$$+ \left\langle \nabla f_{i(t)}(\boldsymbol{x}_t) - \nabla f(\boldsymbol{x}_{t+1}), \boldsymbol{x}_t - \boldsymbol{x}_{t+1} \right\rangle - \frac{\|\boldsymbol{x}_{t+1} - \boldsymbol{x}_t\|^2}{2\eta_t}$$

$$\leq \frac{\|\boldsymbol{y} - \boldsymbol{x}_t\|^2}{2\eta_t} - (1 + \mu\eta_t)\frac{\|\boldsymbol{y} - \boldsymbol{x}_{t+1}\|^2}{2\eta_t} + \eta_t\left(G_{i(t)}^2 + G_{f,1}^2\right) + \Omega_t(\boldsymbol{y}) - \Omega_t(\boldsymbol{x}_t),$$

where the last line follows by

$$\left\langle \nabla f_{i(t)}(\boldsymbol{x}_t) - \nabla f(\boldsymbol{x}_{t+1}), \boldsymbol{x}_t - \boldsymbol{x}_{t+1} \right\rangle$$

$$\overset{(a)}{\leq} \left\| \nabla f_{i(t)}(\boldsymbol{x}_t) - \nabla f(\boldsymbol{x}_{t+1}) \right\| \|\boldsymbol{x}_{t+1} - \boldsymbol{x}_t\| \overset{(b)}{\leq} \left(G_{i(t)} + G_{f,1}\right) \|\boldsymbol{x}_{t+1} - \boldsymbol{x}_t\|$$

$$\overset{(c)}{\leq} \frac{\eta_t\left(G_{i(t)} + G_{f,1}\right)^2}{2} + \frac{\|\boldsymbol{x}_{t+1} - \boldsymbol{x}_t\|^2}{2\eta_t} \leq \eta_t\left(G_{i(t)}^2 + G_{f,1}^2\right) + \frac{\|\boldsymbol{x}_{t+1} - \boldsymbol{x}_t\|^2}{2\eta_t},$$

where $(a)$ is due to Cauchy-Schwarz inequality, $(b)$ is because $f_{i(t)}$ and $f$ are respectively Lipschitz on $\mathrm{dom}\psi$ with the parameter $G_{i(t)}$ and $G_{f,1}$, and $(c)$ is by AM-GM inequality. $\qquad \square$

Next, we are ready to prove our core last-iterate result, Lemma B.2, which provides new sufficient conditions for the last-iterate convergence rate. We remind the reader again that Condition 2 is actually not necessary since we can simply stop the proof at the equation (46) and change $\eta_t^2$ to $\eta_t(\eta_t \vee \eta_{t+1})$ in the final bound.

**Lemma B.2.** *(Full version of Lemma 5.1) Under Assumptions 2.1 and 2.2, suppose the following three conditions hold:*

1. *The index satisfies* $i(t) \overset{\mathcal{D}}{=} \mathrm{Uniform}\,[n]\,, \forall t \in [T]$.

2. *The stepsize* $\eta_t, \forall t \in [T]$ *is non-increasing.*

3. $|\mathbb{E}\,[\Omega_t(\boldsymbol{x}_s)]| \leq \Phi\eta_s, \forall t \in [T]\,, s \in [t]$ *where* $\Phi \geq 0$ *is a constant probably depending on* $T$, $n$, $\mu$, *and* $G_1$ *to* $G_n$

*Then for any* $\boldsymbol{z} \in \mathbb{R}^d$, *Algorithm 1 guarantees*

$$\mathbb{E}\,[F(\boldsymbol{x}_{T+1}) - F(\boldsymbol{z})] \leq O\left(\frac{\|\boldsymbol{z} - \boldsymbol{x}_1\|^2}{\sum_{t=1}^{T}\gamma_t\eta_t} + \left(G_{f,2}^2 + \Phi\right)\sum_{t=1}^{T}\frac{\gamma_t\eta_t^2}{\sum_{s=t}^{T}\gamma_s\eta_s}\right),$$

*where* $\Omega_t(\cdot)$ *is defined in Lemma B.1 and* $\gamma_t \triangleq \prod_{s=1}^{t-1}(1 + \mu\eta_s), \forall t \in [T+1]$.

*Proof.* It is enough to only consider the case $\boldsymbol{z} \in \mathrm{dom}\psi$. In the following proof, we fix a point $\boldsymbol{z} \in \mathrm{dom}\psi$. By Lemma B.1, the following inequality holds for any $t \in [T]$ and $\boldsymbol{y} \in \mathbb{R}^d$,

$$F(\boldsymbol{x}_{t+1}) - F(\boldsymbol{y}) \leq \frac{\|\boldsymbol{y} - \boldsymbol{x}_t\|^2}{2\eta_t} - (1 + \mu\eta_t)\frac{\|\boldsymbol{y} - \boldsymbol{x}_{t+1}\|^2}{2\eta_t} + \eta_t\left(G_{\mathrm{i}(t)}^2 + G_{f,1}^2\right) + \Omega_t(\boldsymbol{y}) - \Omega_t(\boldsymbol{x}_t).$$

Multiplying both sides by $\gamma_t\eta_t$ yields (note that $(1 + \mu\eta_t)\gamma_t = \gamma_{t+1}, \forall t \in [T]$)

$$\gamma_t\eta_t(F(\boldsymbol{x}_{t+1}) - F(\boldsymbol{y})) \leq \frac{\gamma_t\|\boldsymbol{y} - \boldsymbol{x}_t\|^2 - \gamma_{t+1}\|\boldsymbol{y} - \boldsymbol{x}_{t+1}\|^2}{2} + \gamma_t\eta_t^2\left(G_{\mathrm{i}(t)}^2 + G_{f,1}^2\right) + \gamma_t\eta_t(\Omega_t(\boldsymbol{y}) - \Omega_t(\boldsymbol{x}_t)). \quad (31)$$

Next, inspired by (Liu & Zhou, 2024a), we first define the following non-decreasing sequence

$$v_t \triangleq \frac{\gamma_T\eta_T}{\sum_{s=t-1}^T \gamma_s\eta_s}, \forall t \in [T+1] \setminus [1], \quad (32)$$

$$v_1 \triangleq v_2 = \frac{\gamma_T\eta_T}{\sum_{s=1}^T \gamma_s\eta_s}, \quad (33)$$

and then introduce

$$\boldsymbol{z}_t \triangleq \frac{v_1}{v_t}\boldsymbol{z} + \sum_{s=1}^{t-1} \frac{v_{s+1} - v_s}{v_t}\boldsymbol{x}_s, \forall t \in [T+1]. \quad (34)$$

Note that $\boldsymbol{z}_t$ also falls in $\mathrm{dom}\psi$ as it is a convex combination of points in $\mathrm{dom}\psi$ and admits

$$\boldsymbol{z}_{t+1} = \frac{v_t}{v_{t+1}}\boldsymbol{z}_t + \left(1 - \frac{v_t}{v_{t+1}}\right)\boldsymbol{x}_t, \forall t \in [T]. \quad (35)$$

For any $t \in [T]$, we invoke (31) with $\boldsymbol{y} = \boldsymbol{z}_{t+1}$ to obtain

$$\gamma_t\eta_t(F(\boldsymbol{x}_{t+1}) - F(\boldsymbol{z}_{t+1}))$$
$$\leq \frac{\gamma_t\|\boldsymbol{z}_{t+1} - \boldsymbol{x}_t\|^2 - \gamma_{t+1}\|\boldsymbol{z}_{t+1} - \boldsymbol{x}_{t+1}\|^2}{2} + \gamma_t\eta_t^2\left(G_{\mathrm{i}(t)}^2 + G_{f,1}^2\right) + \gamma_t\eta_t(\Omega_t(\boldsymbol{z}_{t+1}) - \Omega_t(\boldsymbol{x}_t))$$
$$\leq \frac{\gamma_t\frac{v_t}{v_{t+1}}\|\boldsymbol{z}_t - \boldsymbol{x}_t\|^2 - \gamma_{t+1}\|\boldsymbol{z}_{t+1} - \boldsymbol{x}_{t+1}\|^2}{2} + \gamma_t\eta_t^2\left(G_{\mathrm{i}(t)}^2 + G_{f,1}^2\right) + \gamma_t\eta_t(\Omega_t(\boldsymbol{z}_{t+1}) - \Omega_t(\boldsymbol{x}_t)), \quad (36)$$

where the second inequality is by $\|\boldsymbol{z}_{t+1} - \boldsymbol{x}_t\|^2 \leq \frac{v_t}{v_{t+1}}\|\boldsymbol{z}_t - \boldsymbol{x}_t\|^2 + \left(1 - \frac{v_t}{v_{t+1}}\right)\|\boldsymbol{x}_t - \boldsymbol{x}_t\|^2 = \frac{v_t}{v_{t+1}}\|\boldsymbol{z}_t - \boldsymbol{x}_t\|^2$ due to the convexity of $\|\cdot\|^2$ and (35). Multiply both sides of (36) by $v_{t+1}$ and sum up from $t = 1$ to $T$ to obtain

$$\sum_{t=1}^T \gamma_t\eta_t v_{t+1}(F(\boldsymbol{x}_{t+1}) - F(\boldsymbol{z}_{t+1}))$$
$$\leq \frac{\gamma_1 v_1\|\boldsymbol{z}_1 - \boldsymbol{x}_1\|^2 - \gamma_{T+1}v_{T+1}\|\boldsymbol{z}_{T+1} - \boldsymbol{x}_{T+1}\|^2}{2} + \sum_{t=1}^T \gamma_t\eta_t^2 v_{t+1}\left(G_{\mathrm{i}(t)}^2 + G_{f,1}^2\right) + \gamma_t\eta_t v_{t+1}(\Omega_t(\boldsymbol{z}_{t+1}) - \Omega_t(\boldsymbol{x}_t))$$
$$\leq \frac{\gamma_1 v_1\|\boldsymbol{z}_1 - \boldsymbol{x}_1\|^2}{2} + \sum_{t=1}^T \gamma_t\eta_t^2 v_{t+1}\left(G_{\mathrm{i}(t)}^2 + G_{f,1}^2\right) + \gamma_t\eta_t v_{t+1}(\Omega_t(\boldsymbol{z}_{t+1}) - \Omega_t(\boldsymbol{x}_t))$$
$$\overset{(34)}{=} \frac{\gamma_1 v_1\|\boldsymbol{z} - \boldsymbol{x}_1\|^2}{2} + \sum_{t=1}^T \gamma_t\eta_t^2 v_{t+1}\left(G_{\mathrm{i}(t)}^2 + G_{f,1}^2\right) + \gamma_t\eta_t v_{t+1}(\Omega_t(\boldsymbol{z}_{t+1}) - \Omega_t(\boldsymbol{x}_t)). \quad (37)$$

Now we focus on the term $\sum_{t=1}^{T} \gamma_t \eta_t v_{t+1} \Omega_t(\boldsymbol{z}_{t+1})$ and observe that

$$
\begin{aligned}
\sum_{t=1}^{T} \gamma_t \eta_t v_{t+1} \Omega_t(\boldsymbol{z}_{t+1}) &= \sum_{t=1}^{T} \gamma_t \eta_t v_{t+1} (f_{\mathrm{i}(t)}(\boldsymbol{z}_{t+1}) - f(\boldsymbol{z}_{t+1})) \\
&\overset{(a)}{\leq} \sum_{t=1}^{T} \gamma_t \eta_t \left[ v_1 f_{\mathrm{i}(t)}(\boldsymbol{z}) + \sum_{s=1}^{t} (v_{s+1} - v_s) f_{\mathrm{i}(t)}(\boldsymbol{x}_s) - v_{t+1} f(\boldsymbol{z}_{t+1}) \right] \\
&\overset{(b)}{=} \sum_{t=1}^{T} \gamma_t \eta_t \left[ v_1 (\Omega_t(\boldsymbol{z}) + f(\boldsymbol{z})) + \sum_{s=1}^{t} (v_{s+1} - v_s)(\Omega_t(\boldsymbol{x}_s) + f(\boldsymbol{x}_s)) - v_{t+1} f(\boldsymbol{z}_{t+1}) \right] \\
&= \sum_{t=1}^{T} \gamma_t \eta_t \left[ v_1 \Omega_t(\boldsymbol{z}) + \sum_{s=1}^{t} (v_{s+1} - v_s) \Omega_t(\boldsymbol{x}_s) \right] \\
&\quad + \sum_{t=1}^{T} \gamma_t \eta_t \left[ v_1 f(\boldsymbol{z}) + \sum_{s=1}^{t} (v_{s+1} - v_s) f(\boldsymbol{x}_s) - v_{t+1} f(\boldsymbol{z}_{t+1}) \right],
\end{aligned}
\tag{38}
$$

where $(a)$ is due to $f_{\mathrm{i}(t)}(\boldsymbol{z}_{t+1}) \leq \frac{v_1}{v_{t+1}} f_{\mathrm{i}(t)}(\boldsymbol{z}) + \sum_{s=1}^{t} \frac{v_{s+1} - v_s}{v_{t+1}} f_{\mathrm{i}(t)}(\boldsymbol{x}_s)$ by the convexity of $f_{\mathrm{i}(t)}$ and the definition of $\boldsymbol{z}_{t+1}$ in (34) and $(b)$ holds by recalling that $\Omega_t(\cdot) = f_{\mathrm{i}(t)}(\cdot) - f(\cdot)$.

Combine (37) and (38) to have

$$
\sum_{t=1}^{T} \gamma_t \eta_t v_{t+1} (F(\boldsymbol{x}_{t+1}) - F(\boldsymbol{z}_{t+1})) \leq \frac{\gamma_1 v_1 \|\boldsymbol{z} - \boldsymbol{x}_1\|^2}{2} + \sum_{t=1}^{T} \gamma_t \eta_t^2 v_{t+1} \left( G_{\mathrm{i}(t)}^2 + G_{f,1}^2 \right) + \square + \blacksquare,
\tag{39}
$$

where

$$
\square \triangleq \sum_{t=1}^{T} \gamma_t \eta_t \left[ v_1 \Omega_t(\boldsymbol{z}) + \sum_{s=1}^{t} (v_{s+1} - v_s) \Omega_t(\boldsymbol{x}_s) - v_{t+1} \Omega_t(\boldsymbol{x}_t) \right],
\tag{40}
$$

$$
\blacksquare \triangleq \sum_{t=1}^{T} \gamma_t \eta_t \left[ v_1 f(\boldsymbol{z}) + \sum_{s=1}^{t} (v_{s+1} - v_s) f(\boldsymbol{x}_s) - v_{t+1} f(\boldsymbol{z}_{t+1}) \right].
\tag{41}
$$

We bound term $\blacksquare$ as follows

$$
\begin{aligned}
\blacksquare &= \sum_{t=1}^{T} \gamma_t \eta_t \left[ v_1 (F(\boldsymbol{z}) - \psi(\boldsymbol{z})) + \sum_{s=1}^{t} (v_{s+1} - v_s)(F(\boldsymbol{x}_s) - \psi(\boldsymbol{x}_s)) - v_{t+1} (F(\boldsymbol{z}_{t+1}) - \psi(\boldsymbol{z}_{t+1})) \right] \\
&= \sum_{t=1}^{T} \gamma_t \eta_t \left[ v_1 F(\boldsymbol{z}) + \sum_{s=1}^{t} (v_{s+1} - v_s) F(\boldsymbol{x}_s) - v_{t+1} F(\boldsymbol{z}_{t+1}) + v_{t+1} \psi(\boldsymbol{z}_{t+1}) - \sum_{s=1}^{t} (v_{s+1} - v_s) \psi(\boldsymbol{x}_s) - v_1 \psi(\boldsymbol{z}) \right] \\
&\overset{(c)}{\leq} \sum_{t=1}^{T} \gamma_t \eta_t \left[ v_1 F(\boldsymbol{z}) + \sum_{s=1}^{t} (v_{s+1} - v_s) F(\boldsymbol{x}_s) - v_{t+1} F(\boldsymbol{z}_{t+1}) \right] \\
&= \sum_{t=1}^{T} \gamma_t \eta_t \left[ \sum_{s=1}^{t} (v_{s+1} - v_s)(F(\boldsymbol{x}_s) - F(\boldsymbol{z})) - v_{t+1} (F(\boldsymbol{z}_{t+1}) - F(\boldsymbol{z})) \right] \\
&= \sum_{s=1}^{T} \left( \sum_{t=s}^{T} \gamma_t \eta_t \right) (v_{s+1} - v_s)(F(\boldsymbol{x}_s) - F(\boldsymbol{z})) - \sum_{t=1}^{T} \gamma_t \eta_t v_{t+1} (F(\boldsymbol{z}_{t+1}) - F(\boldsymbol{z})) \\
&\overset{(d)}{=} \sum_{s=2}^{T} \gamma_{s-1} \eta_{s-1} v_s (F(\boldsymbol{x}_s) - F(\boldsymbol{z})) - \sum_{t=1}^{T} \gamma_t \eta_t v_{t+1} (F(\boldsymbol{z}_{t+1}) - F(\boldsymbol{z})),
\end{aligned}
\tag{42}
$$

where $(c)$ holds by $\psi(\boldsymbol{z}_{t+1}) \leq \frac{v_1}{v_{t+1}} \psi(\boldsymbol{z}) + \sum_{s=1}^{t} \frac{v_{s+1} - v_s}{v_{t+1}} \psi(\boldsymbol{x}_s)$ due to the convexity of $\psi$ and the definition of $\boldsymbol{z}_{t+1}$ in (34) and $(d)$ is by noticing that

- if $s = 1$,

$$v_2 - v_1 \stackrel{(33)}{=} 0; \tag{43}$$

- if $s \in [T] \setminus [1]$,

$$v_{s+1} - v_s \stackrel{(32)}{=} \frac{\gamma_T \eta_T}{\sum_{t=s}^T \gamma_t \eta_t} - \frac{\gamma_T \eta_T}{\sum_{t=s-1}^T \gamma_t \eta_t} = \frac{\gamma_T \eta_T \gamma_{s-1} \eta_{s-1}}{(\sum_{t=s}^T \gamma_t \eta_t)(\sum_{t=s-1}^T \gamma_t \eta_t)} \stackrel{(32)}{=} \frac{\gamma_{s-1} \eta_{s-1} v_s}{\sum_{t=s}^T \gamma_t \eta_t},$$

which implies

$$\left( \sum_{t=s}^T \gamma_t \eta_t \right) (v_{s+1} - v_s) = \gamma_{s-1} \eta_{s-1} v_s. \tag{44}$$

We then plug (42) back into (39) and rearrange terms to get

$$\gamma_T \eta_T v_{T+1} \left( F(\boldsymbol{x}_{T+1}) - F(\boldsymbol{z}) \right) \leq \frac{\gamma_1 v_1 \|\boldsymbol{z} - \boldsymbol{x}_1\|^2}{2} + \sum_{t=1}^T \gamma_t \eta_t^2 v_{t+1} \left( G_{\mathsf{i}(t)}^2 + G_{f,1}^2 \right) + \square.$$

Take expectations and divide by $\gamma_T \eta_T$ on both sides and then use $v_{T+1} \stackrel{(32)}{=} 1$ and $\gamma_1 = 1$ to obtain

$$\mathbb{E}\left[ F(\boldsymbol{x}_{T+1}) - F(\boldsymbol{z}) \right] \leq \frac{v_1 \|\boldsymbol{z} - \boldsymbol{x}_1\|^2}{2\gamma_T \eta_T} + \sum_{t=1}^T \frac{\gamma_t \eta_t^2 v_{t+1}}{\gamma_T \eta_T} \left( \mathbb{E}\left[ G_{\mathsf{i}(t)}^2 \right] + G_{f,1}^2 \right) + \frac{\mathbb{E}[\square]}{\gamma_T \eta_T}$$

$$\leq \frac{v_1 \|\boldsymbol{z} - \boldsymbol{x}_1\|^2}{2\gamma_T \eta_T} + 2G_{f,2}^2 \sum_{t=1}^T \frac{\gamma_t \eta_t^2 v_{t+1}}{\gamma_T \eta_T} + \frac{\mathbb{E}[\square]}{\gamma_T \eta_T}, \tag{45}$$

where the last line holds due to $\mathsf{i}(t) \stackrel{\mathcal{D}}{=} \mathrm{Uniform}\,[n]\,, \forall t \in [T] \Rightarrow \mathbb{E}\left[ G_{\mathsf{i}(t)}^2 \right] = \frac{1}{n} \sum_{i=1}^n G_i^2 = G_{f,2}^2$ and $G_{f,1} \leq G_{f,2}$.

Lastly, we note that

$$\mathbb{E}[\square] \stackrel{(40)}{=} \sum_{t=1}^T \gamma_t \eta_t \left[ v_1 \mathbb{E}\left[ \Omega_t(\boldsymbol{z}) \right] + \sum_{s=1}^t (v_{s+1} - v_s) \mathbb{E}\left[ \Omega_t(\boldsymbol{x}_s) \right] - v_{t+1} \mathbb{E}\left[ \Omega_t(\boldsymbol{x}_t) \right] \right]$$

$$\stackrel{(e)}{=} \sum_{t=1}^T \gamma_t \eta_t \left[ \sum_{s=1}^t (v_{s+1} - v_s) \mathbb{E}\left[ \Omega_t(\boldsymbol{x}_s) \right] - v_{t+1} \mathbb{E}\left[ \Omega_t(\boldsymbol{x}_t) \right] \right]$$

$$\stackrel{(f)}{\leq} \sum_{t=1}^T \gamma_t \eta_t \left[ \sum_{s=1}^t (v_{s+1} - v_s) \cdot \Phi \eta_s + v_{t+1} \cdot \Phi \eta_t \right]$$

$$= \Phi \left[ \sum_{s=1}^T \left( \sum_{t=s}^T \gamma_t \eta_t \right) (v_{s+1} - v_s) \eta_s + \sum_{t=1}^T \gamma_t \eta_t^2 v_{t+1} \right]$$

$$\stackrel{(43),(44)}{=} \Phi \left[ \sum_{s=2}^T \gamma_{s-1} \eta_{s-1} v_s \eta_s + \sum_{t=1}^T \gamma_t \eta_t^2 v_{t+1} \right] \tag{46}$$

$$\stackrel{(g)}{\leq} 2\Phi \sum_{t=1}^T \gamma_t \eta_t^2 v_{t+1}, \tag{47}$$

where $(e)$ is due to $\mathbb{E}\left[ \Omega_t(\boldsymbol{z}) \right] = \mathbb{E}\left[ f_{\mathsf{i}(t)}(\boldsymbol{z}) - f(\boldsymbol{z}) \right] = 0, \forall t \in [T]$ since $\mathsf{i}(t) \stackrel{\mathcal{D}}{=} \mathrm{Uniform}\,[n]\,, \forall t \in [T]$, $(f)$ holds by the condition of $|\mathbb{E}\left[ \Omega_t(\boldsymbol{x}_s) \right]| \leq \Phi \eta_s, \forall t \in [T]\,, s \in [t]$, and $(g)$ is by the condition of the non-increasing $\eta_t, \forall t \in [T]$. We thereby finish the proof by plugging (47) back into (45) and applying the definition of $v_t$ in (32) and (33). $\square$

Though Lemma B.2 is enough for the RR case, as the reader already has seen, it is inadequate for the SS case. So what do we miss? The problem is that we never use the fact $\{\mathsf{i}((k-1)n+1), \cdots, \mathsf{i}(kn)\} = [n]\,, \forall k \in [K]$ if $T = Kn$ for $K \in \mathbb{N}$,

which is the major factor employed in the analysis of (Liu & Zhou, 2024b). So similar to (Liu & Zhou, 2024b), we may also hope for a last-iterate convergence bound that holds almost surely. However, due to a different structure in our Algorithm 1, i.e., the place for the proximal update, the existing analysis and rates in (Liu & Zhou, 2024b) are immediately invalid. Hence, it is unknown whether Algorithm 1 still guarantees a last-iterate rate in the same order as (Liu & Zhou, 2024b) (or equivalently, Proximal GD).

Fortunately, we can still prove a similar almost surely convergence bound with two extra requirements Conditions 2 and 3. Condition 2 here is standard if we consider the whole epoch as a single update like (Liu & Zhou, 2024b). In contrast, Condition 3 is more artificial and makes the rate depend on a new parameter $G_\psi$. We currently do not know whether it can be removed or not.

Lastly, we point out that, if ignoring $G_\psi$ (or one can think it is smaller than $G_{f,1}$), the rate in Lemma B.3 for the general convex case is in the same order as (Liu & Zhou, 2024b). This is why we can obtain the improved result in Theorem A.5. However, for the strongly convex case, Lemma B.3 is worse than (Liu & Zhou, 2024b) because the parameter $\mu$ in our $\widetilde{\gamma}_k$ is worse than a factor of $n$ compared to (Liu & Zhou, 2024b) (in other words, they have $n\mu$). This crucial point leads us to a worse rate $\widetilde{O}\left(\frac{n}{K}\right)$ (e.g., setting $\widetilde{\eta}_k = \frac{2}{\mu k}, \forall k \in [K]$) than Proximal GD. We currently do not know whether it is fixable. If it is, one can improve Theorem A.6 to show $\mathsf{SS}$ also beats Proximal GD for any $K \in \mathbb{N}$ (again, ignoring $G_\psi$).

**Lemma B.3.** *Under Assumptions 2.1 and 2.2, suppose $T = Kn$ where $K \in \mathbb{N}$ and the following three conditions hold:*

1. *The index satisfies $\{i((k-1)n+1), \cdots, i(kn)\} = [n], \forall k \in [K]$.*

2. *The stepsize satisfies $\eta_t = \widetilde{\eta}_{q(t)}, \forall t \in [T]$ where $\widetilde{\eta}_k, \forall k \in [K]$ is another positive sequence.*

3. *$\psi = \varphi + I_\mathcal{C}$ where $\varphi : \mathbb{R}^d \to \mathbb{R}$ is convex and $G_\psi$-Lipschitz on $\mathrm{dom}\psi = \mathcal{C}$.*

*Then for any $z \in \mathbb{R}^d$, Algorithm 1 guarantees almost surely*

$$F(x_{Kn+1}) - F(z) \le O\left(\frac{\|z - x_1\|^2}{n \sum_{k=1}^K \widetilde{\gamma}_k \widetilde{\eta}_k} + n\left(G_{f,1}^2 + G_\psi^2\right) \sum_{k=1}^K \frac{\widetilde{\gamma}_k \widetilde{\eta}_k^2}{\sum_{\ell=k}^K \widetilde{\gamma}_\ell \widetilde{\eta}_\ell}\right),$$

*where $\widetilde{\gamma}_k \triangleq \prod_{\ell=1}^{k-1}(1 + \mu\widetilde{\eta}_\ell), \forall k \in [K+1]$.*

*Proof.* Given $y \in \mathrm{dom}\psi = \mathcal{C}$, for any $t \in [T]$, we have by (29)

$$\langle \nabla f_{i(t)}(x_t), x_{t+1} - y \rangle \le \frac{\|y - x_t\|^2}{2\eta_t} - (1 + \mu\eta_t)\frac{\|y - x_{t+1}\|^2}{2\eta_t} - \frac{\|x_{t+1} - x_t\|^2}{2\eta_t} + \psi(y) - \psi(x_{t+1}).$$

Combine the above inequality with (28) to obtain

$$\begin{aligned}
&f_{i(t)}(x_t) + \psi(x_{t+1}) - f_{i(t)}(y) - \psi(y)\\
&\le \langle \nabla f_{i(t)}(x_t), x_t - x_{t+1} \rangle + \frac{\|y - x_t\|^2}{2\eta_t} - (1 + \mu\eta_t)\frac{\|y - x_{t+1}\|^2}{2\eta_t} - \frac{\|x_{t+1} - x_t\|^2}{2\eta_t}\\
&\le \frac{\|y - x_t\|^2}{2\eta_t} - (1 + \mu\eta_t)\frac{\|y - x_{t+1}\|^2}{2\eta_t} + \frac{\eta_t G_{i(t)}^2}{2},
\end{aligned} \tag{48}$$

where the second step is by applying Cauchy-Schwarz inequality, Assumption 2.2 and AM-GM inequality to have

$$\langle \nabla f_{i(t)}(x_t), x_t - x_{t+1} \rangle \le \left\|\nabla f_{i(t)}(x_t)\right\| \|x_t - x_{t+1}\| \le G_{i(t)} \|x_t - x_{t+1}\| \le \frac{\eta_t G_{i(t)}^2}{2} + \frac{\|x_{t+1} - x_t\|^2}{2\eta_t}.$$

We sum up (48) from $t = (k-1)n + 1$ to $kn$ for a fixed $k \in [K]$ and notice $\eta_t = \widetilde{\eta}_k$ by Condition 2 to obtain

$$\begin{aligned}
&\sum_{t=(k-1)n+1}^{kn} f_{i(t)}(x_t) + \psi(x_{t+1}) - f_{i(t)}(y) - \psi(y)\\
&\le \frac{\left\|y - x_{(k-1)n+1}\right\|^2 - (1 + \mu\widetilde{\eta}_k)\|y - x_{kn+1}\|^2}{2\widetilde{\eta}_k} + \frac{\widetilde{\eta}_k}{2}\sum_{t=(k-1)n+1}^{kn} G_{i(t)}^2.
\end{aligned}$$

One more step, by Condition 1, we can rewrite the above inequality into

$$F(\boldsymbol{x}_{kn+1}) - F(\boldsymbol{y}) \leq \frac{\left\|\boldsymbol{y} - \boldsymbol{x}_{(k-1)n+1}\right\|^2 - (1+\mu\widetilde{\eta}_k)\left\|\boldsymbol{y} - \boldsymbol{x}_{kn+1}\right\|^2}{2n\widetilde{\eta}_k} + \frac{\widetilde{\eta}_k G_{f,2}^2}{2}$$

$$+ \frac{1}{n} \sum_{t=(k-1)n+1}^{kn} f_{\mathsf{i}(t)}(\boldsymbol{x}_{kn+1}) - f_{\mathsf{i}(t)}(\boldsymbol{x}_t) + \psi(\boldsymbol{x}_{kn+1}) - \psi(\boldsymbol{x}_{t+1}). \tag{49}$$

We now prove bound on $\|\boldsymbol{x}_{s+1} - \boldsymbol{x}_s\|$, which will be used later. Observe that when $\psi = \varphi + I_\mathcal{C}$, there is

$$\boldsymbol{x}_{s+1} = \mathrm{argmin}_{\boldsymbol{x}\in\mathbb{R}^d}\psi(\boldsymbol{x}) + \left\langle \nabla f_{\mathsf{i}(s)}(\boldsymbol{x}_s), \boldsymbol{x}\right\rangle + \frac{\|\boldsymbol{x} - \boldsymbol{x}_s\|^2}{2\eta_s}$$

$$= \mathrm{argmin}_{\boldsymbol{x}\in\mathcal{C}}\varphi(\boldsymbol{x}) + \left\langle \nabla f_{\mathsf{i}(s)}(\boldsymbol{x}_s), \boldsymbol{x}\right\rangle + \frac{\|\boldsymbol{x} - \boldsymbol{x}_s\|^2}{2\eta_s},$$

which implies that, by the optimality condition, there exists $\nabla\varphi(\boldsymbol{x}_{s+1}) \in \partial\varphi(\boldsymbol{x}_{s+1})$ such that for any $\boldsymbol{z} \in \mathcal{C}$,

$$\left\langle \nabla\varphi(\boldsymbol{x}_{s+1}) + \nabla f_{\mathsf{i}(s)}(\boldsymbol{x}_s) + \frac{\boldsymbol{x}_{s+1} - \boldsymbol{x}_s}{\eta_s}, \boldsymbol{x}_{s+1} - \boldsymbol{z}\right\rangle \leq 0.$$

In particular, set $\boldsymbol{z} = \boldsymbol{x}_s$ to have

$$\|\boldsymbol{x}_{s+1} - \boldsymbol{x}_s\|^2 \leq \eta_s \left\langle \nabla\varphi(\boldsymbol{x}_{s+1}) + \nabla f_{\mathsf{i}(s)}(\boldsymbol{x}_s), \boldsymbol{x}_s - \boldsymbol{x}_{s+1}\right\rangle$$

$$\Rightarrow \|\boldsymbol{x}_{s+1} - \boldsymbol{x}_s\| \leq \eta_s \left\|\nabla\varphi(\boldsymbol{x}_{s+1}) + \nabla f_{\mathsf{i}(s)}(\boldsymbol{x}_s)\right\| \leq \widetilde{\eta}_k \left(G_\psi + G_{\mathsf{i}(s)}\right), \tag{50}$$

where in the last step we use $\eta_s = \widetilde{\eta}_k$ when $s \in \{(k-1)n+1, \cdots, kn\}$ by Condition 2 and $\varphi$ is $G_\psi$-Lipschitz on $\mathcal{C}$ by Condition 3.

Now notice that

$$\sum_{t=(k-1)n+1}^{kn} f_{\mathsf{i}(t)}(\boldsymbol{x}_{kn+1}) - f_{\mathsf{i}(t)}(\boldsymbol{x}_t)$$

$$\overset{(a)}{\leq} \sum_{t=(k-1)n+1}^{kn} G_{\mathsf{i}(t)}\|\boldsymbol{x}_{kn+1} - \boldsymbol{x}_t\| \leq \sum_{t=(k-1)n+1}^{kn}\sum_{s=t}^{kn} G_{\mathsf{i}(t)}\|\boldsymbol{x}_{s+1} - \boldsymbol{x}_s\|$$

$$\overset{(50)}{\leq} \sum_{t=(k-1)n+1}^{kn}\sum_{s=t}^{kn} \widetilde{\eta}_k G_{\mathsf{i}(t)}\left(G_\psi + G_{\mathsf{i}(s)}\right) \leq \widetilde{\eta}_k\left(n^2 G_{f,1}G_\psi + \frac{n^2 G_{f,1}^2 - nG_{f,2}^2}{2}\right), \tag{51}$$

where $(a)$ is by Assumption 2.2. Moreover, there is

$$\sum_{t=(k-1)n+1}^{kn} \psi(\boldsymbol{x}_{kn+1}) - \psi(\boldsymbol{x}_{t+1})$$

$$= \sum_{t=(k-1)n+1}^{kn} \varphi(\boldsymbol{x}_{kn+1}) - \varphi(\boldsymbol{x}_{t+1}) = \sum_{t=(k-1)n+1}^{kn-1} \varphi(\boldsymbol{x}_{kn+1}) - \varphi(\boldsymbol{x}_{t+1})$$

$$\overset{(b)}{\leq} G_\psi \sum_{t=(k-1)n+1}^{kn-1} \|\boldsymbol{x}_{kn+1} - \boldsymbol{x}_{t+1}\| \leq G_\psi \sum_{t=(k-1)n+1}^{kn-1}\sum_{s=t+1}^{kn} \|\boldsymbol{x}_{s+1} - \boldsymbol{x}_s\|$$

$$\overset{(50)}{\leq} G_\psi \sum_{t=(k-1)n+1}^{kn-1}\sum_{s=t+1}^{kn} \widetilde{\eta}_k\left(G_\psi + G_{\mathsf{i}(s)}\right) \leq \widetilde{\eta}_k n(n-1)\left(\frac{G_\psi^2}{2} + G_\psi G_{f,1}\right), \tag{52}$$

where $(b)$ is by Condition 3.

Combine (49), (51) and (52) to have for any $\boldsymbol{y} \in \operatorname{dom}\psi = \mathcal{C}$ and $k \in [K]$,

$$F(\boldsymbol{x}_{kn+1}) - F(\boldsymbol{y}) \leq \frac{\left\| \boldsymbol{y} - \boldsymbol{x}_{(k-1)n+1} \right\|^2 - (1 + \mu\widetilde{\eta}_k) \left\| \boldsymbol{y} - \boldsymbol{x}_{kn+1} \right\|^2}{2n\widetilde{\eta}_k} + \widetilde{\eta}_k n \left(G_{f,1} + G_\psi\right)^2 .$$

Note that this inequality also holds for $\boldsymbol{y} \notin \mathcal{C}$. Hence, the above result is true for any $\boldsymbol{y} \in \mathbb{R}^d$. We can then follow similar steps of proving Lemma B.2 to finally obtain for any $\boldsymbol{z} \in \mathbb{R}^d$,

$$
\begin{aligned}
F(\boldsymbol{x}_{kn+1}) - F(\boldsymbol{z}) &\leq O\left( \frac{\|\boldsymbol{z} - \boldsymbol{x}_1\|^2}{n \sum_{k=1}^{K} \widetilde{\gamma}_k \widetilde{\eta}_k} + n \left(G_{f,1} + G_\psi\right)^2 \sum_{k=1}^{K} \frac{\widetilde{\gamma}_k \widetilde{\eta}_k^2}{\sum_{\ell=k}^{K} \widetilde{\gamma}_\ell \widetilde{\eta}_\ell} \right) \\
&= O\left( \frac{\|\boldsymbol{z} - \boldsymbol{x}_1\|^2}{n \sum_{k=1}^{K} \widetilde{\gamma}_k \widetilde{\eta}_k} + n \left(G_{f,1}^2 + G_\psi^2\right) \sum_{k=1}^{K} \frac{\widetilde{\gamma}_k \widetilde{\eta}_k^2}{\sum_{\ell=k}^{K} \widetilde{\gamma}_\ell \widetilde{\eta}_\ell} \right),
\end{aligned}
$$

where $\widetilde{\gamma}_k \triangleq \prod_{\ell=1}^{k-1}(1 + \mu\widetilde{\eta}_\ell), \forall k \in [K+1]$. $\qquad\square$

## B.2. Analysis for the RR Sampling Scheme

Due to Lemma B.2, our task reduces to bound $|\mathbb{E}\left[\Omega_t(\boldsymbol{x}_s)\right]|, \forall t \in [T], s \in [t]$. In this subsection, we will show how to bound it under the RR sampling scheme. The main idea is inspired by (Bassily et al., 2020; Sherman et al., 2021; Koren et al., 2022), where they only bound $|\mathbb{E}\left[\Omega_t(\boldsymbol{x}_t)\right]|, \forall t \in [T]$ in a simpler setting, i.e., $\psi = I_\mathcal{C}, G_i \equiv G$, and constant stepsize. Here, we provide a fine-grained analysis for $|\mathbb{E}\left[\Omega_t(\boldsymbol{x}_s)\right]|, \forall t \in [T], s \in [t]$ working for the broader setting considered in our paper. As mentioned, the finer dependence on $G_{f,1}$ and $G_{f,2}$ is the key to help us obtain improved results over (Liu & Zhou, 2024b).

**Lemma B.4.** *Under Assumptions 2.1 and 2.2, suppose the RR sampling scheme is employed, then for any $t \in [T]$ and $s \in [t]$, let $\mathsf{q} \triangleq \mathsf{q}(t)$, Algorithm 1 guarantees*

- $|\mathbb{E}\left[\Omega_t(\boldsymbol{x}_s)\right]| = 0$ *if $s \in [(\mathsf{q}-1)n]$;*

- $|\mathbb{E}\left[\Omega_t(\boldsymbol{x}_s)\right]| \leq \frac{\sqrt{2}G_{f,2}^2}{n} \sum_{j=(\mathsf{q}-1)n+1}^{s-1} \frac{\gamma_j \eta_j}{\gamma_s} + \frac{2\sqrt{2}G_{f,1}G_{f,2}}{n} \sum_{i=(\mathsf{q}-1)n+1}^{s-1} \sqrt{\sum_{j=i}^{s-1} \frac{\gamma_j^2 \eta_j^2}{\gamma_s^2}}$ *if $s \in [t] \setminus [(\mathsf{q}-1)n]$;*

*where $\Omega_t(\cdot)$ and $\gamma_t$ are defined in Lemmas B.1 and B.2, respectively.*

*Proof.* Let $\mathcal{F}_t \triangleq \sigma(\mathsf{i}(1), \cdots, \mathsf{i}(t)), \forall t \in [T]$ denote the natural filtration and $\mathcal{F}_0$ be the trivial $\sigma$-algebra. Note that $\boldsymbol{x}_t \in \mathcal{F}_{t-1}, \forall t \in [T]$. Additionally, we use $\mathsf{r} \triangleq \mathsf{r}(s)$ in the following.

- Given $s \in [(\mathsf{q}-1)n]$, we know $\boldsymbol{x}_s \in \mathcal{F}_{(\mathsf{q}-1)n}$ and $\mathsf{i}(t) \overset{\mathcal{D}}{=} \text{Uniform}\,[n]$ conditioned on $\mathcal{F}_{(\mathsf{q}-1)n}$ under the RR sampling scheme. Therefore, $\mathbb{E}\left[f_{\mathsf{i}(t)}(\boldsymbol{x}_s) \mid \mathcal{F}_{(\mathsf{q}-1)n}\right] = \frac{1}{n} \sum_{i=1}^{n} f_i(\boldsymbol{x}_s) = f(\boldsymbol{x}_s)$, which implies

$$\mathbb{E}\left[\Omega_t(\boldsymbol{x}_s)\right] = \mathbb{E}\left[f_{\mathsf{i}(t)}(\boldsymbol{x}_s) - f(\boldsymbol{x}_s)\right] = \mathbb{E}\left[\mathbb{E}\left[f_{\mathsf{i}(t)}(\boldsymbol{x}_s) \mid \mathcal{F}_{(\mathsf{q}-1)n}\right] - f(\boldsymbol{x}_s)\right] = 0.$$

- Given $s \in [t] \setminus [(\mathsf{q}-1)n]$, we know $\mathsf{i}(t)$ and $\mathsf{i}(s)$ have the same distribution conditioned on $\mathcal{F}_{s-1}$ under the RR sampling scheme. Hence, there is

$$\mathbb{E}\left[f_{\mathsf{i}(t)}(\boldsymbol{x}_s) \mid \mathcal{F}_{s-1}\right] = \mathbb{E}\left[f_{\mathsf{i}(s)}(\boldsymbol{x}_s) \mid \mathcal{F}_{s-1}\right] \Rightarrow \mathbb{E}\left[f_{\mathsf{i}(t)}(\boldsymbol{x}_s)\right] = \mathbb{E}\left[f_{\mathsf{i}(s)}(\boldsymbol{x}_s)\right] \Rightarrow \mathbb{E}\left[\Omega_t(\boldsymbol{x}_s)\right] = \mathbb{E}\left[\Omega_s(\boldsymbol{x}_s)\right]. \quad (53)$$

Note that $\mathsf{q}(s) = \mathsf{q}$ when $s \in [t] \setminus [(\mathsf{q}-1)n]$. Thus, the index $\mathsf{i}(s)$ satisfies

$$\mathsf{i}(s) = \pi_{\mathsf{q}(s)}^{\mathsf{r}(s)} = \pi_{\mathsf{q}}^{\mathsf{r}}. \quad (54)$$

Therefore, we know

$$f(\boldsymbol{x}_s) = \frac{1}{n}\sum_{i=1}^{n} f_i(\boldsymbol{x}_s) = \frac{1}{n}\sum_{i=1}^{r-1} f_{\pi_{\mathsf{q}}^i}(\boldsymbol{x}_s) + \frac{1}{n}\sum_{i=\mathsf{r}}^{n} f_{\pi_{\mathsf{q}}^i}(\boldsymbol{x}_s)$$

$$= \frac{1}{n}\sum_{i=1}^{r-1} f_{\pi_{\mathsf{q}}^i}(\boldsymbol{x}_s) + \frac{n-\mathsf{r}+1}{n}\mathbb{E}\left[f_{\pi_{\mathsf{q}}^r}(\boldsymbol{x}_s) \mid \mathcal{F}_{s-1}\right]$$

$$\overset{(54)}{=} \frac{1}{n}\sum_{i=1}^{r-1} f_{\pi_{\mathsf{q}}^i}(\boldsymbol{x}_s) + \frac{n-\mathsf{r}+1}{n}\mathbb{E}\left[f_{\mathsf{i}(s)}(\boldsymbol{x}_s) \mid \mathcal{F}_{s-1}\right]$$

$$\Rightarrow \mathbb{E}\left[f(\boldsymbol{x}_s)\right] = \frac{1}{n}\sum_{i=1}^{r-1}\mathbb{E}\left[f_{\pi_{\mathsf{q}}^i}(\boldsymbol{x}_s)\right] + \frac{n-\mathsf{r}+1}{n}\mathbb{E}\left[f_{\mathsf{i}(s)}(\boldsymbol{x}_s)\right]$$

$$\Rightarrow \mathbb{E}\left[\Omega_s(\boldsymbol{x}_s)\right] = \frac{1}{n}\sum_{i=1}^{r-1}\mathbb{E}\left[f_{\mathsf{i}(s)}(\boldsymbol{x}_s) - f_{\pi_{\mathsf{q}}^i}(\boldsymbol{x}_s)\right] \overset{(54)}{=} \frac{1}{n}\sum_{i=1}^{r-1}\mathbb{E}\left[f_{\pi_{\mathsf{q}}^r}(\boldsymbol{x}_s) - f_{\pi_{\mathsf{q}}^i}(\boldsymbol{x}_s)\right]. \tag{55}$$

Now for any fixed $i \in [\mathsf{r}-1]$, we introduce $\widehat{\pi}_{\mathsf{q}}(\mathsf{r}, i)$, which is generated by exchanging $\pi_{\mathsf{q}}^r$ and $\pi_{\mathsf{q}}^i$ in $\pi_{\mathsf{q}}$, i.e.,

$$\widehat{\pi}_{\mathsf{q}}(\mathsf{r}, i) \triangleq (\pi_{\mathsf{q}}^1, \cdots, \pi_{\mathsf{q}}^{i-1}, \pi_{\mathsf{q}}^r, \pi_{\mathsf{q}}^{i+1}, \cdots, \pi_{\mathsf{q}}^{r-1}, \pi_{\mathsf{q}}^i, \pi_{\mathsf{q}}^{r+1}, \cdots, \pi_{\mathsf{q}}^n). \tag{56}$$

We then define the following sequence of points,

$$\widehat{\boldsymbol{x}}_{(\mathsf{q}-1)n+1}(\mathsf{r}, i) \triangleq \boldsymbol{x}_{(\mathsf{q}-1)n+1}, \tag{57}$$

$$\widehat{\boldsymbol{x}}_{(\mathsf{q}-1)n+1+j}(\mathsf{r}, i) \triangleq \operatorname{argmin}_{\boldsymbol{x}\in\mathbb{R}^d} \psi(\boldsymbol{x}) + \left\langle \nabla f_{\widehat{\pi}_{\mathsf{q}}^j(\mathsf{r},i)}(\widehat{\boldsymbol{x}}_{(\mathsf{q}-1)n+j}(\mathsf{r}, i)), \boldsymbol{x}\right\rangle + \frac{\left\|\boldsymbol{x} - \widehat{\boldsymbol{x}}_{(\mathsf{q}-1)n+j}(\mathsf{r}, i)\right\|^2}{2\eta_{(\mathsf{q}-1)n+j}}, \forall j \in [\mathsf{r}-1]. \tag{58}$$

By Lemma C.3, $\pi_{\mathsf{q}} \overset{\mathcal{D}}{=} \widehat{\pi}_{\mathsf{q}}(\mathsf{r}, i)$, which implies $(\pi_{\mathsf{q}}, \boldsymbol{x}_s) \overset{\mathcal{D}}{=} (\widehat{\pi}_{\mathsf{q}}(\mathsf{r}, i), \widehat{\boldsymbol{x}}_s(\mathsf{r}, i))$[5]. Hence,

$$\mathbb{E}\left[f_{\pi_{\mathsf{q}}^i}(\boldsymbol{x}_s)\right] = \mathbb{E}\left[f_{\widehat{\pi}_{\mathsf{q}}^i(\mathsf{r},i)}(\widehat{\boldsymbol{x}}_s(\mathsf{r}, i))\right] = \mathbb{E}\left[f_{\pi_{\mathsf{q}}^r}(\widehat{\boldsymbol{x}}_s(\mathsf{r}, i))\right],$$

which gives us

$$\mathbb{E}\left[\Omega_t(\boldsymbol{x}_s)\right] \overset{(53)}{=} \mathbb{E}\left[\Omega_s(\boldsymbol{x}_s)\right] \overset{(55)}{=} \frac{1}{n}\sum_{i=1}^{r-1}\mathbb{E}\left[f_{\pi_{\mathsf{q}}^r}(\boldsymbol{x}_s) - f_{\pi_{\mathsf{q}}^i}(\boldsymbol{x}_s)\right] = \frac{1}{n}\sum_{i=1}^{r-1}\mathbb{E}\left[f_{\pi_{\mathsf{q}}^r}(\boldsymbol{x}_s) - f_{\pi_{\mathsf{q}}^r}(\widehat{\boldsymbol{x}}_s(\mathsf{r}, i))\right]$$

$$\Rightarrow \left|\mathbb{E}\left[\Omega_t(\boldsymbol{x}_s)\right]\right| \le \frac{1}{n}\sum_{i=1}^{r-1}\mathbb{E}\left[\left|f_{\pi_{\mathsf{q}}^r}(\boldsymbol{x}_s) - f_{\pi_{\mathsf{q}}^r}(\widehat{\boldsymbol{x}}_s(\mathsf{r}, i))\right|\right] \overset{(a)}{\le} \frac{1}{n}\sum_{i=1}^{r-1}\mathbb{E}\left[G_{\pi_{\mathsf{q}}^r}\left\|\boldsymbol{x}_s - \widehat{\boldsymbol{x}}_s(\mathsf{r}, i)\right\|\right]$$

$$= \frac{1}{n}\sum_{i=1}^{r-1}\mathbb{E}\left[G_{\pi_{\mathsf{q}}^r}\mathbb{E}\left[\left\|\boldsymbol{x}_s - \widehat{\boldsymbol{x}}_s(\mathsf{r}, i)\right\| \mid \pi_{\mathsf{q}}^r\right]\right] \overset{(b)}{\le} \frac{1}{n}\sum_{i=1}^{r-1}\mathbb{E}\left[G_{\pi_{\mathsf{q}}^r}\sqrt{\mathbb{E}\left[\left\|\boldsymbol{x}_s - \widehat{\boldsymbol{x}}_s(\mathsf{r}, i)\right\|^2 \mid \pi_{\mathsf{q}}^r\right]}\right],$$

where $(a)$ is because $f_{\pi_{\mathsf{q}}^r}$ is $G_{\pi_{\mathsf{q}}^r}$-Lipschitz and $(b)$ is due to Hölder's inequality. Finally, we invoke Lemma B.5 to have

$$\left|\mathbb{E}\left[\Omega_t(\boldsymbol{x}_s)\right]\right| \le \frac{1}{n}\sum_{i=1}^{r-1}\mathbb{E}\left[\sqrt{2}G_{\pi_{\mathsf{q}}^r}^2\frac{\gamma_{(\mathsf{q}-1)n+i}\eta_{(\mathsf{q}-1)n+i}}{\gamma_s} + 2\sqrt{2}G_{\pi_{\mathsf{q}}^r}G_{f,2}\sqrt{\sum_{j=i}^{r-1}\frac{\gamma_{(\mathsf{q}-1)n+j}^2\eta_{(\mathsf{q}-1)n+j}^2}{\gamma_s^2}}\right]$$

$$= \frac{\sqrt{2}G_{f,2}^2}{n}\sum_{i=1}^{r-1}\frac{\gamma_{(\mathsf{q}-1)n+i}\eta_{(\mathsf{q}-1)n+i}}{\gamma_s} + \frac{2\sqrt{2}G_{f,1}G_{f,2}}{n}\sum_{i=1}^{r-1}\sqrt{\sum_{j=i}^{r-1}\frac{\gamma_{(\mathsf{q}-1)n+j}^2\eta_{(\mathsf{q}-1)n+j}^2}{\gamma_s^2}}$$

$$= \frac{\sqrt{2}G_{f,2}^2}{n}\sum_{j=(\mathsf{q}-1)n+1}^{s-1}\frac{\gamma_j\eta_j}{\gamma_s} + \frac{2\sqrt{2}G_{f,1}G_{f,2}}{n}\sum_{i=(\mathsf{q}-1)n+1}^{s-1}\sqrt{\sum_{j=i}^{s-1}\frac{\gamma_j^2\eta_j^2}{\gamma_s^2}},$$

---

[5]Strictly speaking, this equation requires that for any $i \in [n]$ and $\boldsymbol{x} \in \mathbb{R}^d$, $\nabla f_i(\boldsymbol{x})$ is deterministically picked from the subgradient set $\partial f_i(\boldsymbol{x})$, which possibly contains more than one element. We assume it holds since this is realistic.

where we use the fact $(\mathsf{q}-1)n + \mathsf{r} = s$ in the final step.

$\square$

**Lemma B.5.** *Under the same settings in Lemma B.4, let $\widehat{\boldsymbol{x}}_s(\mathsf{r}, i)$ be the point defined by (57) and (58), then we have*

$$\mathbb{E}\left[\|\boldsymbol{x}_s - \widehat{\boldsymbol{x}}_s(\mathsf{r}, i)\|^2 \mid \pi_{\mathsf{q}}^{\mathsf{r}}\right] \leq 2G_{\pi_{\mathsf{q}}^{\mathsf{r}}}^2 \frac{\gamma_{(\mathsf{q}-1)n+i}^2 \eta_{(\mathsf{q}-1)n+i}^2}{\gamma_s^2} + 8G_{f,2}^2 \sum_{j=i}^{\mathsf{r}-1} \frac{\gamma_{(\mathsf{q}-1)n+j}^2 \eta_{(\mathsf{q}-1)n+j}^2}{\gamma_s^2},$$

*where $\gamma_t$ is defined in Lemma B.2.*

*Proof.* Note that $\pi_{\mathsf{q}}^j = \widehat{\pi}_{\mathsf{q}}^j(\mathsf{r}, i)$ for all $j \in [i-1]$ by the definition of $\widehat{\pi}_{\mathsf{q}}(\mathsf{r}, i)$ (see (56)) and $\boldsymbol{x}_{(\mathsf{q}-1)n+1} = \widehat{\boldsymbol{x}}_{(\mathsf{q}-1)n+1}(\mathsf{r}, i)$ by the definition (see (57)). Thus, by the definition of $\widehat{\boldsymbol{x}}_{(\mathsf{q}-1)n+1+j}$ (see (58)), there is

$$\boldsymbol{x}_{(\mathsf{q}-1)n+j} = \widehat{\boldsymbol{x}}_{(\mathsf{q}-1)n+j}(\mathsf{r}, i), \forall j \in [i].$$

In the following, we denote by $\boldsymbol{y}_j \triangleq \boldsymbol{x}_{(\mathsf{q}-1)n+j}$ and $\widehat{\boldsymbol{y}}_j \triangleq \widehat{\boldsymbol{x}}_{(\mathsf{q}-1)n+j}(\mathsf{r}, i), \forall j \in \{i, \cdots, \mathsf{r}\}$. Note that there is $\boldsymbol{y}_i = \widehat{\boldsymbol{y}}_i$.

By Lemma C.2,

$$\begin{aligned}
\|\boldsymbol{y}_{i+1} - \widehat{\boldsymbol{y}}_{i+1}\| &\leq \frac{\left\|\boldsymbol{y}_i - \widehat{\boldsymbol{y}}_i - \eta_{(\mathsf{q}-1)n+i}(\nabla f_{\pi_{\mathsf{q}}^i}(\boldsymbol{y}_i) - \nabla f_{\widehat{\pi}_{\mathsf{q}}^i(\mathsf{r},i)}(\widehat{\boldsymbol{y}}_i))\right\|}{1 + \mu\eta_{(\mathsf{q}-1)n+i}} = \frac{\eta_{(\mathsf{q}-1)n+i}\left\|\nabla f_{\pi_{\mathsf{q}}^i}(\boldsymbol{y}_i) - \nabla f_{\widehat{\pi}_{\mathsf{q}}^i(\mathsf{r},i)}(\boldsymbol{y}_i)\right\|}{1 + \mu\eta_{(\mathsf{q}-1)n+i}} \\
&\overset{(56)}{=} \frac{\eta_{(\mathsf{q}-1)n+i}\left\|\nabla f_{\pi_{\mathsf{q}}^i}(\boldsymbol{y}_i) - \nabla f_{\pi_{\mathsf{q}}^{\mathsf{r}}}(\boldsymbol{y}_i)\right\|}{1 + \mu\eta_{(\mathsf{q}-1)n+i}} \leq \frac{\eta_{(\mathsf{q}-1)n+i}(G_{\pi_{\mathsf{q}}^i} + G_{\pi_{\mathsf{q}}^{\mathsf{r}}})}{1 + \mu\eta_{(\mathsf{q}-1)n+i}},
\end{aligned} \tag{59}$$

where the last step is by the Lipschitz property of $f_i$. We invoke Lemma C.2 again to obtain for any $i + 1 \leq j \leq \mathsf{r} - 1$,

$$\begin{aligned}
\|\boldsymbol{y}_{j+1} - \widehat{\boldsymbol{y}}_{j+1}\| &\leq \frac{\left\|\boldsymbol{y}_j - \widehat{\boldsymbol{y}}_j - \eta_{(\mathsf{q}-1)n+j}(\nabla f_{\pi_{\mathsf{q}}^j}(\boldsymbol{y}_j) - \nabla f_{\widehat{\pi}_{\mathsf{q}}^j(\mathsf{r},i)}(\widehat{\boldsymbol{y}}_j))\right\|}{1 + \mu\eta_{(\mathsf{q}-1)n+j}} \\
&\overset{(56)}{=} \frac{\left\|\boldsymbol{y}_j - \widehat{\boldsymbol{y}}_j - \eta_{(\mathsf{q}-1)n+j}(\nabla f_{\pi_{\mathsf{q}}^j}(\boldsymbol{y}_j) - \nabla f_{\pi_{\mathsf{q}}^j}(\widehat{\boldsymbol{y}}_j))\right\|}{1 + \mu\eta_{(\mathsf{q}-1)n+j}},
\end{aligned}$$

which implies

$$\begin{aligned}
\|\boldsymbol{y}_{j+1} - \widehat{\boldsymbol{y}}_{j+1}\|^2 &\leq \frac{\|\boldsymbol{y}_j - \widehat{\boldsymbol{y}}_j\|^2 - 2\eta_{(\mathsf{q}-1)n+j}\left\langle\boldsymbol{y}_j - \widehat{\boldsymbol{y}}_j, \nabla f_{\pi_{\mathsf{q}}^j}(\boldsymbol{y}_j) - \nabla f_{\pi_{\mathsf{q}}^j}(\widehat{\boldsymbol{y}}_j)\right\rangle + \eta_{(\mathsf{q}-1)n+j}^2\left\|\nabla f_{\pi_{\mathsf{q}}^j}(\boldsymbol{y}_j) - \nabla f_{\pi_{\mathsf{q}}^j}(\widehat{\boldsymbol{y}}_j)\right\|^2}{(1 + \mu\eta_{(\mathsf{q}-1)n+j})^2} \\
&\leq \frac{\|\boldsymbol{y}_j - \widehat{\boldsymbol{y}}_j\|^2 + 4\eta_{(\mathsf{q}-1)n+j}^2 G_{\pi_{\mathsf{q}}^j}^2}{(1 + \mu\eta_{(\mathsf{q}-1)n+j})^2},
\end{aligned} \tag{60}$$

where the last line is by

$$\left\langle\boldsymbol{y}_j - \widehat{\boldsymbol{y}}_j, \nabla f_{\pi_{\mathsf{q}}^j}(\boldsymbol{y}_j) - \nabla f_{\pi_{\mathsf{q}}^j}(\widehat{\boldsymbol{y}}_j)\right\rangle \overset{\text{Assumption 2.1}}{\geq} 0,$$

$$\left\|\nabla f_{\pi_{\mathsf{q}}^j}(\boldsymbol{y}_j) - \nabla f_{\pi_{\mathsf{q}}^j}(\widehat{\boldsymbol{y}}_j)\right\|^2 \overset{\text{Assumption 2.2}}{\leq} 4G_{\pi_{\mathsf{q}}^j}^2.$$

Finally, unrolling (60) recursively to obtain

$$\|\boldsymbol{y}_{\mathsf{r}} - \widehat{\boldsymbol{y}}_{\mathsf{r}}\|^2 \leq \frac{\left\|\boldsymbol{y}_{i+1} - \widehat{\boldsymbol{y}}_{i+1}\right\|^2}{\prod_{\ell=i+1}^{\mathsf{r}-1}(1+\mu\eta_{(\mathsf{q}-1)n+\ell})^2} + \sum_{j=i+1}^{\mathsf{r}-1} \frac{4\eta_{(\mathsf{q}-1)n+j}^2 G_{\pi_{\mathsf{q}}^j}^2}{\prod_{\ell=j}^{\mathsf{r}-1}(1+\mu\eta_{(\mathsf{q}-1)n+\ell})^2}$$

$$\overset{(59)}{\leq} \frac{\eta_{(\mathsf{q}-1)n+i}^2 (G_{\pi_{\mathsf{q}}^i} + G_{\pi_{\mathsf{q}}^{\mathsf{r}}})^2}{\prod_{\ell=i}^{\mathsf{r}-1}(1+\mu\eta_{(\mathsf{q}-1)n+\ell})^2} + \sum_{j=i+1}^{\mathsf{r}-1} \frac{4\eta_{(\mathsf{q}-1)n+j}^2 G_{\pi_{\mathsf{q}}^j}^2}{\prod_{\ell=j}^{\mathsf{r}-1}(1+\mu\eta_{(\mathsf{q}-1)n+\ell})^2}$$

$$\leq \frac{2\eta_{(\mathsf{q}-1)n+i}^2 (G_{\pi_{\mathsf{q}}^i}^2 + G_{\pi_{\mathsf{q}}^{\mathsf{r}}}^2)}{\prod_{\ell=i}^{\mathsf{r}-1}(1+\mu\eta_{(\mathsf{q}-1)n+\ell})^2} + \sum_{j=i+1}^{\mathsf{r}-1} \frac{4\eta_{(\mathsf{q}-1)n+j}^2 G_{\pi_{\mathsf{q}}^j}^2}{\prod_{\ell=j}^{\mathsf{r}-1}(1+\mu\eta_{(\mathsf{q}-1)n+\ell})^2},$$

$$\leq \frac{2\eta_{(\mathsf{q}-1)n+i}^2 G_{\pi_{\mathsf{q}}^{\mathsf{r}}}^2}{\prod_{\ell=i}^{\mathsf{r}-1}(1+\mu\eta_{(\mathsf{q}-1)n+\ell})^2} + \sum_{j=i}^{\mathsf{r}-1} \frac{4\eta_{(\mathsf{q}-1)n+j}^2 G_{\pi_{\mathsf{q}}^j}^2}{\prod_{\ell=j}^{\mathsf{r}-1}(1+\mu\eta_{(\mathsf{q}-1)n+\ell})^2}$$

Therefore, we know

$$\mathbb{E}\left[\|\boldsymbol{y}_{\mathsf{r}} - \widehat{\boldsymbol{y}}_{\mathsf{r}}\|^2 \mid \pi_{\mathsf{q}}^{\mathsf{r}}\right] \leq \frac{2\eta_{(\mathsf{q}-1)n+i}^2 G_{\pi_{\mathsf{q}}^{\mathsf{r}}}^2}{\prod_{\ell=i}^{\mathsf{r}-1}(1+\mu\eta_{(\mathsf{q}-1)n+\ell})^2} + \sum_{j=i}^{\mathsf{r}-1} \frac{4\eta_{(\mathsf{q}-1)n+j}^2 \mathbb{E}\left[G_{\pi_{\mathsf{q}}^j}^2 \mid \pi_{\mathsf{q}}^{\mathsf{r}}\right]}{\prod_{\ell=j}^{\mathsf{r}-1}(1+\mu\eta_{(\mathsf{q}-1)n+\ell})^2}$$

$$\overset{(a)}{=} \frac{2\eta_{(\mathsf{q}-1)n+i}^2 G_{\pi_{\mathsf{q}}^{\mathsf{r}}}^2}{\prod_{\ell=i}^{\mathsf{r}-1}(1+\mu\eta_{(\mathsf{q}-1)n+\ell})^2} + \sum_{j=i}^{\mathsf{r}-1} \frac{8\eta_{(\mathsf{q}-1)n+j}^2 G_{f,2}^2}{\prod_{\ell=j}^{\mathsf{r}-1}(1+\mu\eta_{(\mathsf{q}-1)n+\ell})^2}$$

$$\overset{(b)}{=} \frac{2\gamma_{(\mathsf{q}-1)n+i}^2 \eta_{(\mathsf{q}-1)n+i}^2 G_{\pi_{\mathsf{q}}^{\mathsf{r}}}^2}{\gamma_s^2} + \sum_{j=i}^{\mathsf{r}-1} \frac{8\gamma_{(\mathsf{q}-1)n+j}^2 \eta_{(\mathsf{q}-1)n+j}^2 G_{f,2}^2}{\gamma_s^2},$$

where $(a)$ is by (w.l.o.g., we assume $n \geq 2$ now, otherwise, our final bound holds automatically when $n = 1$ since $\boldsymbol{x}_s = \widehat{\boldsymbol{x}}_s(\mathsf{r}, i)$ in that case)

$$\mathbb{E}\left[G_{\pi_{\mathsf{q}}^j}^2 \mid \pi_{\mathsf{q}}^{\mathsf{r}}\right] = \frac{nG_{f,2}^2 - G_{\pi_{\mathsf{q}}^{\mathsf{r}}}^2}{n-1} \leq \frac{nG_{f,2}^2}{n-1} \leq 2G_{f,2}^2, \forall j \neq \mathsf{r},$$

and $(b)$ is due to $\gamma_t = \prod_{s=1}^{t-1}(1+\mu\eta_s), \forall t \in [T+1]$. We hence obtain the desired bound on $\mathbb{E}\left[\|\boldsymbol{x}_s - \widehat{\boldsymbol{x}}_s(\mathsf{r}, i)\|^2 \mid \pi_{\mathsf{q}}^{\mathsf{r}}\right]$ as $\boldsymbol{y}_{\mathsf{r}} = \boldsymbol{x}_{(\mathsf{q}-1)n+\mathsf{r}} = \boldsymbol{x}_s$ and $\widehat{\boldsymbol{y}}_{\mathsf{r}} = \widehat{\boldsymbol{x}}_{(\mathsf{q}-1)n+\mathsf{r}}(\mathsf{r}, i) = \widehat{\boldsymbol{x}}_s(\mathsf{r}, i)$. □

### B.3. Analysis for the SS Sampling Scheme

This subsection helps us to bound $|\mathbb{E}\left[\Omega_t(\boldsymbol{x}_s)\right]|, \forall t \in [T], s \in [t]$ for the SS sampling scheme. The proof is inspired by (Koren et al., 2022). Again, our result can be viewed as a finer generalization than theirs and hence requires more careful analysis.

**Lemma B.6.** *Under Assumptions 2.1 and 2.2, suppose the SS sampling scheme is employed, then for any $t \in [T]$ and $s \in [t]$, let $\mathsf{r} \triangleq \mathsf{r}(t)$, Algorithm 1 guarantees*

$$|\mathbb{E}\left[\Omega_t(\boldsymbol{x}_s)\right]| \leq 4G_{f,2}^2 \sum_{j=1}^{s-1} \frac{\gamma_j \eta_j}{\gamma_s} \left(\mathbb{1}\left[\mathsf{r}(j) = \mathsf{r}\right] + \frac{\mathbb{1}\left[\mathsf{r}(j) \neq \mathsf{r}\right]}{n-1}\right)$$

$$+ \frac{2}{n} \sum_{i=1}^{n} G_i \sqrt{\sum_{j=1}^{s-1} \frac{\gamma_j^2 \eta_j^2}{\gamma_s^2} \left(G_{f,2}^2 + G_i^2 \mathbb{1}\left[\mathsf{r}(j) = \mathsf{r}\right] + \frac{nG_{f,2}^2 - G_i^2}{n-1} \mathbb{1}\left[\mathsf{r}(j) \neq \mathsf{r}\right]\right)},$$

*where $\Omega_t(\cdot)$ and $\gamma_t$ are defined in Lemmas B.1 and B.2, respectively.*

*Proof.* Under the SS sampling scheme, for any $t \in [T]$, $\boldsymbol{x}_t$ can be recognized as being generated by a deterministic map $\mathcal{A}_t$[6] from the permutation $\pi$ to $\mathbb{R}^d$ when the initial point $\boldsymbol{x}_1$ and the stepsize $\eta_t, \forall t \in [T]$ are fixed. In other words, we can

---

[6]Same as Footnote 5, we also assume $\nabla f_i(\boldsymbol{x})$ is deterministically picked from $\partial f_i(\boldsymbol{x})$ for any $i \in [n]$ and $\boldsymbol{x} \in \mathbb{R}^d$.

write

$$\boldsymbol{x}_t = \mathcal{A}_t(\pi), \forall t \in [T].$$

We also recall the following fact about the index

$$\mathsf{i}(t) = \pi^{\mathsf{r}(t)} = \pi^{\mathsf{r}}. \tag{61}$$

Hence, there is

$$\mathbb{E}\left[f_{\mathsf{i}(t)}(\boldsymbol{x}_s)\right] = \mathbb{E}\left[f_{\pi^{\mathsf{r}}}(\mathcal{A}_s(\pi))\right] = \sum_{i=1}^{n} \mathbb{E}\left[f_i(\mathcal{A}_s(\pi))\mathbb{1}\left[\pi^{\mathsf{r}} = i\right]\right].$$

For any $i \in [n]$, let $\widehat{\pi}(\mathsf{r}, \star_i)$ denote the permutation obtained by exchanging $\pi^{\mathsf{r}}$ and $\pi^{\star_i}$ where $\star_i$ is the unique index satisfying $\pi^{\star_i} = i$. By applying Lemma C.4 with $\phi(\cdot) = f_i(\mathcal{A}_s(\cdot))$, there is

$$\mathbb{E}\left[f_i(\mathcal{A}_s(\pi))\mathbb{1}\left[\pi^{\mathsf{r}} = i\right]\right] = \frac{1}{n}\mathbb{E}\left[f_i(\mathcal{A}_s(\widehat{\pi}(\mathsf{r}, \star_i)))\right], \forall i \in [n],$$

which implies

$$\mathbb{E}\left[f_{\mathsf{i}(t)}(\boldsymbol{x}_s)\right] = \sum_{i=1}^{n} \mathbb{E}\left[f_i(\mathcal{A}_s(\pi))\mathbb{1}\left[\pi^{\mathsf{r}} = i\right]\right] = \frac{1}{n}\sum_{i=1}^{n} \mathbb{E}\left[f_i(\mathcal{A}_s(\widehat{\pi}(\mathsf{r}, \star_i)))\right].$$

Therefore,

$$
\begin{aligned}
|\mathbb{E}\left[\Omega_t(\boldsymbol{x}_s)\right]| &= \left|\frac{1}{n}\sum_{i=1}^{n} \mathbb{E}\left[f_i(\mathcal{A}_s(\widehat{\pi}(\mathsf{r}, \star_i))) - f_i(\mathcal{A}_s(\pi))\right]\right| \leq \frac{1}{n}\sum_{i=1}^{n} \mathbb{E}\left[|f_i(\mathcal{A}_s(\widehat{\pi}(\mathsf{r}, \star_i))) - f_i(\mathcal{A}_s(\pi))|\right] \\
&\overset{(a)}{\leq} \frac{1}{n}\sum_{i=1}^{n} \mathbb{E}\left[G_i \|\mathcal{A}_s(\widehat{\pi}(\mathsf{r}, \star_i)) - \mathcal{A}_s(\pi)\|\right] = \frac{1}{n}\sum_{i=1}^{n} G_i \mathbb{E}\left[\|\widehat{\boldsymbol{x}}_s(\mathsf{r}, \star_i) - \boldsymbol{x}_s\|\right],
\end{aligned}
$$

where $(a)$ is because $f_i$ is $G_i$-Lipschitz on $\mathrm{dom}\psi$ and $\widehat{\boldsymbol{x}}_s(\mathsf{r}, \star_i)$ is the output of running Algorithm 1 with the same initial point $\boldsymbol{x}_1$ and the stepsize $\eta_t, \forall t \in [T]$ under the SS sampling scheme but using the permutation $\widehat{\pi}(\mathsf{r}, \star_i)$, i.e.,

$$\widehat{\boldsymbol{x}}_1(\mathsf{r}, \star_i) \triangleq \boldsymbol{x}_1, \tag{62}$$

$$\widehat{\boldsymbol{x}}_{j+1}(\mathsf{r}, \star_i) \triangleq \mathrm{argmin}_{\boldsymbol{x} \in \mathbb{R}^d} \psi(\boldsymbol{x}) + \left\langle \nabla f_{\widehat{\pi}^{\mathsf{r}(j)}(\mathsf{r}, \star_i)}(\widehat{\boldsymbol{x}}_j(\mathsf{r}, \star_i)), \boldsymbol{x} \right\rangle + \frac{\|\boldsymbol{x} - \widehat{\boldsymbol{x}}_j(\mathsf{r}, \star_i)\|^2}{2\eta_j}, \forall j \in [s-1]. \tag{63}$$

Finally, by Lemma B.7, we have

$$
\begin{aligned}
\frac{1}{n}\sum_{i=1}^{n} G_i \mathbb{E}\left[\|\widehat{\boldsymbol{x}}_s(\mathsf{r}, \star_i) - \boldsymbol{x}_s\|\right] \leq & \frac{1}{n}\sum_{i=1}^{n} G_i \cdot 2\left(G_{f,1} + G_i\right)\sum_{j=1}^{s-1}\frac{\gamma_j\eta_j}{\gamma_s}\left(\mathbb{1}\left[\mathsf{r}(j) = \mathsf{r}\right] + \frac{\mathbb{1}\left[\mathsf{r}(j) \neq \mathsf{r}\right]}{n-1}\right) \\
& + \frac{1}{n}\sum_{i=1}^{n} G_i \cdot 2\sqrt{\sum_{j=1}^{s-1}\frac{\gamma_j^2\eta_j^2}{\gamma_s^2}\left(G_{f,2}^2 + G_i^2\mathbb{1}\left[\mathsf{r}(j) = \mathsf{r}\right] + \frac{nG_{f,2}^2 - G_i^2}{n-1}\mathbb{1}\left[\mathsf{r}(j) \neq \mathsf{r}\right]\right)} \\
= & 2\left(G_{f,1}^2 + G_{f,2}^2\right)\sum_{j=1}^{s-1}\frac{\gamma_j\eta_j}{\gamma_s}\left(\mathbb{1}\left[\mathsf{r}(j) = \mathsf{r}\right] + \frac{\mathbb{1}\left[\mathsf{r}(j) \neq \mathsf{r}\right]}{n-1}\right) \\
& + \frac{2}{n}\sum_{i=1}^{n} G_i \sqrt{\sum_{j=1}^{s-1}\frac{\gamma_j^2\eta_j^2}{\gamma_s^2}\left(G_{f,2}^2 + G_i^2\mathbb{1}\left[\mathsf{r}(j) = \mathsf{r}\right] + \frac{nG_{f,2}^2 - G_i^2}{n-1}\mathbb{1}\left[\mathsf{r}(j) \neq \mathsf{r}\right]\right)}.
\end{aligned}
$$

The proof is completed by using the fact $G_{f,1} \leq G_{f,2}$. $\qquad\square$

**Lemma B.7.** *Under the same settings in Lemma B.6, let $\widehat{\boldsymbol{x}}_s(\mathsf{r}, \star_i)$ be the point defined by (62) and (63), then we have*

$$\mathbb{E}\left[\|\boldsymbol{x}_s - \widehat{\boldsymbol{x}}_s(\mathsf{r}, \star_i)\|\right] \le 2\left(G_{f,1} + G_i\right) \sum_{j=1}^{s-1} \frac{\gamma_j \eta_j}{\gamma_s}\left(\mathbb{1}\left[\mathsf{r}(j) = \mathsf{r}\right] + \frac{\mathbb{1}\left[\mathsf{r}(j) \ne \mathsf{r}\right]}{n-1}\right)$$

$$+ 2\sqrt{\sum_{j=1}^{s-1} \frac{\gamma_j^2 \eta_j^2}{\gamma_s^2}\left(G_{f,2}^2 + G_i^2 \mathbb{1}\left[\mathsf{r}(j) = \mathsf{r}\right] + \frac{n G_{f,2}^2 - G_i^2}{n-1}\mathbb{1}\left[\mathsf{r}(j) \ne \mathsf{r}\right]\right)},$$

*where $\gamma_t$ is defined in Lemma B.2.*

*Proof.* For simplicity, we denote $\delta_s \triangleq \|\boldsymbol{x}_s - \widehat{\boldsymbol{x}}_s(\mathsf{r}, \star_i)\|, \forall s \in [t]$. Moreover, let $\widehat{\mathsf{i}}(s)$ represent the index trajectory generated under the permutation $\widehat{\pi}(\mathsf{r}, \star_i)$, i.e.,

$$\widehat{\mathsf{i}}(s) \triangleq \widehat{\pi}^{\mathsf{r}(s)}(\mathsf{r}, \star_i), \forall s \in [t-1]. \tag{64}$$

By Lemma C.2, we have

$$\delta_{s+1}^2 \le \frac{\delta_s^2 - 2\eta_s\left\langle \boldsymbol{x}_s - \widehat{\boldsymbol{x}}_s(\mathsf{r}, \star_i), \nabla f_{\mathsf{i}(s)}(\boldsymbol{x}_s) - \nabla f_{\widehat{\mathsf{i}}(s)}(\widehat{\boldsymbol{x}}_s(\mathsf{r}, \star_i))\right\rangle + \eta_s^2\left\|\nabla f_{\mathsf{i}(s)}(\boldsymbol{x}_s) - \nabla f_{\widehat{\mathsf{i}}(s)}(\widehat{\boldsymbol{x}}_s(\mathsf{r}, \star_i))\right\|^2}{(1 + \mu\eta_s)^2}$$

$$\overset{(a)}{\le} \frac{\delta_s^2 + 2\eta_s\left\|\nabla f_{\mathsf{i}(s)}(\boldsymbol{x}_s) - \nabla f_{\widehat{\mathsf{i}}(s)}(\boldsymbol{x}_s)\right\|\delta_s + \eta_s^2\left\|\nabla f_{\mathsf{i}(s)}(\boldsymbol{x}_s) - \nabla f_{\widehat{\mathsf{i}}(s)}(\widehat{\boldsymbol{x}}_s(\mathsf{r}, \star_i))\right\|^2}{(1 + \mu\eta_s)^2}$$

$$\overset{(b)}{\le} \frac{\delta_s^2 + 2\eta_s\left\|\nabla f_{\mathsf{i}(s)}(\boldsymbol{x}_s) - \nabla f_{\widehat{\mathsf{i}}(s)}(\boldsymbol{x}_s)\right\|\delta_s + 2\eta_s^2\left(G_{\mathsf{i}(s)}^2 + G_{\widehat{\mathsf{i}}(s)}^2\right)}{(1 + \mu\eta_s)^2}, \tag{65}$$

where $(a)$ is by

$$\left\langle \boldsymbol{x}_s - \widehat{\boldsymbol{x}}_s(\mathsf{r}, \star_i), \nabla f_{\mathsf{i}(s)}(\boldsymbol{x}_s) - \nabla f_{\widehat{\mathsf{i}}(s)}(\widehat{\boldsymbol{x}}_s(\mathsf{r}, \star_i))\right\rangle$$

$$= \left\langle \boldsymbol{x}_s - \widehat{\boldsymbol{x}}_s(\mathsf{r}, \star_i), \nabla f_{\widehat{\mathsf{i}}(s)}(\boldsymbol{x}_s) - \nabla f_{\widehat{\mathsf{i}}(s)}(\widehat{\boldsymbol{x}}_s(\mathsf{r}, \star_i))\right\rangle + \left\langle \boldsymbol{x}_s - \widehat{\boldsymbol{x}}_s(\mathsf{r}, \star_i), \nabla f_{\mathsf{i}(s)}(\boldsymbol{x}_s) - \nabla f_{\widehat{\mathsf{i}}(s)}(\boldsymbol{x}_s)\right\rangle$$

$$\overset{\text{Assumption } 2.1}{\ge} \left\langle \boldsymbol{x}_s - \widehat{\boldsymbol{x}}_s(\mathsf{r}, \star_i), \nabla f_{\mathsf{i}(s)}(\boldsymbol{x}_s) - \nabla f_{\widehat{\mathsf{i}}(s)}(\boldsymbol{x}_s)\right\rangle \ge -\left\|\nabla f_{\mathsf{i}(s)}(\boldsymbol{x}_s) - \nabla f_{\widehat{\mathsf{i}}(s)}(\boldsymbol{x}_s)\right\|\delta_s,$$

and $(b)$ holds due to

$$\left\|\nabla f_{\mathsf{i}(s)}(\boldsymbol{x}_s) - \nabla f_{\widehat{\mathsf{i}}(s)}(\widehat{\boldsymbol{x}}_s(\mathsf{r}, \star_i))\right\| \overset{\text{Assumption } 2.2}{\le} G_{\mathsf{i}(s)} + G_{\widehat{\mathsf{i}}(s)} \Rightarrow \left\|\nabla f_{\mathsf{i}(s)}(\boldsymbol{x}_s) - \nabla f_{\widehat{\mathsf{i}}(s)}(\widehat{\boldsymbol{x}}_s(\mathsf{r}, \star_i))\right\|^2 \le 2\left(G_{\mathsf{i}(s)}^2 + G_{\widehat{\mathsf{i}}(s)}^2\right).$$

Now recall that $\gamma_t = \prod_{s=1}^{t-1}(1 + \mu\eta_s), \forall t \in [T+1]$. Multiply both sides of (65) by $\gamma_{s+1}^2$ to obtain

$$\gamma_{s+1}^2 \delta_{s+1}^2 \le \gamma_s^2 \delta_s^2 + 2\gamma_s\eta_s\left\|\nabla f_{\mathsf{i}(s)}(\boldsymbol{x}_s) - \nabla f_{\widehat{\mathsf{i}}(s)}(\boldsymbol{x}_s)\right\|\gamma_s\delta_s + 2\gamma_s^2\eta_s^2\left(G_{\mathsf{i}(s)}^2 + G_{\widehat{\mathsf{i}}(s)}^2\right)$$

$$\Rightarrow \gamma_{s+1}^2 \delta_{s+1}^2 \le \gamma_1^2 \delta_1^2 + \sum_{j=1}^{s} 2\gamma_j\eta_j\left\|\nabla f_{\mathsf{i}(j)}(\boldsymbol{x}_j) - \nabla f_{\widehat{\mathsf{i}}(j)}(\boldsymbol{x}_j)\right\|\gamma_j\delta_j + 2\gamma_j^2\eta_j^2\left(G_{\mathsf{i}(j)}^2 + G_{\widehat{\mathsf{i}}(j)}^2\right)$$

$$= \sum_{j=1}^{s} 2\gamma_j\eta_j\left\|\nabla f_{\mathsf{i}(j)}(\boldsymbol{x}_j) - \nabla f_{\widehat{\mathsf{i}}(j)}(\boldsymbol{x}_j)\right\|\gamma_j\delta_j + 2\gamma_j^2\eta_j^2\left(G_{\mathsf{i}(j)}^2 + G_{\widehat{\mathsf{i}}(j)}^2\right), \tag{66}$$

where the last equation is by $\delta_1 = \|\boldsymbol{x}_1 - \widehat{\boldsymbol{x}}_1(\mathsf{r}, \star_i)\| \overset{(62)}{=} 0$.

Next, we use induction to prove

$$\gamma_s^2 \delta_s^2 \le 4\left(\sum_{j=1}^{s-1} \gamma_j\eta_j\left\|\nabla f_{\mathsf{i}(j)}(\boldsymbol{x}_j) - \nabla f_{\widehat{\mathsf{i}}(j)}(\boldsymbol{x}_j)\right\|\right)^2 + \sum_{j=1}^{s-1} 4\gamma_j^2\eta_j^2\left(G_{\mathsf{i}(j)}^2 + G_{\widehat{\mathsf{i}}(j)}^2\right), \forall s \in [t]. \tag{67}$$

For $s = 1$, (67) holds as $\delta_1 = 0$. Suppose (67) is true for all indices in $[s]$ where $s \in [t-1]$. Then for $s+1$,

- if $\gamma_{s+1}^2 \delta_{s+1}^2 \le \max_{k \in [s]} \gamma_k^2 \delta_k^2$, we know

$$\gamma_{s+1}^2 \delta_{s+1}^2 \le \max_{k \in [s]} \gamma_k^2 \delta_k^2$$

$$\overset{(67)}{\le} \max_{k \in [s]} \left[ 4 \left( \sum_{j=1}^{k-1} \gamma_j \eta_j \left\| \nabla f_{i(j)}(\boldsymbol{x}_j) - \nabla f_{\widehat{i}(j)}(\boldsymbol{x}_j) \right\| \right)^2 + \sum_{j=1}^{k-1} 4\gamma_j^2 \eta_j^2 \left( G_{i(j)}^2 + G_{\widehat{i}(j)}^2 \right) \right]$$

$$\le 4 \left( \sum_{j=1}^{s} \gamma_j \eta_j \left\| \nabla f_{i(j)}(\boldsymbol{x}_j) - \nabla f_{\widehat{i}(j)}(\boldsymbol{x}_j) \right\| \right)^2 + \sum_{j=1}^{s} 4\gamma_j^2 \eta_j^2 \left( G_{i(j)}^2 + G_{\widehat{i}(j)}^2 \right);$$

- if $\gamma_{s+1}^2 \delta_{s+1}^2 > \max_{k \in [s]} \gamma_k^2 \delta_k^2$, we know

$$\gamma_{s+1}^2 \delta_{s+1}^2 \overset{(66)}{\le} \sum_{j=1}^{s} 2\gamma_j \eta_j \left\| \nabla f_{i(j)}(\boldsymbol{x}_j) - \nabla f_{\widehat{i}(j)}(\boldsymbol{x}_j) \right\| \gamma_j \delta_j + 2\gamma_j^2 \eta_j^2 \left( G_{i(j)}^2 + G_{\widehat{i}(j)}^2 \right)$$

$$\le 2 \left( \sum_{j=1}^{s} \gamma_j \eta_j \left\| \nabla f_{i(j)}(\boldsymbol{x}_j) - \nabla f_{\widehat{i}(j)}(\boldsymbol{x}_j) \right\| \right) \gamma_{s+1} \delta_{s+1} + 2\gamma_j^2 \eta_j^2 \left( G_{i(j)}^2 + G_{\widehat{i}(j)}^2 \right)$$

$$\overset{(c)}{\le} \frac{\gamma_{s+1}^2 \delta_{s+1}^2}{2} + 2 \left( \sum_{j=1}^{s} \gamma_j \eta_j \left\| \nabla f_{i(j)}(\boldsymbol{x}_j) - \nabla f_{\widehat{i}(j)}(\boldsymbol{x}_j) \right\| \right)^2 + \sum_{j=1}^{s} 2\gamma_j^2 \eta_j^2 \left( G_{i(j)}^2 + G_{\widehat{i}(j)}^2 \right)$$

$$\Rightarrow \gamma_{s+1}^2 \delta_{s+1}^2 \le 4 \left( \sum_{j=1}^{s} \gamma_j \eta_j \left\| \nabla f_{i(j)}(\boldsymbol{x}_j) - \nabla f_{\widehat{i}(j)}(\boldsymbol{x}_j) \right\| \right)^2 + \sum_{j=1}^{s} 4\gamma_j^2 \eta_j^2 \left( G_{i(j)}^2 + G_{\widehat{i}(j)}^2 \right),$$

where $(c)$ is due to AM-GM inequality.

Therefore, we always have

$$\gamma_{s+1}^2 \delta_{s+1}^2 \le 4 \left( \sum_{j=1}^{s} \gamma_j \eta_j \left\| \nabla f_{i(j)}(\boldsymbol{x}_j) - \nabla f_{\widehat{i}(j)}(\boldsymbol{x}_j) \right\| \right)^2 + \sum_{j=1}^{s} 4\gamma_j^2 \eta_j^2 \left( G_{i(j)}^2 + G_{\widehat{i}(j)}^2 \right).$$

By induction, (67) holds for any $s \in [t]$, which implies

$$\gamma_s \delta_s \le 2 \sum_{j=1}^{s-1} \gamma_j \eta_j \left\| \nabla f_{i(j)}(\boldsymbol{x}_j) - \nabla f_{\widehat{i}(j)}(\boldsymbol{x}_j) \right\| + 2 \sqrt{\sum_{j=1}^{s-1} \gamma_j^2 \eta_j^2 \left( G_{i(j)}^2 + G_{\widehat{i}(j)}^2 \right)}$$

$$\Rightarrow \gamma_s \mathbb{E}\left[ \delta_s \right] \le 2 \underbrace{\mathbb{E}\left[ \sum_{j=1}^{s-1} \gamma_j \eta_j \left\| \nabla f_{i(j)}(\boldsymbol{x}_j) - \nabla f_{\widehat{i}(j)}(\boldsymbol{x}_j) \right\| \right]}_{\triangleq \bigcirc} + 2 \underbrace{\mathbb{E}\left[ \sqrt{\sum_{j=1}^{s-1} \gamma_j^2 \eta_j^2 \left( G_{i(j)}^2 + G_{\widehat{i}(j)}^2 \right)} \right]}_{\triangleq \bullet}. \tag{68}$$

- For term $\bigcirc$, note that if $\star_i = r$, then $\pi = \widehat{\pi}(r, \star_i) \Rightarrow \nabla f_{i(j)}(\boldsymbol{x}_j) = \nabla f_{\widehat{i}(j)}(\boldsymbol{x}_j), \forall j \in [s-1]$. So there is

$$\left\| \nabla f_{i(j)}(\boldsymbol{x}_j) - \nabla f_{\widehat{i}(j)}(\boldsymbol{x}_j) \right\| = \left\| \nabla f_{i(j)}(\boldsymbol{x}_j) - \nabla f_{\widehat{i}(j)}(\boldsymbol{x}_j) \right\| \mathbb{1}\left[ \star_i \ne r \right]$$

$$= \sum_{\ell \in [n] \setminus \{r\}} \left\| \nabla f_{i(j)}(\boldsymbol{x}_j) - \nabla f_{\widehat{i}(j)}(\boldsymbol{x}_j) \right\| \mathbb{1}\left[ \pi^\ell = i \right].$$

When $\pi^\ell = i$ for some $\ell \in [n] \setminus \{r\}$, we observe that $\widehat{i}(j) \overset{(64)}{=} \widehat{\pi}^{r(j)}(r, \ell) = \pi^{r(j)} = i(j)$ if $r(j) \neq r$ and $r(j) \neq \ell$, which implies

$$\left\| \nabla f_{i(j)}(\boldsymbol{x}_j) - \nabla f_{\widehat{i}(j)}(\boldsymbol{x}_j) \right\| \mathbb{1}\left[\pi^\ell = i\right] = \left\| \nabla f_{i(j)}(\boldsymbol{x}_j) - \nabla f_{\widehat{i}(j)}(\boldsymbol{x}_j) \right\| \mathbb{1}\left[\pi^\ell = i\right] \mathbb{1}\left[r(j) = r \text{ or } \ell\right]$$
$$= \left\| \nabla f_{\pi^r}(\boldsymbol{x}_j) - \nabla f_i(\boldsymbol{x}_j) \right\| \mathbb{1}\left[\pi^\ell = i\right] \mathbb{1}\left[r(j) = r \text{ or } \ell\right]$$
$$\leq (G_{\pi^r} + G_i)\, \mathbb{1}\left[\pi^\ell = i\right] \mathbb{1}\left[r(j) = r \text{ or } \ell\right],$$

where the second to last step is by $\left\{i(j), \widehat{i}(j)\right\} = \left\{\pi^{r(j)}, \widehat{\pi}^{r(j)}(r, \ell)\right\} = \{\pi^r, i\}$ under the events $\pi^\ell = i$ and $r(j) = r$ or $\ell$. Thus, for any $j \in [s-1]$,

$$\left\| \nabla f_{i(j)}(\boldsymbol{x}_j) - \nabla f_{\widehat{i}(j)}(\boldsymbol{x}_j) \right\| \leq \sum_{\ell \in [n] \setminus \{r\}} (G_{\pi^r} + G_i)\, \mathbb{1}\left[\pi^\ell = i\right] \mathbb{1}\left[r(j) = r \text{ or } \ell\right]$$

$$\Rightarrow \mathbb{E}\left[\left\| \nabla f_{i(j)}(\boldsymbol{x}_j) - \nabla f_{\widehat{i}(j)}(\boldsymbol{x}_j) \right\|\right] \leq \sum_{\ell \in [n] \setminus \{r\}} \mathbb{E}\left[(G_{\pi^r} + G_i)\, \mathbb{1}\left[\pi^\ell = i\right]\right] \mathbb{1}\left[r(j) = r \text{ or } \ell\right]$$

$$\overset{(d)}{=} \sum_{\ell \in [n] \setminus \{r\}} \left(\frac{nG_{f,1} - G_i}{n(n-1)} + \frac{G_i}{n}\right) \mathbb{1}\left[r(j) = r \text{ or } \ell\right]$$

$$= \left(G_{f,1} + \frac{n-2}{n}G_i\right)\left(\mathbb{1}\left[r(j) = r\right] + \frac{\mathbb{1}\left[r(j) \neq r\right]}{n-1}\right)$$

$$\leq (G_{f,1} + G_i)\left(\mathbb{1}\left[r(j) = r\right] + \frac{\mathbb{1}\left[r(j) \neq r\right]}{n-1}\right),$$

where $(d)$ is by, for any fixed $\ell \in [n] \setminus \{r\}$, there are

$$\mathbb{E}\left[G_{\pi^r}\mathbb{1}\left[\pi^\ell = i\right]\right] = \sum_{k \in [n] \setminus \{i\}} G_k \mathbb{P}\left[\pi^\ell = i, \pi^r = k\right] = \frac{1}{n(n-1)} \sum_{k \in [n] \setminus \{i\}} G_k = \frac{nG_{f,1} - G_i}{n(n-1)},$$

and

$$\mathbb{E}\left[G_i\mathbb{1}\left[\pi^\ell = i\right]\right] = \frac{G_i}{n}.$$

We thereby have

$$\bigcirc \leq (G_{f,1} + G_i) \sum_{j=1}^{s-1} \gamma_j \eta_j \left(\mathbb{1}\left[r(j) = r\right] + \frac{\mathbb{1}\left[r(j) \neq r\right]}{n-1}\right). \tag{69}$$

- For term $\bullet$, for any fixed $j \in [s-1]$, we claim the following three facts hold

$$i(j) \overset{\mathcal{D}}{=} \text{Uniform}\,[n]\,, \quad \widehat{i}(j) = 1 \text{ if } r(j) = r, \quad \widehat{i}(j) \overset{\mathcal{D}}{=} \text{Uniform}\,[n] \setminus \{i\} \text{ if } r(j) \neq r,$$

in which the first one follows by the definition of the SS sampling scheme, the second one is true by recalling $\widehat{i}(j) \overset{(64)}{=} \widehat{\pi}^{r(j)}(r, \star_i) = i$ if $r(j) = r$, and the third one is by (64) and Lemma C.5. Therefore, by Hölder's inequality

$$\bullet \leq \sqrt{\sum_{j=1}^{s-1} \gamma_j^2 \eta_j^2 \left(\mathbb{E}\left[G_{i(j)}^2\right] + \mathbb{E}\left[G_{\widehat{i}(j)}^2\right]\right)}$$

$$= \sqrt{\sum_{j=1}^{s-1} \gamma_j^2 \eta_j^2 \left(G_{f,2}^2 + G_i^2 \mathbb{1}\left[r(j) = r\right] + \frac{nG_{f,2}^2 - G_i^2}{n-1} \mathbb{1}\left[r(j) \neq r\right]\right)}. \tag{70}$$

Finally, we conclude by plugging (69) and (70) back into (68) and dividing both sides by $\gamma_s$. $\qquad\square$

## C. Auxiliary Lemmas

This section includes some technical results applied in the analysis presented in the previous sections.

We first provide two algebraic inequalities for the stepsize proportional to $\mathsf{q}(T) - \mathsf{q}(t) + 1$, which is used for the RR sampling scheme in Theorem A.1.

**Lemma C.1.** *Suppose* $\eta_t = \eta_\star(\mathsf{q}(T) - \mathsf{q}(t) + 1), \forall t \in [T]$ *where* $\eta_\star > 0$ *is a constant, then there are*

$$\sum_{t=1}^{T} \eta_t \geq \frac{\eta_\star \mathsf{q}(T) T}{2} \quad and \quad \sum_{t=1}^{T} \frac{\eta_t^2}{\sum_{s=t}^{T} \eta_s} \leq \frac{9\eta_\star(\mathsf{q}(T) + \log n)}{2}.$$

*Proof.* Note that for any $t \in [T]$, there is

$$\sum_{s=t}^{T} \eta_s = \sum_{s=(\mathsf{q}(T)-1)n+1}^{T} \eta_s + \sum_{s=\mathsf{q}(t)n+1}^{(\mathsf{q}(T)-1)n} \eta_s + \sum_{s=t}^{\mathsf{q}(t)n} \eta_s$$

$$= \eta_\star \left[ \mathsf{r}(T) + n \left( \sum_{j=\mathsf{q}(t)+1}^{\mathsf{q}(T)-1} \mathsf{q}(T) - j + 1 \right) + (n - \mathsf{r}(t) + 1)(\mathsf{q}(T) - \mathsf{q}(t) + 1) \right]$$

$$= \eta_\star \left[ \mathsf{r}(T) + \frac{n}{2}(\mathsf{q}(T) - \mathsf{q}(t) - 1)(\mathsf{q}(T) - \mathsf{q}(t) + 2) + (n - \mathsf{r}(t) + 1)(\mathsf{q}(T) - \mathsf{q}(t) + 1) \right]. \tag{71}$$

In particular, for $t = 1$,

$$\sum_{s=1}^{T} \eta_s = \eta_\star \left[ \mathsf{r}(T) + \frac{n}{2}(\mathsf{q}(T) - 2)(\mathsf{q}(T) + 1) + n\mathsf{q}(T) \right] = \eta_\star \left[ \frac{n}{2}\mathsf{q}(T)(\mathsf{q}(T) + 1) + \mathsf{r}(T) - n \right].$$

- If $\mathsf{q}(T) = 1$ (i.e., $T \in [n]$), we have

$$\sum_{s=1}^{T} \eta_s = \eta_\star \mathsf{r}(T) = \eta_\star \mathsf{q}(T) T \geq \frac{\eta_\star \mathsf{q}(T) T}{2}.$$

- If $\mathsf{q}(T) \geq 2$ (i.e., $T \geq n + 1$), we have

$$\sum_{s=1}^{T} \eta_s = \eta_\star \left[ \frac{n}{2}\mathsf{q}(T)(\mathsf{q}(T) + 1) + \mathsf{r}(T) - n \right] \overset{(a)}{\geq} \eta_\star \left[ \frac{n}{2}\mathsf{q}(T)(\frac{T}{n} + 1) + \mathsf{r}(T) - n \right]$$

$$= \eta_\star \left[ \frac{\mathsf{q}(T)T}{2} + \frac{\mathsf{q}(T)}{2}n + \mathsf{r}(T) - n \right] \overset{(b)}{\geq} \eta_\star \left[ \frac{\mathsf{q}(T)T}{2} + \mathsf{r}(T) \right] \geq \frac{\eta_\star \mathsf{q}(T)T}{2},$$

where $(a)$ is by $\mathsf{q}(T) \geq \frac{T}{n}$ and $(b)$ is due to $\mathsf{q}(T) \geq 2$ now.

Hence, we always have

$$\sum_{s=1}^{T} \eta_s \geq \frac{\eta_\star \mathsf{q}(T) T}{2}.$$

Next, we observe

$$\sum_{t=1}^{T} \frac{\eta_t^2}{\sum_{s=t}^{T} \eta_s} \overset{(71)}{=} \eta_\star \sum_{t=(\mathsf{q}(T)-1)n+1}^{T} \frac{1}{\mathsf{r}(T) - \mathsf{r}(t) + 1}$$

$$+ \eta_\star \sum_{k=1}^{\mathsf{q}(T)-1} \sum_{i=1}^{n} \frac{(\mathsf{q}(T) - k + 1)^2}{\mathsf{r}(T) + \frac{n}{2}(\mathsf{q}(T) - k - 1)(\mathsf{q}(T) - k + 2) + (n - i + 1)(\mathsf{q}(T) - k + 1)}. \tag{72}$$

- For the first part in (72), we have

$$\sum_{t=(\mathsf{q}(T)-1)n+1}^{T} \frac{1}{\mathsf{r}(T) - \mathsf{r}(t) + 1} = \sum_{i=1}^{\mathsf{r}(T)} \frac{1}{i} \leq 1 + \log \mathsf{r}(T).$$

- For the second part in (72), under relabeling the index, we have

$$\sum_{k=1}^{\mathsf{q}(T)-1} \sum_{i=1}^{n} \frac{(\mathsf{q}(T) - k + 1)^2}{\mathsf{r}(T) + \frac{n}{2}(\mathsf{q}(T) - k - 1)(\mathsf{q}(T) - k + 2) + (n - i + 1)(\mathsf{q}(T) - k + 1)}$$

$$= \sum_{k=2}^{\mathsf{q}(T)} \sum_{i=1}^{n} \frac{k^2}{\mathsf{r}(T) + \frac{n}{2}(k - 2)(k + 1) + ik} \leq \sum_{i=1}^{n} \frac{4}{\mathsf{r}(T) + 2i} + \sum_{k=3}^{\mathsf{q}(T)} \frac{2k^2}{(k - 2)(k + 1)}$$

$$\leq 2 \log \left(1 + \frac{2n}{\mathsf{r}(T)}\right) + \frac{9}{2}(\mathsf{q}(T) - 2) \leq 2 \log \left(1 + \frac{2n}{\mathsf{r}(T)}\right) + \frac{9}{2}\mathsf{q}(T) - \frac{9}{2}.$$

So there is

$$\sum_{t=1}^{T} \frac{\eta_t^2}{\sum_{s=t}^{T} \eta_s} \leq \eta_\star \left[\log \mathsf{r}(T) + 2 \log \left(1 + \frac{2n}{\mathsf{r}(T)}\right) + \frac{9}{2}\mathsf{q}(T) - \frac{7}{2}\right]$$

$$\leq \eta_\star \left[2 \log(\mathsf{r}(T) + 2n) + \frac{9}{2}\mathsf{q}(T) - \frac{7}{2}\right]$$

$$\leq \eta_\star \left[2 \log n + \frac{9}{2}\mathsf{q}(T) + 2 \log 3 - \frac{7}{2}\right]$$

$$\leq \frac{9\eta_\star(\mathsf{q}(T) + \log n)}{2}.$$

$\square$

Next, we introduce Lemma C.2, which gives a general upper bound on the distance between two points output by different proximal updates but using the same stepsize and plays a key role in bounding $|\mathbb{E}\left[\Omega_t(\boldsymbol{x}_s)\right]|, \forall t \in [T], s \in [t]$.

**Lemma C.2.** *Under Assumption 2.1, given $\bar{\boldsymbol{x}}, \bar{\boldsymbol{y}}, \boldsymbol{g}_{\bar{\boldsymbol{x}}}, \boldsymbol{g}_{\bar{\boldsymbol{y}}} \in \mathbb{R}^d$ and $\eta > 0$, let*

$$\tilde{\boldsymbol{x}} \triangleq \operatorname{argmin}_{\boldsymbol{x} \in \mathbb{R}^d} \psi(\boldsymbol{x}) + \langle \boldsymbol{g}_{\bar{\boldsymbol{x}}}, \boldsymbol{x} \rangle + \frac{\|\boldsymbol{x} - \bar{\boldsymbol{x}}\|^2}{2\eta},$$

$$\tilde{\boldsymbol{y}} \triangleq \operatorname{argmin}_{\boldsymbol{y} \in \mathbb{R}^d} \psi(\boldsymbol{y}) + \langle \boldsymbol{g}_{\bar{\boldsymbol{y}}}, \boldsymbol{y} \rangle + \frac{\|\boldsymbol{y} - \bar{\boldsymbol{y}}\|^2}{2\eta},$$

*then there is*

$$\|\tilde{\boldsymbol{x}} - \tilde{\boldsymbol{y}}\| \leq \frac{\|\bar{\boldsymbol{x}} - \bar{\boldsymbol{y}} - \eta(\boldsymbol{g}_{\bar{\boldsymbol{x}}} - \boldsymbol{g}_{\bar{\boldsymbol{y}}})\|}{1 + \mu\eta}.$$

*Proof.* By the definition of $\tilde{\boldsymbol{x}}$, there exists $\nabla\psi(\tilde{\boldsymbol{x}}) \in \partial\psi(\tilde{\boldsymbol{x}})$ such that $\boldsymbol{0} = \nabla\psi(\tilde{\boldsymbol{x}}) + \boldsymbol{g}_{\bar{\boldsymbol{x}}} + \frac{\tilde{\boldsymbol{x}} - \bar{\boldsymbol{x}}}{\eta}$, which implies

$$\langle \eta\nabla\psi(\tilde{\boldsymbol{x}}) + \tilde{\boldsymbol{x}}, \tilde{\boldsymbol{x}} - \tilde{\boldsymbol{y}} \rangle = \langle \bar{\boldsymbol{x}} - \eta\boldsymbol{g}_{\bar{\boldsymbol{x}}}, \tilde{\boldsymbol{x}} - \tilde{\boldsymbol{y}} \rangle.$$

Similarly, we have

$$\langle \eta\nabla\psi(\tilde{\boldsymbol{y}}) + \tilde{\boldsymbol{y}}, \tilde{\boldsymbol{y}} - \tilde{\boldsymbol{x}} \rangle = \langle \bar{\boldsymbol{y}} - \eta\boldsymbol{g}_{\bar{\boldsymbol{y}}}, \tilde{\boldsymbol{y}} - \tilde{\boldsymbol{x}} \rangle.$$

Sum up the above two equations to obtain

$$\eta \langle \nabla\psi(\tilde{\boldsymbol{x}}) - \nabla\psi(\tilde{\boldsymbol{y}}), \tilde{\boldsymbol{x}} - \tilde{\boldsymbol{y}} \rangle + \|\tilde{\boldsymbol{x}} - \tilde{\boldsymbol{y}}\|^2 = \langle \bar{\boldsymbol{x}} - \bar{\boldsymbol{y}} - \eta(\boldsymbol{g}_{\bar{\boldsymbol{x}}} - \boldsymbol{g}_{\bar{\boldsymbol{y}}}), \tilde{\boldsymbol{x}} - \tilde{\boldsymbol{y}} \rangle.$$

Note that Assumption 2.1 implies
$$\langle \nabla \psi(\tilde{\boldsymbol{x}}) - \nabla \psi(\tilde{\boldsymbol{y}}), \tilde{\boldsymbol{x}} - \tilde{\boldsymbol{y}} \rangle \geq \mu \|\tilde{\boldsymbol{x}} - \tilde{\boldsymbol{y}}\|^2.$$

Hence, there is
$$\|\tilde{\boldsymbol{x}} - \tilde{\boldsymbol{y}}\|^2 \leq \frac{\langle \bar{\boldsymbol{x}} - \bar{\boldsymbol{y}} - \eta(\boldsymbol{g}_{\bar{\boldsymbol{x}}} - \boldsymbol{g}_{\bar{\boldsymbol{y}}}), \tilde{\boldsymbol{x}} - \tilde{\boldsymbol{y}} \rangle}{1 + \mu\eta} \Rightarrow \|\tilde{\boldsymbol{x}} - \tilde{\boldsymbol{y}}\| \leq \frac{\|\bar{\boldsymbol{x}} - \bar{\boldsymbol{y}} - \eta(\boldsymbol{g}_{\bar{\boldsymbol{x}}} - \boldsymbol{g}_{\bar{\boldsymbol{y}}})\|}{1 + \mu\eta}.$$

$\square$

Finally, we provide some useful facts related to the random permutation inspired by (Sherman et al., 2021; Koren et al., 2022). We recall that $S_n$ is the symmetric group of $[n]$, i.e., the set containing all permutations of $[n]$.

**Lemma C.3.** *Suppose $\pi = (\pi^1, \cdots, \pi^n)$ is uniformly distributed on $S_n$, for any $\mathsf{r} \in [n]$ and $i \in [\mathsf{r} - 1]$, we define $\widehat{\pi}(\mathsf{r}, i)$ by exchanging $\pi^{\mathsf{r}}$ and $\pi^i$ in $\pi$, then there is*
$$\pi \overset{\mathcal{D}}{=} \widehat{\pi}(\mathsf{r}, i).$$

*Proof.* It is enough to prove that, for any fixed $o \in S_n$, there is
$$\mathbb{P}\left[\pi = o\right] = \mathbb{P}\left[\widehat{\pi}(\mathsf{r}, i) = o\right],$$

which clearly holds as both sides equal $\frac{1}{n!}$. $\square$

**Lemma C.4.** *Suppose $\pi = (\pi^1, \cdots, \pi^n)$ is uniformly distributed on $S_n$, for any $\mathsf{r}, i \in [n]$, we define $\widehat{\pi}(\mathsf{r}, \star_i)$ by exchanging $\pi^{\mathsf{r}}$ and $\pi^{\star_i}$ in $\pi$ where $\star_i$ is the unique index satisfying $\pi^{\star_i} = i$, then there is*
$$\mathbb{E}\left[\phi(\pi)\mathbb{1}\left[\pi^{\mathsf{r}} = i\right]\right] = \frac{1}{n}\mathbb{E}\left[\phi(\widehat{\pi}(\mathsf{r}, \star_i))\right],$$

*where $\phi : S_n \to \mathbb{R}$ can ba any deterministic map.*

*Proof.* We observe that

$$\mathbb{E}\left[\phi(\widehat{\pi}(\mathsf{r}, \star_i))\right] = \mathbb{E}\left[\sum_{j=1}^n \phi(\widehat{\pi}(\mathsf{r}, \star_i))\mathbb{1}\left[\star_i = j\right]\right] = \mathbb{E}\left[\sum_{j=1}^n \phi(\widehat{\pi}(\mathsf{r}, j))\mathbb{1}\left[\pi^j = i\right]\right] = \sum_{j=1}^n \mathbb{E}\left[\phi(\widehat{\pi}(\mathsf{r}, j))\mathbb{1}\left[\pi^j = i\right]\right]. \quad (73)$$

For any fixed $j \in [n]$, let $\mathcal{T}_{\mathsf{r}\leftrightarrow j} : S_n \to S_n$ be the operation of exchanging the value of $o^{\mathsf{r}}$ and $o^j$ for $o \in S_n$. Then we know

$$\mathbb{E}\left[\phi(\widehat{\pi}(\mathsf{r}, j))\mathbb{1}\left[\pi^j = i\right]\right] = \frac{1}{n!}\sum_{o \in S_n} \phi(\mathcal{T}_{\mathsf{r}\leftrightarrow j}(o))\mathbb{1}\left[o^j = i\right] = \frac{1}{n!}\sum_{o \in S_n} \phi(\mathcal{T}_{\mathsf{r}\leftrightarrow j}(o))\mathbb{1}\left[(\mathcal{T}_{\mathsf{r}\leftrightarrow j}(o))^{\mathsf{r}} = i\right]$$
$$= \frac{1}{n!}\sum_{o \in S_n} \phi(o)\mathbb{1}\left[o^{\mathsf{r}} = i\right] = \mathbb{E}\left[\phi(\pi)\mathbb{1}\left[\pi^{\mathsf{r}} = i\right]\right], \quad (74)$$

where the second to last step is by $\{\mathcal{T}_{\mathsf{r}\leftrightarrow j}(o) : o \in S_n\} = S_n$. Combine (73) and (74) to finally obtain

$$\mathbb{E}\left[\phi(\pi)\mathbb{1}\left[\pi^{\mathsf{r}} = i\right]\right] = \frac{1}{n}\sum_{j=1}^n \mathbb{E}\left[\phi(\widehat{\pi}(\mathsf{r}, j))\mathbb{1}\left[\pi^j = i\right]\right] = \frac{1}{n}\mathbb{E}\left[\phi(\widehat{\pi}(\mathsf{r}, \star_i))\right].$$

$\square$

**Lemma C.5.** *Suppose $\pi = (\pi^1, \cdots, \pi^n)$ is uniformly distributed on $S_n$, for any $\mathsf{r}, i \in [n]$, we define $\widehat{\pi}(\mathsf{r}, \star_i)$ by exchanging $\pi^{\mathsf{r}}$ and $\pi^{\star_i}$ in $\pi$ where $\star_i$ is the unique index satisfying $\pi^{\star_i} = i$, then there is*
$$\widehat{\pi}^k(\mathsf{r}, \star_i) \overset{\mathcal{D}}{=} \mathrm{Uniform}\left[n\right] \setminus \{i\}, \forall k \in [n], k \neq \mathsf{r}.$$

*Proof.* Given $k \in [n]$ and $k \neq \mathsf{r}$, let $j \in [n]$ be fixed. We define the function $\phi(\pi) \triangleq \mathbb{1}\left[\pi^k = j\right]$ and notice that

$$\mathbb{P}\left[\widehat{\pi}^k(\mathsf{r}, \star_i) = j\right] = \mathbb{E}\left[\phi(\widehat{\pi}(\mathsf{r}, \star_i))\right] \overset{\text{Lemma C.4}}{=} n\mathbb{E}\left[\phi(\pi)\mathbb{1}\left[\pi^{\mathsf{r}} = i\right]\right] = n\mathbb{P}\left[\pi^{\mathsf{r}} = i, \pi^k = j\right] = \begin{cases} 0 & j = i \\ \frac{1}{n-1} & j \neq i \end{cases},$$

which concludes the result. $\square$

## D. Lower Bound for Strongly Convex $\psi$

We present a lower bound that can be applied to the first-order algorithm containing a proximal update.

Given $F = f + \psi : \mathbb{R}^d \to \overline{\mathbb{R}}$ satisfying Assumptions 2.1 and 2.2 and an initial point $\boldsymbol{x}_1 \in \mathbb{R}^d$, we consider a family of algorithms obeying the following update rules in a total of $T$ iterations,

$$\boldsymbol{y}_{t+1} \in \boldsymbol{x}_1 + \text{Span} \cup_{s \in [t]} \{\boldsymbol{x}_s - \boldsymbol{x}_1, \boldsymbol{y}_s - \boldsymbol{x}_1, \nabla f(\boldsymbol{x}_s), \nabla f(\boldsymbol{y}_s)\}, \tag{75}$$

$$\boldsymbol{x}_{t+1} = \text{argmin}_{\boldsymbol{x} \in \mathbb{R}^d} \psi(\boldsymbol{x}) + \frac{\|\boldsymbol{x} - \boldsymbol{y}_{t+1}\|^2}{2\gamma_t}, \tag{76}$$

where $\boldsymbol{y}_1 = \boldsymbol{x}_1$, $\nabla f(\boldsymbol{z}_t) \in \partial f(\boldsymbol{z}), \forall t \in [T]$ for $\boldsymbol{z} \in \{\boldsymbol{x}, \boldsymbol{y}\}$, and $\gamma_t, \forall t \in [T]$ is a positive sequence. Note that (75) can be viewed as a generalization of the existing span assumption (Nesterov et al., 2018), as it contains more information based on the output of the proximal update in (76). In particular, (75) and (76) recover Proximal GD with the stepsize sequence $\eta_t, \forall t \in [T]$ by setting $\boldsymbol{y}_{t+1} = \boldsymbol{x}_1 + \boldsymbol{x}_t - \boldsymbol{x}_1 - \eta_t \nabla f(\boldsymbol{x}_t) = \boldsymbol{x}_t - \eta_t \nabla f(\boldsymbol{x}_t)$ and $\gamma_t = \eta_t$.

Now we are ready to prove the lower bound. As mentioned in Footnote 1, the proof is only a simple variation of the existing analysis in (Bubeck et al., 2015).

**Theorem D.1.** *For any given $D_\star > 0$, $G > 0$, $\mu > 0$, $T \in \mathbb{N}$ satisfying $T \geq \frac{G^2}{\mu^2 D_\star^2} - 1$, $d \in \mathbb{N}$ satisfying $d \geq T + 1$, and $\boldsymbol{x}_1 \in \mathbb{R}^d$, there exist a function $F = f + \psi : \mathbb{R}^d \to \mathbb{R}$ where $f$ is convex and $G$-Lipschitz on $\mathbb{R}^d$ and $\psi$ is $\mu$-strongly convex on $\mathbb{R}^d$ and a subgradient oracle $\nabla f$ such that any algorithm in the form of (75) and (76) starting with $\boldsymbol{x}_1$ has*

$$\min_{t \in [T]} F(\boldsymbol{x}_{t+1}) - F(\boldsymbol{x}_\star) \geq \frac{G^2}{2\mu(T+1)},$$

*where $\boldsymbol{x}_\star = \text{argmin}_{\boldsymbol{x} \in \mathbb{R}^d} F(\boldsymbol{x})$ satisfying $\|\boldsymbol{x}_\star - \boldsymbol{x}_1\| \leq D_\star$.*

*Proof.* W.l.og., we assume $\boldsymbol{x}_1 = \boldsymbol{0}$. For a general point $\boldsymbol{x}_1 \in \mathbb{R}^d$, one can change every $\boldsymbol{x}$ to $\boldsymbol{x} - \boldsymbol{x}_1$ in the following definition of $f$ and $\psi$ and then conclude by similar steps.

The construction of the hard instance is essentially the same as (Bubeck et al., 2015). Let $f(\boldsymbol{x}) \triangleq G \max_{j \in [T+1]} \boldsymbol{x}[j]$ and $\psi(\boldsymbol{x}) \triangleq \frac{\mu}{2} \|\boldsymbol{x}\|^2$, where $\boldsymbol{x}[j]$ is the $j$-th coordinate of $\boldsymbol{x}$. Note that

$$\partial f(\boldsymbol{x}) = G \cdot \text{Conv} \left\{ \boldsymbol{e}_j : j \in \text{argmax}_{i \in [T+1]} \boldsymbol{x}[i] \right\},$$

where $\boldsymbol{e}_j \in \mathbb{R}^d$ denotes the vector that takes 1 in the $j$-th coordinate and 0 in any other place. So $f$ is $G$-Lipschitz on $\mathbb{R}^d$. $\psi$ is $\mu$-strongly convex on $\mathbb{R}^d$ from its definition.

Next, we claim the minimum value $F(\boldsymbol{x}_\star) = -\frac{G^2}{2\mu(T+1)}$ where $\boldsymbol{x}_\star$ satisfies

$$\boldsymbol{x}_\star[j] = \begin{cases} -\frac{G}{\mu(T+1)} & j \in [T+1], \\ 0 & j \in [d] \setminus [T+1]. \end{cases}$$

We consider two cases:

- $\max_{j \in [T+1]} \boldsymbol{x}[j] \geq 0$. We observe $\boldsymbol{0}$ falls into this case and note that $F(\boldsymbol{x}) \geq \psi(\boldsymbol{x}) \geq 0 = F(\boldsymbol{0})$.

- $\max_{j \in [T+1]} \boldsymbol{x}[j] < 0$. We observe $\boldsymbol{x}_\star$ falls into this case. Now suppose $\max_{j \in [T+1]} \boldsymbol{x}[j] = -a$ for some $a > 0$, we have $|\boldsymbol{x}[j]| \geq a, \forall j \in [T+1]$. Thus, there is

$$F(\boldsymbol{x}) = -Ga + \frac{\mu}{2} \sum_{j=1}^d |\boldsymbol{x}[j]|^2 = -Ga + \frac{\mu}{2} \sum_{j=1}^{T+1} |\boldsymbol{x}[j]|^2 + \frac{\mu}{2} \sum_{j=T+2}^d |\boldsymbol{x}[j]|^2$$

$$\geq -Ga + \frac{\mu(T+1)a^2}{2} \geq -\frac{G^2}{2\mu(T+1)} = F(\boldsymbol{x}_\star).$$

We thereby have

$$\min_{\boldsymbol{x}\in\mathbb{R}^d} F(\boldsymbol{x}) = \min\{F(\boldsymbol{0}), F(\boldsymbol{x}_\star)\} = F(\boldsymbol{x}_\star) = -\frac{G^2}{2\mu(T+1)}. \tag{77}$$

Moreover, $\|\boldsymbol{x}_\star\| = \frac{G}{\mu\sqrt{T+1}} \le D_\star$ once $T \ge \frac{G^2}{\mu^2 D_\star^2} - 1$.

Because $\boldsymbol{x}_1 = \boldsymbol{0}$ now, we have $\boldsymbol{y}_{t+1} \in \text{Span} \cup_{s\in[t]} \{\boldsymbol{x}_s, \boldsymbol{y}_s, \nabla f(\boldsymbol{x}_s), \nabla f(\boldsymbol{y}_s)\}$. Moreover, we can explicitly write

$$\boldsymbol{x}_{t+1} = \text{argmin}_{\boldsymbol{x}\in\mathbb{R}^d} \psi(\boldsymbol{x}) + \frac{\|\boldsymbol{x} - \boldsymbol{y}_{t+1}\|^2}{2\gamma_t} = \text{argmin}_{\boldsymbol{x}\in\mathbb{R}^d} \frac{\mu\|\boldsymbol{x}\|^2}{2} + \frac{\|\boldsymbol{x} - \boldsymbol{y}_{t+1}\|^2}{2\gamma_t}$$

$$= \frac{\boldsymbol{y}_{t+1}}{1+\mu\gamma_t} \in \text{Span} \cup_{s\in[t]} \{\boldsymbol{x}_s, \boldsymbol{y}_s, \nabla f(\boldsymbol{x}_s), \nabla f(\boldsymbol{y}_s)\}.$$

Now we define the subgradient oracle

$$\nabla f(\boldsymbol{x}) \triangleq \boldsymbol{e}_{j_{\boldsymbol{x}}} \text{ where } j_{\boldsymbol{x}} \triangleq \min \text{argmax}_{i\in[T+1]} \boldsymbol{x}\,[i].$$

By induction, one can show $\text{Span} \cup_{s\in[t]} \{\boldsymbol{x}_s, \boldsymbol{y}_s, \nabla f(\boldsymbol{x}_s), \nabla f(\boldsymbol{y}_s)\} \subseteq \text{Span}\{\boldsymbol{e}_1, \cdots, \boldsymbol{e}_t\}, \forall t \in [T]$. As such, $\boldsymbol{x}_{t+1} \in \text{Span}\{\boldsymbol{e}_1, \cdots, \boldsymbol{e}_t\}, \forall t \in [T]$, which implies $F(\boldsymbol{x}_{t+1}) \ge f(\boldsymbol{x}_{t+1}) \ge G\boldsymbol{x}_{t+1}\,[t+1] = 0, \forall t \in [T]$. We hence conclude

$$\min_{t\in[T]} F(\boldsymbol{x}_{t+1}) - F(\boldsymbol{x}_\star) \overset{(77)}{\ge} \frac{G^2}{2\mu(T+1)}.$$

$\square$

