# OpenReview forum: "Improved Last-Iterate Convergence of Shuffling Gradient Methods for Nonsmooth Convex Optimization"
_ICML.cc/2025/Conference — ICML 2025 poster_

### Official Review · Reviewer_P57t · 2025-03-10

**Overall Recommendation:** 4

**Summary:**

This work studies last-iterate guarantees under different shuffling models. In the RR model, the authors establish last-iterate (and suffix-averaging) guarantees, similar to the iterate-averaging results of (Koren et al., 2022), with slightly improved rates in the SS model. The obtained guarantees generally match or improve upon prior work on last-iterate guarantees in shuffling models (Liu & Zhou, 2024b). The results apply to both convex and strongly convex functions and support a broader class of $\psi$ functions compared to $I_c$ in (Koren et al., 2022).

## update after rebuttal

I thank the authors for their response and continue to support the acceptance of this work.

**Claims And Evidence:**

The claims are supported by a thorough theoretical analysis across multiple shuffling models.

**Essential References Not Discussed:**

Essential references are discussed.

**Experimental Designs Or Analyses:**

N/A

**Methods And Evaluation Criteria:**

N/A

**Other Comments Or Suggestions:**

Overall, the paper presents a strong contribution to an important topic, and the reviewer recommends its acceptance.

**Other Strengths And Weaknesses:**

**Strengths:**

- The analysis of shuffling schemes and last-iterate guarantees represents a valuable contribution to stochastic optimization, better reflecting real-world practice compared to sampling with replacement and standard iterate averaging.
- The paper considers a broad range of settings, including convex, strongly convex, and general $\psi$ functions.

**Weaknesses:**

- Some gaps remain compared to the best-known lower bounds from (Koren et al., 2022). Specifically, it is unclear whether the last-iterate guarantee is inherently worse than iterate averaging, given that the best-known lower bound when averaging across all epochs is $n^{-1/4} K^{-3/4}$. That said, the theoretical contributions of the paper remain solid even without resolving this issue.

**Questions For Authors:**

- In your view, what is the main challenge in closing the gap to the $n^{-1/4} K^{-3/4}$ lower bound established in (Koren et al., 2022)? Is the last iterate inherently worse, or should the lower bound for averaging across multiple epochs be improved?

**Relation To Broader Scientific Literature:**

The results greatly improve upon (Liu & Zhou, 2024b).
The technique builds heavily on (Liu & Zhou, 2024a) and, to a lesser extent, (Koren et al., 2022). However, the considered setting is more general, and Algorithm 1 aims to accommodate a broader range of shuffling schemes. This generality is reflected in Lemma 5.1, the use of $\Phi$ on page 8, and the corresponding analysis.

**Theoretical Claims:**

Since the paper presents numerous technical results, most theoretical claims are established in the appendix, which is extensive. The reviewer primarily focused on Lemma 5.1 (Lemmas B.2 and B.3 in the appendix), which appear correct, and partially examined Lemma B.4 (RR scheme).

---

> ### Author Rebuttal · Authors · 2025-03-30
>
> We appreciate the reviewer's positive feedback. We will answer the reviewer's question below.
>
> **Question.** Thanks for the deep question. We will discuss it from the following two perspectives.
>
>   -  As mentioned in our Subsection 1.2 (see Lines 122-142, right column), the lower bound in [1] can be read as $\Omega\left(\frac{1}{J^{1/4}n^{1/4}\sqrt{K}}+\frac{1}{\sqrt{nK}}\right)$ for the suffix average of the last $J$ epohs, i.e., $\frac{1}{Jn}\sum_{j=K-J}^{K-1}\sum_{i=1}^{n}\boldsymbol{x}_{jn+i+1}$. Therefore, for the average over all $K$ epochs, the lower bound is $\Omega\left(\frac{1}{n^{1/4}K^{3/4}}+\frac{1}{\sqrt{nK}}\right)$. In other words, the rate $\Omega\left(\frac{1}{n^{1/4}K^{3/4}}\right)$ for the average iterate could only be possible in the small epoch regime, i.e., $K\leq n$.
>
>   - To be honest, we have no idea whether this rate is tight. If it is indeed achievable, then it means that there exists at least one problem instance such that the average iterate can improve over the last iterate by a factor of  $K^{-1/4}$, which is highly surprising and even seems impossible in our opinion since we have never seen an optimization method exibithing such a property if the stepsize is fine picked. As such, we suspect the lower bound for the average iterate should be improved.
>
>
> **References**
>
> [1] Koren, Tomer, et al. "Benign underfitting of stochastic gradient descent." Advances in Neural Information Processing Systems 35 (2022): 19605-19617.

---

### Official Review · Reviewer_4WZ4 · 2025-03-13

**Overall Recommendation:** 4

**Summary:**

This paper studies the Last iterate convergence of proximal gradient methods for non-smooth (strongly) convex optimisation problem, with Random Reshuffle (RR) and Single Shuffle (SS) strategies. The paper considers the General Proximal Gradient Method, where proximal step is implemented at every step which is more natural than most recent works' variant (e.g. Liu & Zhou 2024) where the proximal step is implemented at every epoch. Further, the paper establishes an array of convergence results for RR and SS in general convex and strongly convex setting. For RR, the rates in the paper are always better than the SOTA by some polynomial factors in $n$, the number of individual functions. For SS, the algorithm might not converge when $n$ is small, but when the composite part $\psi$ becomes the characteristic function over some feasible sets, the paper established some more refined convergence rate that improves over the SOTA.

## Update after rebuttal
I maintain that this is a very good paper and should be accepted. I will keep the score 4.

**Claims And Evidence:**

The paper is theoretical in nature and I discuss the claims and evidence in the Theoretical Claims section.

**Essential References Not Discussed:**

There are no essential references that the paper omits.

**Experimental Designs Or Analyses:**

not applicable.

**Methods And Evaluation Criteria:**

The methods and evaluation criteria is valid.

**Other Comments Or Suggestions:**

- Table 1 looks confusing and I suggest that the authors clearly mark the upper bounds and lower bounds.

**Other Strengths And Weaknesses:**

see the next section.

**Questions For Authors:**

There are a few questions that I would like to ask the authors about:

- Regarding last iterate convergence: as the authors pointed out, the last-iterate convergence in Liu & Zhou (2024) is obtained following the technique of Zamani & Clineur (2023). While the proofs in this paper seems to be substantially different from that of L&Z, I wonder if the authors are still following the ideas of Z&C to obtain the last-iterate convergence guarantees? As noted in [1], the techniques in Z&C gives rise to stepsize schedulers that are, in some sense, separate from some backbone algorithms with only avg convergence guarantees. If the last-iterate convergence of this work also somewhat follows the ideas of Z&C, I wonder if it's possible that the observations in [1] can be applied to the results here as well? Is it possible that some (hopefully) simpler descent analysis for the avg iterate can be presented, and the scheduler part can then be added on top separately to obtain the last-iterate convergence?

- Some further discussions on the source of improvement (or indeed, deterioration, in some cases) of the convergence rates: On one hand, the algorithm considered in this paper is somewhat more natural than the one in L&Z. On the other hand, the paper also presented different analysis techniques than the previous works, where the authors included in their considerations the randomness of RR and SS in each epoch. I wonder if the authors could comment on whether the improvement in the convergence rate for RR comes purely from this new insight in the analysis, or is the the fact that now proximal operator is applied at each step is also important for obtaining the improvement for RR? Similarly, can the authors comment on why, in the general case, is the convergence results for SS worse than that of L&Z? Is it purely a deficiency in the analysis (and it's absolutely fine if the authors have no idea on how to resolve it), or is the difference in the algorithm causing the difficulties in the analysis? Despite the fact that the algorithms in this paper and L&Z become the same when no proximal step is taken, can the authors also discuss why the results in Theorem 4.5 can be improved in Theorem 4.6 under the additional assumption? Is it because some difficulties with the every-step proximal operator can be overcame when the proximal operator is simple enough (suggesting that perhaps the difficulties might come from the proximal operator in Theorem 4.5), or there are some further insights in the analysis?


[1] Defazio, Aaron, et al. "Optimal linear decay learning rate schedules and further refinements." arXiv preprint arXiv:2310.07831 (2023).

**Relation To Broader Scientific Literature:**

The earlier Liu & Zho (2024) work gave a series of last-iterate convergence results of shuffling proximal gradient methods in various settings, some matching known lower bounds. However, their rates in the nonsmooth Lipschitz continuous setting did not demonstrate any improvements over the simple proximal (full) gradient method. This work follows up on the work of Liu & Zhou (2024) in the nonsmooth Lipschitz continuous setting in a spectacular way, proving improved convergence rates for RR in both the convex and strongly convex setting. The results for SS are however worse than that of Liu & Zhou (2024), failing to show convergence in some settings. But with some additional assumptions, the authors managed to improve the rates of Liu & Zhou even for SS.

On the algorithm side, the method studied in the paper differs from most of the recent works, where the proximal operator is applied at every step, instead of at every epoch. The epoch-wise proximal operator implicitly treats the shuffling method as a way to accumulate some approximate of the full gradients. The method considered in this paper is therefore in my opinion much more natural, and principled.

I believe that this paper makes an important contribution towards of the field of shuffling gradient method.

**Theoretical Claims:**

The paper gives a number of convergence theorems, in particular, Theorem 4.2, Corollary 4.3, Theorem 4.4, Theorem 4.5, Theorem 4.6, and Theorem 4.7, all of which are backed by proofs in the appendices.

At the core of the these convergence rates, the authors gave a general last-iterate descent analysis in Lemma 5.1. I briefly went through Appendix B.1 which gives the proof for Lemma 5.1. While I didn't check the analysis therein line-by-line, the techniques seem solid to me.

---

> ### Author Rebuttal · Authors · 2025-03-30
>
> We thank the reviewer for the positive comments. We will answer the reviewer's questions below.
>
> **Q1.** Our analysis is still inspired by and related to [1] but with some necessary changes to fit the shuffling method. However, whether the framework of [2] can be used to simplify the proof is unclear to us. We briefly explain why in the following.
>
>    - After a quick read, we think the core result of [2] is their Theorem 1, which relates the last-iterate convergence to the regret guarantee in online learning. As far as we can check, their Lemma 7 (the key to prove Theorem 1) relies on the fact that the gradient oracle $g_t$ is an unbiased estimator of $\nabla f(\boldsymbol{x}_t)$ conditioning on the history.
>
>    - In contrast, the gradient oracle in our setting is $g_t=\nabla f_{\textsf{i}(t)}(\boldsymbol{x}_t)$, which is unfortunately biased due to the shuffling-based index $\textsf{i}(t)$.
>
> As such, their analysis immediately failed in our setting. Hence, how to make the idea in [2] work under the shuffling scheme remains unknown currently.
>
> **Q2.** Indeed, the improvement (or deterioration) comes from both the analysis and the algorithmic change. Simply speaking, the current analysis that considers randomness naturally leads us to make the proximal update happen in every step. More precisely, if we want to utilize randomness in the analysis, it is natural to recognize every single step as an update (one can think about SGD as an example). Therefore, the difference between our algorithm and [3] arises since the latter's analysis is epoch-wise, which therefore requires the proximal update to happen at the end of every epoch.
>
>    - For RR, such a different view is enough to improve the convergence, as commented by the reviewer (also pointed out in our Section 5).
>
>   - But for SS, things become tricky. As shown by our results, the new view (and hence the variation in the algorithm) is enough to guarantee better rates in the small epoch regime, but is inadequate in the large epoch regime. It turns out that one critical point missed in the proof is the deterministic property of the shuffling method, i.e., the index in every epoch goes over the entire set $[n]$ (the key property used in [3]). Hence, we could expect a better result for SS if this fact is used in the analysis (as stated in the last paragraph in Section 5). However, this is not an easy task due to the algorithmic change, especially for a general $\psi$. But for some $\psi$ (including but not limited to $\psi=I_\mathcal{C}$ in Theorem 4.6), we still can make it work. For the most general case and more details, we kindly refer the reviewer to our Lemma B.3 and the discussion in Lines 1427-1445.
>
> **References**
>
> [1] Zamani, Moslem, and François Glineur. "Exact convergence rate of the last iterate in subgradient methods." arXiv preprint arXiv:2307.11134 (2023).
>
> [2] Defazio, Aaron, et al. "Optimal linear decay learning rate schedules and further refinements." arXiv preprint arXiv:2310.07831 (2023).
>
> [3] Liu, Zijian, and Zhengyuan Zhou. "On the Last-Iterate Convergence of Shuffling Gradient Methods." International Conference on Machine Learning. PMLR, 2024.

---

> > ### Comment · Reviewer_4WZ4 · 2025-04-02
> >
> > I would like to thank the authors for the answers. I think a score of 4 is appropriate for the work and wish you good luck.

---

### Official Review · Reviewer_JJsr · 2025-03-14

**Overall Recommendation:** 4

**Summary:**

- The paper investigates the convergence rates of shuffling SGD for nonsmooth (strongly) convex function. While the convergence behavior of shuffling SGD under smoothness assumption has been widely studied in recent literature, its investigation under Lipschitz continuity remains relatively less explored. The paper focuses on two shuffling strategies (RR and SS) in the context of the subgradient-proximal method.
- The paper makes a fine-grained analysis on $G_{f,1}$ and $G_{f.2}$ to establish better convergence rates compared to prior works. For RR, when the objective is convex, Theorem 4.2 achieves $O(\frac{\sqrt{G_{f,1}G_{f,2}}}{n^{1/4}K^{1/2}})$, and when the objective is strongly convex, Theorem 4.4 achieves $O(\frac{G_{f,1}G_{f,2}}{n^{1/2}K})$. Both rates improve upon previously known results, particularly when the Lipschitz cnostants of individual components are similar. Similarly, for SS, Theorem 4.5 and Theorem 4.7 demonstrates improved convergence rates over prior studies in both the convex and strongly convex setting, provided that the number of total epoch is small.

**Claims And Evidence:**

Most claims are supported by theorems and propositions.

**Essential References Not Discussed:**

Essential references are appropriately cited and discussed in the paper.

**Experimental Designs Or Analyses:**

The paper does not include any experiments.

**Methods And Evaluation Criteria:**

This paper is purely theoretical and does not involve empirical evaluation or benchmark datasets.

**Other Comments Or Suggestions:**

Minor typos:

- Line 1299, 1301: $y\rightarrow z_{t+1}$
- Line 1311: $T+2 \rightarrow T+1$

Minor Suggestion:

- In line 99L, the paper states that “For RR, our new rates are always better than the best-known bounds in (Liu & Zhou, 2024b) by up to a factor of $\Theta(n^{-1/4})$ in the general convex case.” I believe this sentence slightly overclaims the contribution, as there may be no gain when $G_{f,2}\approx \sqrt{n}G_{f,1}$. Thus, I suggest removing the term “always” from the sentence.

**Other Strengths And Weaknesses:**

Strengths

- The paper is well-written with clear and detailed explanations.
- The derived convergence rates for RR are strong. The results hold for the last iterate and general (strongly) convex $\psi$. Also, this paper is the first to prove that RR converges faster than Proximal GD in this setting.

Weaknesses

- The convergence rates for SS  seem weak, compared to those for RR. As the authors pointed out, both Theorem 4.5 and Theorem 4.7 do not guarantee convergence to $0$ as $T \rightarrow \infty$. While Theorem 4.6 provides a vanishing bound, it requires an additional condition on $\psi$. In particular, it is somewhat unusual that the convergence rate does not go to $0$ even in the strongly convex setting.

**Questions For Authors:**

Q1. In lines 831–849, the authors state that RR is slower than proximal SGD, at least in the convex setting. In contrast, under the smoothness assumption, RR has been extensively studied and shown to converge faster than SGD. Do the authors have any insights or intuition on why RR exhibits slower convergence under the Lipschitz continuity assumption?

Q2. Below Corollary 4.3, the paper claims that when $G_i \equiv G$, the result matches the lower bound $\Omega(\frac{1}{n^{1/4}K^{1/2}})$ shown by [Koren et al., 2022] proved for $\psi=I_C$. Does the lower bound construction in [Koren et al., 2022] also satisfy $G_i \equiv G$?

Q3. The authors clearly state the significance of their work when $G_{f,2} \approx G_{f,1}$. However, when $G_{f,2}\approx \sqrt{n}G_{f,1}$ (which is not well discussed in the paper), the results in the paper do not offer any improvement over prior works; for RR, both the rates in Theorem 4.2 and 4.5 match those in (Liu & Zhou, 2024b), and for SS, the critical epoch $K_*$ becomes 1. Do the authors believe that the current rate for this case is already optimal, or is there potential for achieving a better rate?

**Relation To Broader Scientific Literature:**

The paper improves the convergence rate of RR and SS compared to previously known rates under the Lipschitz continuity assumption. This assumption is more realistic in modern machine learning frameworks than traditional smoothness assumption.

**Theoretical Claims:**

I briefly checked the proofs of Lemma B.1, B.2, B.3, B.4, and B.6 (which I think serve as the backbone for proving the main theorems), and did not identify any critical issues. However, since the proof is highly technical, I was not able to fully verify the entire proof framework in detail.

---

> ### Author Rebuttal · Authors · 2025-03-29
>
> We thank the reviewer for the valuable feedback. We would like to answer the reviewer's questions below.
>
> **Typos&Suggestions.** Thanks for carefully reading and the helpful comments. We have corrected/modified them accordingly.
>
> **Q1.** Thanks for the interesting question. Honestly, we do not have too many insights on this phenomenon and don't want to provide a misleading explanation. We hope it can be fully understood in future research.
>
> **Q2.** Yes, as stated in Theorem 5 of [1], the lower bound construction is for $G_i\equiv 4$. We also mentioned this point in Remark d under Table 1.
>
> **Q3.** We note that $G_{f,2}\approx \sqrt{n}G_{f,1}$ is a very special case, but doesn't need to be worried about. Suppose we have $B$ gradient evaluation budgets, then the rate of every related algorithm is as follows (for simplicity, we consider the non-strongly convex case):
>
>    - GD: $O(\frac{G_{f,1}D}{\sqrt{T}})=O(\frac{\sqrt{n}G_{f,1}D}{\sqrt{nT}})=O(\frac{\sqrt{n}G_{f,1}D}{\sqrt{B}})$.
>    - SGD: $O(\frac{G_{f,2}D}{\sqrt{T}})=O(\frac{G_{f,2}D}{\sqrt{B}})\approx O(\frac{\sqrt{n}G_{f,1}D}{\sqrt{B}})$.
>    - Theorem 4.7 in [2]: $O(\frac{G_{f,1}D}{\sqrt{K}})=O(\frac{\sqrt{n}G_{f,1}D}{\sqrt{nK}})=O(\frac{\sqrt{n}G_{f,1}D}{\sqrt{B}})$.
>    - Our Theorem 4.2 for RR: $O(\frac{n^{1/4}\sqrt{G_{f,1}G_{f,2}D}}{\sqrt{T}})=O(\frac{n^{1/4}\sqrt{G_{f,1}G_{f,2}D}}{\sqrt{B}})\approx O(\frac{\sqrt{n}G_{f,1}D}{\sqrt{B}})$.
>
> As such, all algorithms have the same rate. Therefore, we believe our analysis for RR is optimal in this case.
>
> Lastly, for SS, our Theorems 4.5 and 4.7 are never optimal for whatever case. Hence, we only need to discuss Theorem 4.6. In this special case, as one can check, it degenerates to the same rate $O(\frac{\sqrt{n}G_{f,1}D}{\sqrt{B}})$ given above. Thus, we believe Theorem 4.6 in this special case is also unimprovable.
>
> **References**
>
> [1] Koren, Tomer, et al. "Benign underfitting of stochastic gradient descent." Advances in Neural Information Processing Systems 35 (2022): 19605-19617.
>
> [2] Liu, Zijian, and Zhengyuan Zhou. "On the Last-Iterate Convergence of Shuffling Gradient Methods." International Conference on Machine Learning. PMLR, 2024.

---

> > ### Comment · Reviewer_JJsr · 2025-04-03
> >
> > Thank you for the response. I raise my score to 4.

---

### Official Review · Reviewer_wDyR · 2025-03-14

**Overall Recommendation:** 4

**Summary:**

This paper studies shuffling-based variants of proximal SGD on nonsmooth convex optimization, where proximal SGD steps are taken using indices following randomly sampled or arbitrary permutations. Prior works on shuffling-based SGD mostly focus on smooth cases, and this paper tackles the more difficult nonsmooth case where Lipschitz gradients assumption is not available. The paper shows random reshuffling (RR) enjoys convergence faster than proximal GD for the first time in the literature, and proves that single shuffling (SS) converges faster than proximal GD at least in the "low epoch" regime, in both convex and strongly convex cases.

Unfortunately, even with the improvements, the rates are slower than proximal SGD (where samples are chosen with replacement); the presence of a lower bound on RR/SS (due to Koren et al 2022) suggests that it may be the case that proximal RR/SS is fundamentally slower than proximal SGD (with-replacement).

## Update after rebuttal
Reviewers were asked to update the reviews, but for this paper I have not much to add. I keep my positive evaluation.

**Claims And Evidence:**

I defer the discussion on the strengths and weaknesses of the developed theory to the "Strengths and Weaknesses" section. The paper does not contain any claims based on experimental results.

**Essential References Not Discussed:**

I don't know of other essential references that were not cited in the paper.

**Experimental Designs Or Analyses:**

N/A. No experiments included, which I think is not a shortcoming given that it's a theory paper.

**Methods And Evaluation Criteria:**

This paper does not propose a new method, and it analyzes existing methods theoretically. No empirical evaluation is considered necessary.

**Other Comments Or Suggestions:**

Some suggestions are made in the Weaknesses part above.

**Other Strengths And Weaknesses:**

Strengths
1. The paper is well-written and reads well.
2. The paper analyzes the shuffling-based proximal gradient method while taking the proximal operator at every iteration, not every epoch (which is the version analyzed in most existing results). I like it because it is closer to (with-replacement) Proximal SGD.
3. The new technique (Lemma 5.1 and the analysis on $\Phi$) sounds intriguing, because handling the index dependency within epochs has always been a huge bottleneck in the analysis of shuffling based algorithms, especially in the "low epoch" regime. I think the technique developed in the paper can have a broader impact beyond proximal SGD.
4. The paper discusses several interesting future research directions.

Weaknesses
1. The biggest shortcoming I can point out is that the SS convergence results (except for Theorem 4.6) are slightly disappointing in the sense that the rates do not converge all the way to zero as the number of epochs grows to infinity.
2. Minor clarity issue 1: upon looking at Table 1, I was confused for a long time why one cannot combine the "ANY" shuffling results by Liu & Zhou 2024b and the SS bounds shown in the paper to get the best of both worlds. I did not realize that the papers consider *different* algorithms until Section 4.2. Indeed, Section 3 points out the difference, but I failed to make the connection to the rates in Table 1. It would be helpful if the authors more explicitly emphasize the differences of the considered algorithms in different rows of Table 1.
3. Minor clarity issue 2: The fact that RR rates are slower (and perhaps fundamentally so) than proximal (with-replacement) SGD is not revealed to the readers until the end of Section 4.1. Although I appreciate the authors' honesty, I believe this should be highlighted earlier, because it is easy for readers to "extrapolate" their prior knowledge from the smooth case to the nonsmooth case and falsely assume that the "slower baseline" that the paper is talking about is SGD, not GD.

**Questions For Authors:**

Some minor questions:

1. One question about Koren et al (2022): In their Theorem 6(ii) on SS, the rate $O (\frac{GD}{ n^{1/4} K^{1/4}})$ is shown only for $K \geq n$. Can you elaborate on how you can derive the other term $\frac{GD}{\sqrt{n}}$ in Table 1?

2. The fact that SS bounds decrease all the way to zero as $K \to \infty$ (except for Theorem 4.6) is slightly disappointing. Is there any hope for improvements, or some other techniques that allow the bounds to converge to zero at least in the special case of $\psi \equiv 0$?

**Relation To Broader Scientific Literature:**

This paper studies popular variants of proximal SGD, so it may have some broader impact on other scientific areas that involve nonsmooth optimization.

**Theoretical Claims:**

I unfortunately did not have the time to check the details of the proofs in the supplementary material. I find the proof sketch convincing, but I am not entirely sure if the new technique developed by the authors is technically sound.

---

> ### Author Rebuttal · Authors · 2025-03-29
>
> We thank the reviewer for the endorsement of our work. We would like to address the reviewer's concerns below.
>
> **W1&Q2.** To make the bounds for SS decrease to $0$, it indeed needs some other technique in addition to the analysis sketched in Section 5 (as mentioned at the end of the same section). In a high-level way, we need to utilize both the randomness and the deterministic property of SS, in which the former is described in Section 5 and the latter refers to that the index goes over the whole set $[n]$ in every epoch. These two key points are formalized in Lemmas B.2 and B.3, respectively. As such, proving SS converging to $0$ when $K\to \infty$ is possible once the conditions in Lemmas B.2 and B.3 are fulfilled. A special case is $\psi=I_\mathcal{C}$ used in Theorem 4.6, which further includes the situation $\psi\equiv 0$ (i.e., take $\mathcal{C}=\mathbb{R}^d$). For a more detailed discussion and some inadequate points of Lemma B.3, we kindly refer the reviewer to Lines 1427-1445.
>
> **W2&W3.** Thanks for the suggestion. We will try to incorporate your comments in the revision when more space is available.
>
> **Q1.** On Page 39 of the arXiv version of [1] (or Page 20 of the supplementary of the NeurIPS version), they obtained the rate in the order of $O\left(\eta G^2 (\sqrt{nK}+K)+\frac{D^2}{\eta nK}\right)$ where $\eta$ is the stepsize. Thus, the best $\eta =\Theta\left(\frac{D}{G\sqrt{(\sqrt{nK}+ K)nK}}\right)$ gives us the rate $O\left(GD\sqrt{\frac{\sqrt{nK}+K}{nK}}\right)=O\left(\frac{GD}{n^{1/4}K^{1/4}} \lor \frac{GD}{\sqrt{n}}\right)$, where we use $O(a+b)=O(a \lor b)$ for $a,b\geq 0$. Moreover, we believe the statement $K\geq n$ in their Theorem 6 in the main text is a typo and should be corrected to $K\leq n$ as used in their supplementary (see the page we mentioned at the beginning).
>
> **References**
>
> [1] Koren, Tomer, et al. "Benign underfitting of stochastic gradient descent." Advances in Neural Information Processing Systems 35 (2022): 19605-19617.

---

> > ### Comment · Reviewer_wDyR · 2025-04-03
> >
> > Thank you very much for the response. I have decided to keep my positive score unchanged.

---

### Decision · Program_Chairs · 2025-05-01

**Decision:**

Accept (poster)

**Comment:**

The paper studies the last iterate convergence of proximal gradient methods for non-smooth (strongly) convex optimization problem, with Random Reshuffle (RR) and Single Shuffle (SS) schemes. All the reviewers recommend acceptance of the paper. Please incorporate the discussions into the final version of the paper.